# Lifting regenerative barriers promotes epithelial cell fate plasticity supporting lineage conversion

Maria T. Bejar [1,2] ✉, Paula Jimenez-Gomez [1,2], Ilias Moutsopoulos [1,3,8], Harikrishnan Ajith [1,2,8], Bartomeu Colom [4,7], Jamie McGinn[1,7], Seungmin Han[1,5], Greta Skrupskelyte [1,2], Fernando J. Calero-Nieto [1,3], Berthold Göttgens [1,3], Irina I. Mohorianu [1], Benjamin D. Simons [1,5,6] & Maria P. Alcolea [1,2] ✉

The ability of adult epithelial cells to rewire their cell fate programme in response to injury has emerged as a new paradigm in stem cell biology. This plasticity supersedes the concept of strict stem cell hierarchies, granting cells access to a wider repertoire of fate choices. Yet, in order to prevent a disordered cellular response, this process must be finely regulated. Here we investigate the little-known regulatory processes that restrict fate permissibility in adult cells, and keep plasticity in check. Using a 3D regenerative culture system, that enables co-culturing epithelium and stroma of different origins, we demonstrate that oesophageal cells exposed to the ectopic signals of the dermis are capable of switching their identity towards skin. Lineage tracing experiments and histological analysis, however, reveal that the oesophageal-to-skin lineage conversion process is highly inefficient, pointing to the existence of barriers limiting cell fate re-specification. Single-cell RNA sequencing capturing the temporality of this process shows that cells transitioning towards skin identity resist the natural progression towards tissue maturation by remaining in a persistent regenerative state marked by a particularly strong hypoxic signature. Gain and loss of function experiments demonstrate that the HIF1a-SOX9 axis acts as a key modulator of epithelial cell fate plasticity, restricting changes in identity during tissue regeneration. Taken together, our results reveal the existence of lineage conversion barriers that must be resolved for cells to respond to signals instructing alternative fate choices, shedding light on the principles underlying the full regenerative capacity of adult epithelial cells.

Traditionally, each adult cell type has been thought to emerge from a designated pool of lineage-restricted stem/progenitor cells (SCs)[1-7]. This model has been challenged by mounting evidence suggesting that cell fate, particularly in an epithelial context, is more dynamic than originally thought[8-11]. It has become widely accepted that, in response to certain tissue perturbations, epithelial cells retain the ability to reacquire developmental features, rewiring their programme of cell fate towards alternative

outcomes/identities in a process known as cell fate plasticity[12–22]. However, despite its unique potential for regenerative medicine, the mechanisms governing cell fate plasticity remain largely unknown.

Under physiological conditions, epithelial plasticity represents a well-established mechanism activated in response to tissue injury. Cells in the damaged area are granted access to a wider range of cell fate choices than those available under normal homeostasis[13–17,19–22]. Fate plasticity may include the reactivation of stem and regenerative properties in committed cells via de-differentiation[14–16,23,24], or enable cells to transdifferentiate into alternative identities in order to replenish adjacent tissue structures that may have been damaged[13,19–21]. Ultimately, this process expands the pool of cells that can respond to tissue injury beyond traditional SCs, providing a mechanism of defence that ensures the rapid and efficient replacement of lost or damaged tissue.

Plasticity, however, is a non-trivial process; increasing permissibility to acquire other fates predisposes epithelial tissues to disease and abnormal growth[8,9]. Hence, in order to safeguard integrity, tissues must be equipped with finely regulated mechanisms limiting the range of fates accessible to cells during regeneration. Unveiling the unknown processes that limit plasticity represents an outstanding challenge in stem cell biology, critical to unearthing the basic rules that dictate the full regenerative capacity of epithelial cells.

In an attempt to explore the limits of cell plasticity, previous studies have shown that changes in fate can go well beyond physiological constraints. Epithelial cells artificially exposed to stromal cues instructing alternative fates are able to switch their identity, giving rise to ectopic structures (such as hair, feathers, and secretory glands), not typically found in the native tissue[25–31]. These tissue recombination approaches, which combine epithelium and stroma of different origin, represent an ideal experimental model to investigate regenerative fate plasticity. However, the restricted number of re-specification events and limited sample availability in those assays represent a constraint[32]. The recent advent of single-cell sequencing technologies[33] has now opened the possibility to revisit tissue recombination methods to study cell fate plasticity with single-cell resolution at an unprecedented level of detail.

Here we develop an ex vivo regenerative assay that models oesophageal-to-skin lineage conversion to investigate the mechanisms regulating one of the most remarkable modes of adult cell fate plasticity, i.e., transdifferentiation. In this heterotypic system, based on a tissue recombination assay[7], we expose the squamous epithelium of the mouse oesophagus (OE; of endodermal origin) to the stroma of the murine skin, which contains niches for specific stem cell populations including hair follicles (HFs) and sebaceous glands (all of them of ectodermal origin). The uncomplicated architecture of the OE, lacking epithelial appendages (such as hair follicles or glands), makes it a particularly suitable model to investigate fate re-specification (Fig. 1A)[2,7,34–40]. Our data demonstrate that, although oesophageal cells are able to switch towards skin identity in response to dermal cues, this process is highly inefficient. Single-cell transcriptional profiling reveals that the majority of transitioning cells remain stalled between the two lineages in an unresolved regenerative state marked by the overexpression of HIF1a-regulated genes. Lifting the regenerative blockage, by inhibiting HIF1a or its downstream effector SOX9, favours cell fate re-specification, enabling further progression from oesophageal towards skin identity. Notably, SOX9, known to play a key role in hair follicle stem cell specification and maintenance in the skin[41–45], exhibited a context-dependent activity, regulating the capacity of oesophageal cells to undergo cell fate plasticity in response to perturbations. Our results reveal the existence of lineage barriers, inherent to tissue regeneration. We propose that these barriers may represent a mechanism to prevent cells from responding prematurely to inadequate or incomplete instructive cues during tissue repair.

## Results
### Adult oesophageal cells re-epithelialise the denuded dermis forming typical skin appendages
In order to investigate the initial mechanisms triggering the activation of adult cell fate plasticity in response to regenerative signals, we developed a heterotypic wound healing assay[7] to co-culture tissues of different origins. This enabled us to grow adult oesophageal tissue over the living denuded stroma of the skin. Based on earlier tissue recombination studies[25–31], we hypothesised that this heterotypic system would prompt cell fate re-specification towards skin identity, enabling us to capture early plasticity events.

Briefly, pieces of intact oesophageal tissue (including epithelium and stroma from the middle half of the oesophagus)[7] were placed over a larger piece of denuded skin dermis (Fig. 1B and Supplementary Fig. 1A) that had previously been peeled off from its overlying epidermis. The peeled dermis consistently retained the architectural footprint of its appendages and associated stromal niches (Supplementary Fig. 1B, C; see Methods), exposing the oesophagus to different dermal compartments that constitute niches to specific skin stem cell populations. These heterotypic tissue composites were cultured for up to 10 days under minimal media conditions (lacking any added growth factors)[7], over which period the emerging signs of oesophageal-to-skin lineage conversion events were explored. To be able to track the tissue of origin (OE vs skin), different reporter mouse lines were used; oesophageal tissues constitutively expressed tdTomato (red; *mTmG* or *nTnG* mouse strains), while skin derivatives were EGFP+ (green; *H2B-EGFP* mouse strain), unless otherwise stated (Fig. 1B). Similar tissue constructs where oesophageal tissue and skin were grown over their respective denuded stroma were used as controls (Supplementary Fig. 1A).

Over time, the OE growing over the dermis recapitulated the initial steps of a wound healing response[46], migrating out of their original tissue, progressively replacing the missing skin, and forming a de novo epithelium (Oesophageal-derived-skin; Fig. 1C and Supplementary Fig. 1D; see Methods). This newly formed epithelium was amenable for whole-mounting techniques (Fig. 1D), which enabled the plastic behaviour of cells to be studied at whole tissue level[7,35].

The emerging oesophageal-derived skin showed typical in vivo features, being sustained by basal cell proliferation and retaining low Caspase 3 activity (Supplementary Fig. 1E–I), which confirmed the integrity of the in vitro derived tissues. Interestingly, adult OE cells (red) were able to fully re-epithelialise the bare dermis and empty skin HF sockets, creating a de novo stratified epithelium (Fig. 1E) and forming structures reminiscent of in vivo HFs, hereafter referred as oesophageal-derived HFs (**oe**HFs; Fig. 1C–F). Re-epithelialisation of the denuded skin and HF niches by oesophageal cells in heterotypic cultures represented a highly efficient process, with $91.4 \pm 1.8\%$ of the original EGFP+ HFs being replaced by oesophageal-derived HFs-like structures (tdTomato; red) at day (D) 10 post-culture (Supplementary Fig. 1J).

Remarkably, oesophageal-derived hair follicles **oe**HFs (red) showed features reminiscent of the in vivo appendages. These included pigmented fibre-like structures (Fig. 1G and Supplementary Fig. 1K) positive for Fontana-Masson stain, which labels melanin (Supplementary Fig. 1L). Oesophageal-derived HFs were also positive for Oil Red O staining, indicating the accumulation of sebum in the proximal area where sebaceous glands are typically found (Supplementary Fig. 1M).

These results showed that the squamous epithelium of the adult oesophagus has the ability to grow and re-epithelialise the ectopic niche of the adult dermis, restoring the skin architecture and forming HF-like structures.

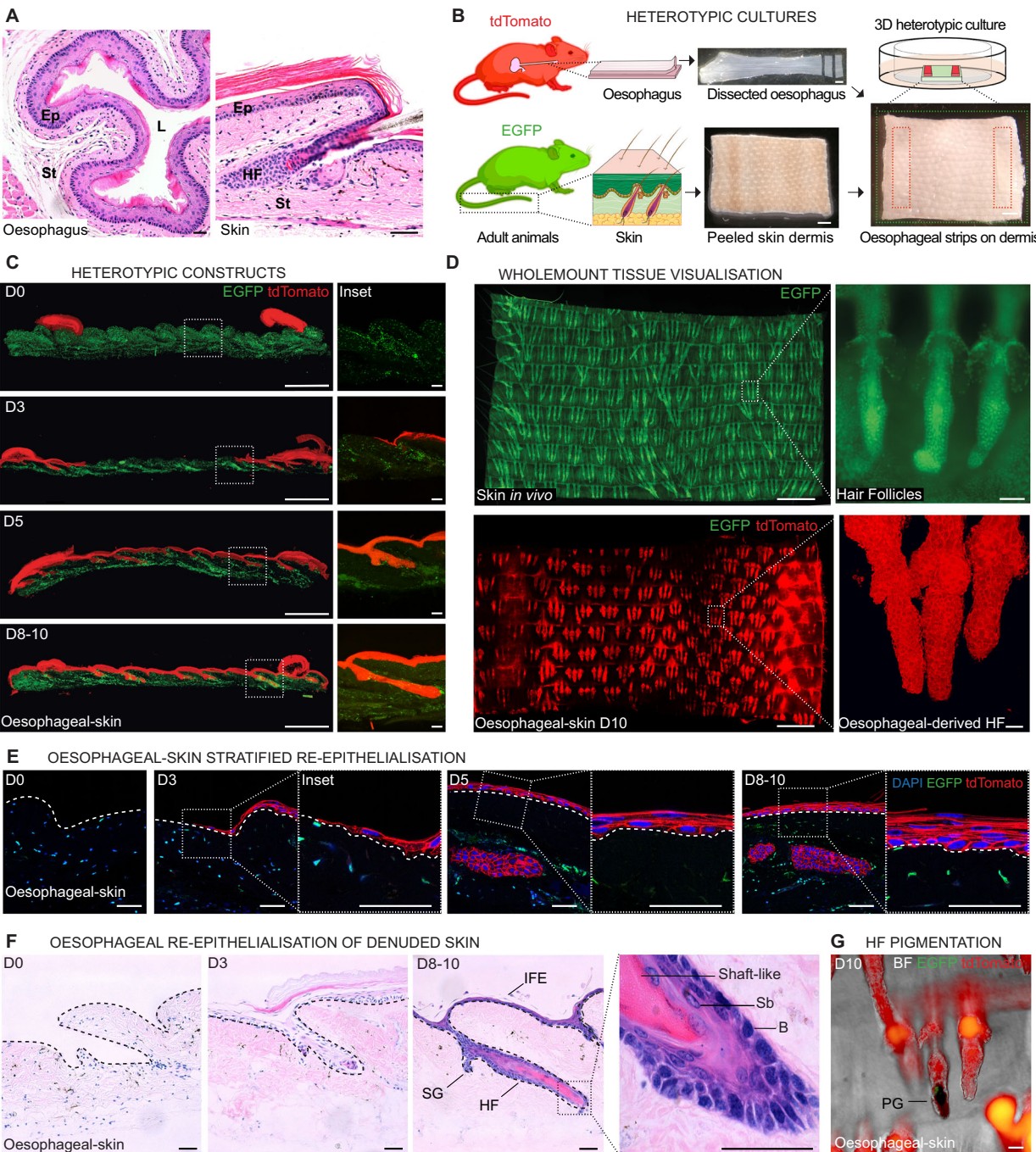

**Fig. 1 | Oesophageal-dermal 3D heterotypic culture as a model of ectopic niche regeneration. A** H&E sections of adult oesophagus (left) and skin (right) showing the characteristic tissue structure. Ep, epithelium; L, lumen; St, stroma; HF, hair follicle. **B** Schematic representation of the 3D heterotypic culture strategy. Two strips of oesophageal epithelium (5x1mm) obtained from adult tdTomato (red) reporter mice were laid over a piece of denuded/peeled tail skin dermis (7x9mm) from constitutive *H2B-EGFP* (green) adult mice, and cultured for up to 10 days in minimal media without added growth factors. During this time, new epithelium of oesophageal origin formed between the strips, re-epithelialising the dermis. Created in BioRender. Alcolea, M. (2025) https://BioRender.com/18bkcyd using two images adapted from Servier Medical Art (https://smart.servier.com), licensed under CC BY 4.0 (https://creativecommons.org/licenses/by/4.0/). **C** Tissue cryosections of heterotypic cultures at the indicated time points up to 10 days post-culture. Insets (dotted lines, right) illustrate the temporality of the dermal re-epithelialisation process by oesophageal-derived cells (red). EGFP skin dermis in green. **D** Basal projected views of typical wholemount confocal z-stacks from control *H2B-EGFP* adult skin (top) and oesophageal-derived skin (tdTomato; bottom) after 10 days (D10) in heterotypic culture as in (**B**). Insets (dotted lines, right) depict typical skin hair follicles (HF; top) and oesophageal-derived HF structures (bottom). **E** Tissue cryosections, as in (**C**), showing the progressive stratification during re-epithelialisation over time. Insets show higher magnification of oesophageal-derived interfollicular epidermis. **F** H&E sections of heterotypic cultures showing formation of oesophageal-derived skin over time. Dashed lines, basal membrane. Inset (dotted lines, right panel) depicts oesophageal-derived HF at endpoint. HF, hair follicle; IFE, interfollicular epidermis; SG, sebaceous gland; B, basal layer; Sb, suprabasal layer. **G** Brightfield (BF) image of oesophageal-derived skin shows pigmentation within hair follicle-like structures. PG, pigment. **Scale bars**. 50 μm (**A**, **E**, **F**, **G**, and insets in **C**, **D**); 1 mm (**B**, left panels in **C** and **D**). **Mouse lines**. Constitutive *H2B-EGFP* and *mTmG* mouse lines were used as sources of skin (green) and oesophageal tissue (red), respectively. See also Supplementary Fig. 1.

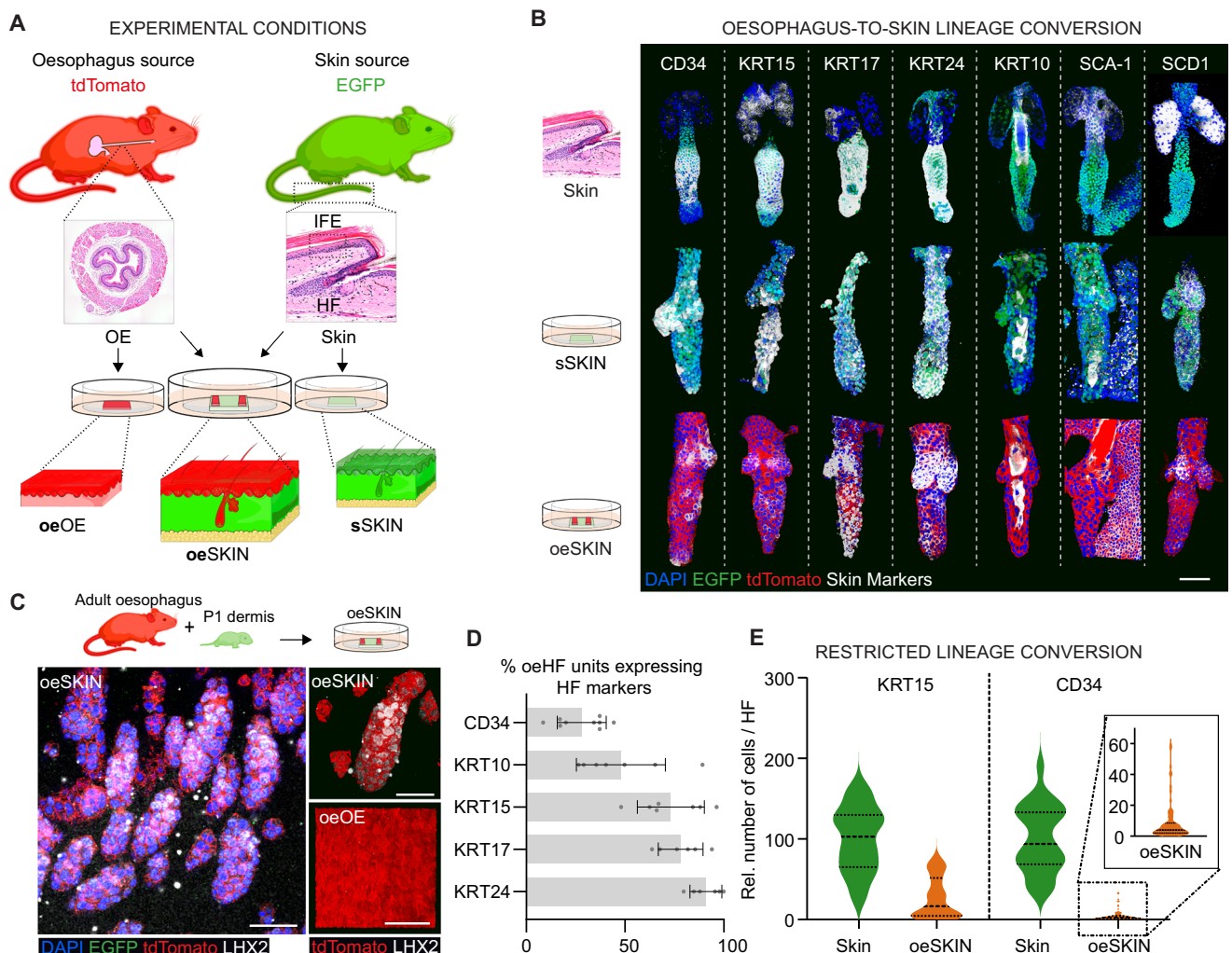

**Fig. 2 | Oesophageal cells undergo changes in cell identity when exposed to denuded dermal skin in vitro. A** Schematic illustrating the different in vivo and in vitro samples analysed across tissues. The experimental **oe**SKIN from heterotypic cultures (Fig. 1B) was compared to the respective in vivo and in vitro controls. OE (oesophageal epithelium) and Skin, represent in vivo control tissues. **oe**OE and **s**SKIN denote in vitro controls from the oesophagus and skin, respectively. **B** Oesophageal-derived skin shows expression of typical HF markers. 3D rendered confocal images of wholemount z-stacks showing expression of the typical HF and skin markers (CD34, KRT15, KRT17, KRT24, KRT10, and SCA-1), including the sebaceous gland marker SCD1. Wholemounts derived from in vivo EGFP skin (top row), in vitro EGFP epidermal-derived skin (**s**SKIN; middle row), and tdTomato oesophageal-derived skin from heterotypic cultures (**oe**SKIN; bottom row). Scale bar 50 μm. **C** Experimental schematic and 3D rendered confocal z-stacks images showing LHX2 expression in **oe**SKIN wholemounts (left and top right) derived from heterotypic cultures. Cultures derived from tdTomato oesophagus grown on EGFP

peeled dermis at postnatal day 1 (P; late morphogenesis). Note LHX2 marker expression is not detectable in oesophageal controls (tdTomato in vitro derived oesophageal epithelium; **oe**OE, bottom right). **D** Graph showing the percentage of **oe**HF units expressing CD34, KRT10, KRT15, KRT17, and KRT24 relative to total **oe**HF quantified from images in (**B**). Data expressed as mean ± SD. A minimum of 150 HFs and 6–9 replicates (all points shown) were quantified per marker from 3 mice. **E** Violin plots showing the distribution of CD34 and KRT15 positive cells per HF in in vivo skin and oesophageal-derived skin (**oe**SKIN; dashed line, median; dotted lines, quartiles). A minimum of 4000 cells were analysed per marker and sample type (skin and **oe**SKIN; $n = 4$–14 animals). **Fluorescent labels**. Blue, DAPI; Green, EGFP; red, tdTomato; Greyscale, HF markers. **Mouse lines**. Constitutive *H2B-EGFP* and *mTmG* mouse lines were used as sources of skin (green) and oesophageal tissue (red), respectively. Panels **A**–**C** created in BioRender. Alcolea, M. (2025) https://BioRender.com/k4mnofh See also Supplementary Fig. 2. Source data are provided as a Source Data file.

## Oesophageal-to-skin re-specification ex vivo; heterogeneous lineage conversion response

To further assess whether the adult OE cells re-epithelialising the adult dermis were actively switching their identity towards skin lineage, heterotypic epithelial wholemounts (**oe**SKIN; OE grown over denuded dermis) were immunostained for typical oesophageal and HF markers and compared to their respective in vivo tissue wholemounts (OE and SKIN) as well as the individual tissues cultured under comparable ex vivo conditions (**oe**OE and **s**SKIN; oesophagus and skin reconstituted over their own native stroma; Fig. 2A).

Confocal analysis assessing skin marker expression in **oe**SKIN wholemounts showed extensive signs of lineage conversion towards epidermal fate (Fig. 2B and Supplementary Fig. 2A–C). HF markers,

such as Keratin 15 (KRT15), Keratin 17 (KRT17) and Keratin 24 (KRT24)[47–49], as well as the well-known HF stem cell marker CD34[43,50] were specifically localised to the **oe**HFs in **oe**SKIN samples, supporting the notion of an active oesophageal-to-skin cell fate switch taking place. Other upregulated markers included SCD1 (which marks sebaceous glands), as well as Keratin 10 (KRT10), and SCA-1 (typically labelling the infundibulum and interfollicular epidermis) (Fig. 2B and Supplementary Fig. 2A–D).

Of note, **oe**HFs showed atypical morphology and ectopic expression, with markers localised outside their anatomical position within in vivo skin HFs. This, however, was observed to be a recurrent feature of ex vivo reconstituted HFs regardless of their skin or oesophageal origin (Compare **s**SKIN and **oe**SKIN *versus* Skin in Fig. 2B and

Supplementary Fig. 2A) in line with previous reports[51–55]. Reassuringly, oesophageal control samples in vivo (OE) and in vitro (oeOE) confirmed the lineage specificity of skin markers (Supplementary Fig. 2B, C)[56]. Additionally, **oe**HFs were positive for the basal epithelial marker KRT14, while failing to express the fibroblast marker PDGFRα, further verifying the epithelial origin of the CD34+ cells in **oe**SKIN (Supplementary Fig. 2E).

The upregulation of epidermal markers in the **oe**SKIN (**oe**IFE and **oe**HF) was accompanied by the downregulation of oesophageal markers, including Keratin 4 (KRT4, differentiation marker in the oesophagus; Supplementary Fig. 2F) and SOX2 (a basal oesophageal marker; Supplementary Fig. 2G, H). Overall, our marker characterisation analysis demonstrates that adult epithelial cells retain the ability to change their identity when exposed to an adult foreign niche, indicating that fate re-specification is not restricted to developing tissues[25,32].

Additional heterotypic experiments, where the adult OE was grown over the dermis of developing skin, one day post birth (P1), showed the specific upregulation of the marker LHX2, known to label developing HFs in vivo (Fig. 2C), in **oe**HFs. This observation revealed the critical role that the stroma plays in the identity switch, being able to rewire the **oe**HF fate towards developmental stages. In line with this, dermal papillae derived from the host skin (EGFP) were found in heterotypic cultures (Supplementary Fig. 2I, J). This indicates that the instructive signals promoting lineage conversion originate in the dermal niche, reminiscent of normal skin physiology[57,58]. Next, we sought to determine whether oesophageal plasticity was restricted to the skin or reflected a more widespread phenomenon. To this end, we repeated our heterotypic cultures, exposing oesophageal cells to the ectopic stroma of other epithelial tissues, i.e., tongue and bladder. At the culture end-point, oesophageal cells expressed Keratin 8 (KRT8), a marker typically expressed in simple epithelial tissues such as the bladder and glandular regions of the tongue (Supplementary Fig. 2K–M). Intriguingly, epidermal epithelium cultured on oesophageal stroma did not acquire oesophageal markers, suggesting intrinsic differences in plastic potential across epithelial tissues (Supplementary Fig. 2N). Despite tangible evidence proving the ability of oesophageal cells to undergo environmental reprogramming and switch towards skin identity, further inspection of the **oe**HF data revealed the limited nature of the fate conversion process. In particular, the expression of HF markers was highly variable across **oe**HFs appendages, ranging from widespread expression in the case of KRT17 and KRT24 ( ~ 78% and ~91% **oe**HFs expressing these markers, respectively), to only a reduced proportion of **oe**HFs (~28%) expressing CD34 (Fig. 2D). The expression was similarly variable within individual **oe**HFs, with the number of cells expressing CD34 and KRT15 representing just a minor proportion when compared to native skin (Fig. 2E). These results unveiled the inefficiency of the OE-to-skin lineage conversion process.

We conclude that adult oesophageal epithelial cells exposed to the ectopic stromal signals during tissue re-epithelialisation are able to actively switch towards alternative epithelial identities, as instructed by the stromal niche. The heterogeneous and limited expression of HF markers in oesophageal-derived HFs, however, revealed the restricted nature of the cell fate plasticity programme during epithelial regeneration.

### Polyclonal contribution of oesophageal cells to de novo hair follicle formation

To better understand the lineage conversion process, we went on to investigate the OE contribution to **oe**HFs formation. To this end, we traced the behaviour of individual cells in heterotypic cultures using a genetic lineage tracing approach[40]. $R26^{CreERT}/R26^{Confetti}$ mice were treated with tamoxifen resulting in OE clonal cell labelling 48 h post-induction (Fig. 3A); approximately 5 in 100 cells expressed one of the four fluorescent confetti reporters (YFP, CFP, RFP or GFP)[59]. Labelled oesophagi were collected and cultured over wild-type denuded dermis for 8 days. Clonal labelling was subsequently analysed by confocal microscopy to track the behaviour of the cells forming the **oe**SKIN. The data revealed that over the 8-day time course, OE labelled cells experienced a marked expansion (Fig. 3B). Starting at an initial clonal labelling covering just ~5% of the basal layer, fluorescent OE cells expanded and re-epithelialised the dermis colonising on their way two-thirds of all the **oe**HF structures present in the tissues (66.37% ± 6.07; mean ± SD). As a result, upon clonal expansion, approximately 66% of all **oe**HFs formed contained at least one or more labelled cells, revealing that, in line with previous observations in oesophageal wound healing[7], OE cells undergo a widespread recruitment during **oe**HF formation.

To further understand the cellular dynamics behind **oe**HF formation, we next asked whether these structures were of monoclonal or polyclonal origin. We anticipated that if **oe**HFs emerged from a discrete cell population in the tissue, monoclonal **oe**HFs derived from single cells (i.e., entirely labelled by a single fluorescent reporter) would be found. The results, however, showed that all labelled **oe**HFs analysed comprised a minimum of two and up to five clonal units (Fig. 3C, D), arguing in favour of the polyclonal origin of newly formed **oe**HFs.

Collectively, our results indicated that, during the lineage conversion process, oesophageal cells expand contributing polyclonally and in a widespread manner to de novo **oe**HF formation.

### Lineage conversion in oesophageal-derived HFs is dictated by the dermal niche

So far, our data demonstrates that, despite the extensive oesophageal contribution to dermal regeneration, the ability of OE cells to switch their identity towards skin remains strictly limited. Given the inefficiency of the process and the emerging recognition of heterogeneity within the oesophageal basal cell population[60–62], it is important to consider the dynamics of basal cell contribution to the lineage conversion process. To determine whether the basis of the inefficient OE-to-skin fate re-specification was associated with restricted competency of a heterogeneous cell population, we analysed the KRT15 and CD34 expression within **oe**HF confetti labelled clones. We reasoned that if the conversion process was linked to a particular cell population, cells originally primed to undergo lineage conversion would produce clones preferentially enriched for cells expressing HF markers (KRT15 and CD34), indicative of their clonal origin. Interestingly, neither KRT15, nor CD34 expression were linked to particular clonal events, with areas enriched for KRT15 or CD34 being largely polyclonal instead (Fig. 3E, F). Additionally, clones largely negative for CD34 were seen to express this marker in sporadic cells positioned in close proximity to other CD34+ cells, forming clusters of cells undergoing lineage conversion independent of their clonal origin (Fig. 3G). A similar level of marker polyclonality was also observed in in vitro skin controls (sSKIN; Fig. S3A–C), where cell fate plasticity in response to tissue repair is well established[13,20,63]. Overall, these results suggest that, in our regenerative 3D assay, marker expression is context dependent, i.e. relies on signals from the dermal niche, rather than being a clone-specific process.

To demonstrate that marker expression, indeed, relied on niche signals, we repeated our heterotypic culture assay using decellularised dermis devoid of stromal cells (Supplementary Fig. 3D). The results showed that OE cells cultured over the acellular dermal scaffold exhibited an impaired re-epithelialisation, **oe**HF formation, and lacked CD34 expression (out of 82 **oe**HFs analysed, none expressed CD34; Supplementary Fig. 3E–H), supporting the instructive role of dermal signals. Interestingly, this data also shows that, under the right experimental conditions, adult stromal cells retain the ability to instruct identity changes in adult epithelia, beyond developmental stages[25,32].

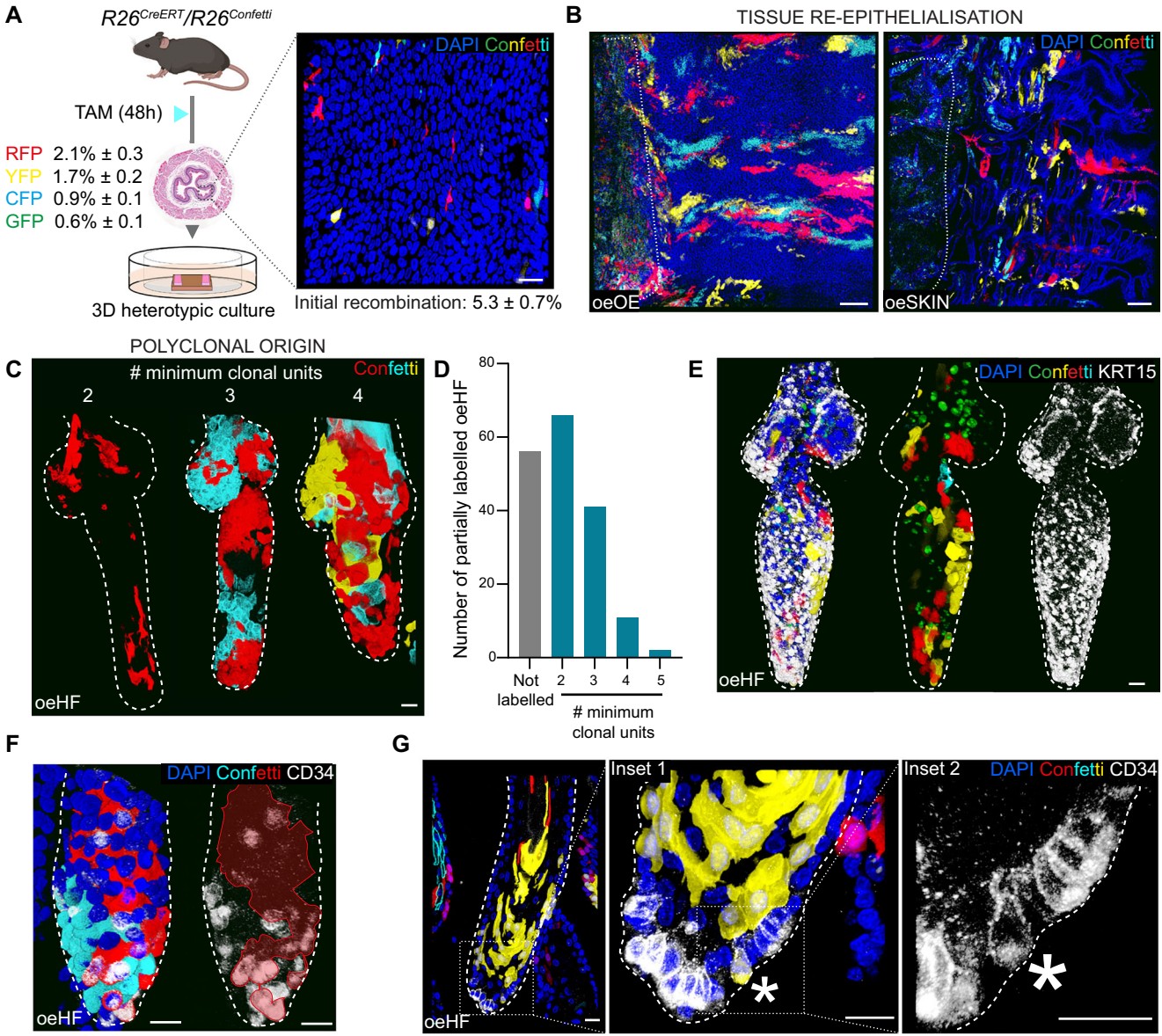

**Fig. 3 | Oesophageal-to-skin lineage conversion is polyclonal and instructed by skin dermal signals. A** *R26^CreERT^/ R26R^Confetti^* mice were administered tamoxifen (TAM) and oesophagi were harvested 48 h post-induction to establish 3D heterotypic cultures. Inset (right) shows 3D rendered basal view of an OE wholemount at the time of collection. Initial recombination efficiency is indicated as percentage of labelled cells for each of the Confetti reporters (CFP (cyan), GFP (green), YFP (yellow) and RFP (red)) relative to OE basal cells. Data expressed as mean ± SEM; n = 4920 basal cells out of 3 animals. Created in BioRender. Alcolea, M. (2025) https://BioRender.com/2gfnch4. **B** Epithelial wholemount 3D rendered images of oesophageal epithelium grown over native stroma (**oe**OE; left) or over the denuded skin dermis (**oe**SKIN; right) 8 days post-culture. Images show widespread oesophageal cell contribution during the re-epithelialisation process. Dotted lines mark original oesophageal tissue strips. **C** Representative 3D rendered side views showing polyclonal contribution to **oe**HF. **D** Number of reporters found to be labelling individual **oe**HF structures. "**Not labelled**" indicates

that no reporter was found to be expressed. Due to clonal fragmentation, clonal events are reported as the "**minimum number of clonal units**". Not labelled areas were considered minimal clonal units if coexisting with another reporter. Data is shown as absolute number of **oe**HF. A total of 178 **oe**HF structures from 4 animals were analysed. **E**, **F** 3D rendered side views of confocal z-stacks showing polyclonal contribution within KRT15+ (**E**) and CD34+ (**F**) cells in **oe**HFs. KRT15+ and CD34+ cells were not associated to clonal events. **G** Representative confocal images showing side views of **oe**HF depict CD34- clonal unit (in yellow) containing an individual CD34+ cell (marked by asterisk; dotted lines, insets to the right). Dashed lines delimit hair follicles borders. **Fluorescent labels**. Blue, DAPI; Cyan, Green, Red and Yellow depict Confetti clones; Greyscale, CD34 or KRT15. **Scale bars**. 20 μm (**A**,**C**, **E**, **F**, and **G**), 200 μm (**B**). **Mouse lines**. C57BL/6 wild-type mice were used as source of tail dermis. 48h-induced *R26^CreERT^/R26^Confetti^* mice were the source of oesophageal tissue. See also Supplementary Fig. 3A–C. Source data are provided as a Source Data file.

Together, these results indicate that the change in identity depends on niche signals from the skin dermis, rather than on the oesophageal cell of origin. While we cannot discard that multiple oesophageal basal cell pools may contribute to lineage conversion, our lineage tracing data suggest no single basal subpopulation is solely responsible for this process. Remarkably, the observation that, out of all the OE cells exposed to the dermal HF niche, only a reduced proportion were able to acquire CD34 and KRT15 expression (Fig. 2D, E)

suggests the existence of inherent barriers limiting the lineage conversion process.

## Single-cell RNA sequencing identifies a regenerative blockage limiting oesophageal-to-skin lineage conversion

To investigate the nature of any existing barriers affecting cell fate respecification, we first explored transcriptional changes in cells exposed to different niches, (endogenous *versus* ectopic), using our

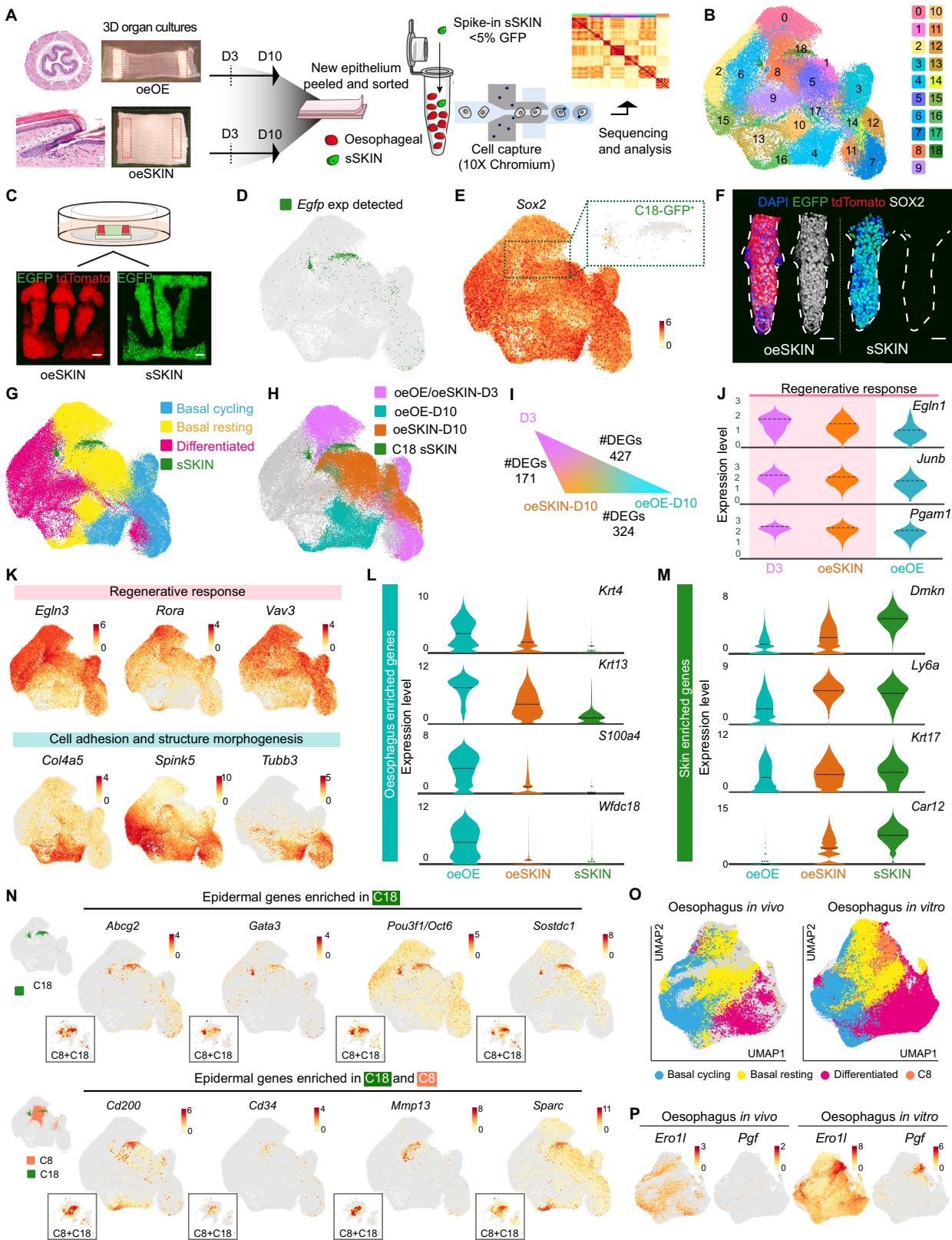

heterotypic model. For this, single-cell RNA sequencing (scRNA-seq) was performed on tdTomato OE (from *nTnG* mice) growing over its own denuded stroma (**oe**OE) or the skin dermis (**oe**SKIN) at two different time points; early during epithelial regeneration at day (D) 3, and after full re-epithelialisation at D10 (experimental end-point; Fig. 4A). Sequenced samples were spiked with 5% EGFP epidermal **s**SKIN cells (i.e. ex vivo skin grown over its endogenous stroma; Fig. 4C

and Supplementary Fig. 3I). The **s**SKIN cells served as a reference control to pinpoint OE cells transitioning towards skin identity (Fig. 4A–C; Source Data). The EGFP **s**SKIN reference cells were found in a unique cluster (C), C18 (Fig. 4B–F and Supplementary Fig. 3J, K); defined as cells presenting high levels of *Egfp* and *Gt(ROSA)26Sor* (Fig. 4D and Supplementary Fig. 3J) that lack of *Sox2* expression (Fig. 4E, F)[64–66]. The identity of C18 cells was further validated by

**Fig. 4 | Single-cell transcriptional analysis reveals a regenerative blockage in oesophageal-derived skin. A** Overview of the scRNA-seq workflow (10x Genomics). Viable epithelial **oe**OE and **oe**SKIN cells from organotypic cultures were sorted for tdTomato+ and EGFP+ at days 3 and 10 (D3, D10). tdTomato+ samples were spiked with ~5% EGFP+ epidermal-derived **s**SKIN cells as in vitro skin reference. Minimum of 3 libraries per sample and time point (x10 Chromium). **B** UMAP representing 19 cell clusters (Louvain, 0–18). **C** Representative 3D-rendered images of tdTomato **oe**SKIN and **s**SKIN grown on EGFP dermis ex vivo as heterotypic cultures. **D** Cluster 18 contains **s**SKIN reference control cells (EGFP+). UMAP showing enriched *Egfp* gene expression. **E** UMAP showing *Sox2* expression. Zoomed-in inset (dotted lines) shows negligible expression in EGFP **s**SKIN, C18. Colour bar indicates log2-transformed normalised expression levels. **F** Representative 3D-rendered images reveal SOX2 expression remains as a distinctive marker distinguishing **oe**SKIN (left) and **s**SKIN (right). **G** Annotated cell-type distribution across UMAP space. **H** UMAP showing transcription patterns by sample type (**oeOE/oeSKIN-D3**, **oeOE-D10** and **oeSKIN-D10**), and **s**SKIN C18. **I** Differentially expressed genes (DEGs) in basal cells shows similarity between **D3** and **oeSKIN-D10** compared to **oeOE-D10**. **J** Violin plots showing tissue regeneration and maturation gene expression (genes enriched in **D3** and **oeSKIN-D10** shaded in pink). Log2-transformed normalised expression data. Dashed lines represent mean values. **K** UMAPs of selected DEGs upregulated in **D3**+**oeSKIN-D10** (pink) or **oeOE-D10** (cyan). Colour bars indicate log2-transformed normalised expression levels. **L, M** Violin plots of oesophageal (**L**) and skin (**M**) enriched genes reveal identity differences between **oeOE-D10** and **oeSKIN-D10**. (**N**) UMAPs of classical epidermal (top), and hair follicle-related (bottom) genes enriched in cluster 18 (reference) and 8 (transitioning). Colour bars indicate log2-transformed normalised expression levels. Insets show zoomed cells in C8/C18. **O, P** UMAPs after integration with in vivo oesophageal scRNA-seq dataset (McGinn et al., 2021). Colour bars indicate log-transformed normalised expression. **Scale bars**. 20 µm. **Mouse lines**. Constitutive *H2B-EGFP* were used as source of tail dermis, and *nTnG* of oesophageal tissue. Panels A and C created in BioRender. Alcolea, M. (2025) https://BioRender.com/huzygrn See also Supplementary Figs. 3I–M and 4A–D. Source data are provided as a Source Data file.

---

implementing a tdTomato+/EGFP+ semi-supervised gene classifier approach (Supplementary Fig. 3K; see Methods).

Having identified the **s**SKIN reference population, we went on to annotate oesophageal-derived cells. Consistent with in vivo observations[7,36], there were three distinctive oesophageal cell populations in all sample types (Fig. 4G; Source Data; Methods): i) actively cycling basal progenitors; ii) resting basal cells; and iii) differentiated cells (Supplementary Fig. 3L, M).

The UMAP distribution and differential gene expression analysis of the single-cell data revealed the existence of three distinctive transcriptional patterns across the experimental conditions analysed (Fig. 4H): **oeOE/oeSKIN-D3** denoted an early re-epithelialisation state at D3 post-culture (Supplementary Fig. 4A, left panels), **oeOE-D10** and **oeSKIN-D10** represented oesophageal cells grown over their own denuded stroma or that of the skin dermis at D10 post-culture, respectively (Supplementary Fig. 4A, right panels). Interestingly, although D3 samples showed an indistinguishable profile independent of the origin of the stroma used as substrate (**oe**OE *versus* **oe**SKIN), there were marked differences at D10 (Fig. 4H and Supplementary Fig. 4A). Further analysis of the basal cell compartment, where progenitor cells reside[7], showed that OE cells grown over the skin niche, **oeSKIN-D10**, showed a gene expression pattern more closely related to early timepoints, **oeOE/oeSKIN-D3**, than their timepoint counterpart **oeOE-D10** (Fig. 4I). Indeed, gene enrichment analysis of the associated gene expression patterns revealed that both, **D3** and **oeSKIN-D10**, shared a strong regenerative signature, being particularly enriched for pathways involved in active wound healing and tissue repair (Fig. 4J, K and Supplementary Fig. 4B, D). **oeOE-D10** samples instead showed a natural progression away from a regenerative state marked by an increased expression of genes associated with epithelial differentiation, keratinisation and tissue maturation (Fig. 4K **and** Supplementary Fig. 4B–D).

We concluded that oesophageal-derived skin fails to progress to the same extent as **oeOE-D10** samples, remaining locked in an unresolved regenerative state where cells become permissive to environmental cues instructing an identity switch. Indeed, marker gene expression analysis showed that, as **oeSKIN-D10** transitions towards skin identity, oesophageal marker expression drops, while epidermal genes become upregulated when compared to their endogenous control **oeOE-D10** (Fig. 4L, M).

In order to determine the specific mechanisms underlying the lineage conversion process, it was first necessary to define the set of oesophageal cells undergoing the transition towards skin/HF lineages and establish the transcriptional changes that set them apart. To this end, we searched for OE-derived clusters enriched in epidermal gene expression, as defined by the reference cluster **s**SKIN-C18. Cluster 8 (C8) emerged as a cluster enriched in cells transitioning towards skin identity (Fig. 4N). Indeed, the **s**SKIN-C18 reference cluster was found to partially merge with C8 of oesophageal origin, marking the UMAP area where cells undergoing fate conversion are located (Fig. 4N). In line with our histological characterisation (Fig. 2), the number of oesophageal cells showing signs of conversion was limited, illustrating the inefficiency of the lineage conversion process at the molecular level.

Given that C8, which contained transitioning cells potentially representing a plastic state, was present in both oesophageal cells grown on dermis (**oe**SKIN) as well as on their native stroma (**oe**OE), we went on to explore the existence of any potential physiological counterparts in the in vivo oesophageal epithelium. To this end, we integrated the current in vitro dataset with our previous in vivo scRNA-seq data, including adult homeostatic tissue as well as earlier postnatal stages[36]. This analysis revealed that, although cell clusters were largely comparable between the two datasets, the signature of C8, marked by the upregulation of tissue repair genes, was unique to in vitro conditions (Fig. 4O, P). This indicates that C8 constitutes a state not present under normal homeostatic conditions, that is instead associated with the active regenerative state imposed by the ex vivo system.

To investigate whether the oesophageal-to-skin lineage conversion process involved epigenetic reprogramming, we performed single-cell assay for transposase-accessible chromatin using sequencing (scATAC-seq) to profile genome-wide chromatin accessibility in in vitro oesophageal-derived skin (**oe**SKIN), alongside control samples of in vitro oesophagus (**oe**OE) and in vitro skin (**s**SKIN) (Fig. 5A and Supplementary Fig. 4E, F). The data revealed that, upon exposure to ectopic skin stroma, oesophageal cells exhibited altered chromatin accessibility at 29.9% of genomic regions compared to **oe**OE. Notably, 38.7% of these differentially accessible regions align with those characteristic of the **s**SKIN chromatin landscape (Supplementary Fig. 4G; Source Data). In line with the observed lineage conversion, we detected increased accessibility at loci associated with epidermal identity and decreased accessibility at loci linked to oesophageal identity (Fig. 5B, C and Supplementary Fig. 4G). Additionally, the scATAC-seq analysis further revealed the inefficiency of the lineage conversion process, with only a limited number of oesophageal cells transitioning towards skin identity (see cluster 6; Supplementary Fig. 4E, F). This supports the existence of inherent barriers to cell fate plasticity.

Since our data, so far, suggested that **oe**SKIN cells remain locked at a regenerative state, we next asked whether altering the in vitro regenerative state, associated with dermal re-epithelialisation and de novo skin reconstitution, influences the lineage conversion process. To this end, we adapted our ex vivo system to limit the ability of the OE to re-epithelialise the dermis. This was achieved by using peeled OE lacking the underlying stroma, which is known to promote cell migration and epithelial expansion during wound healing[46,67]

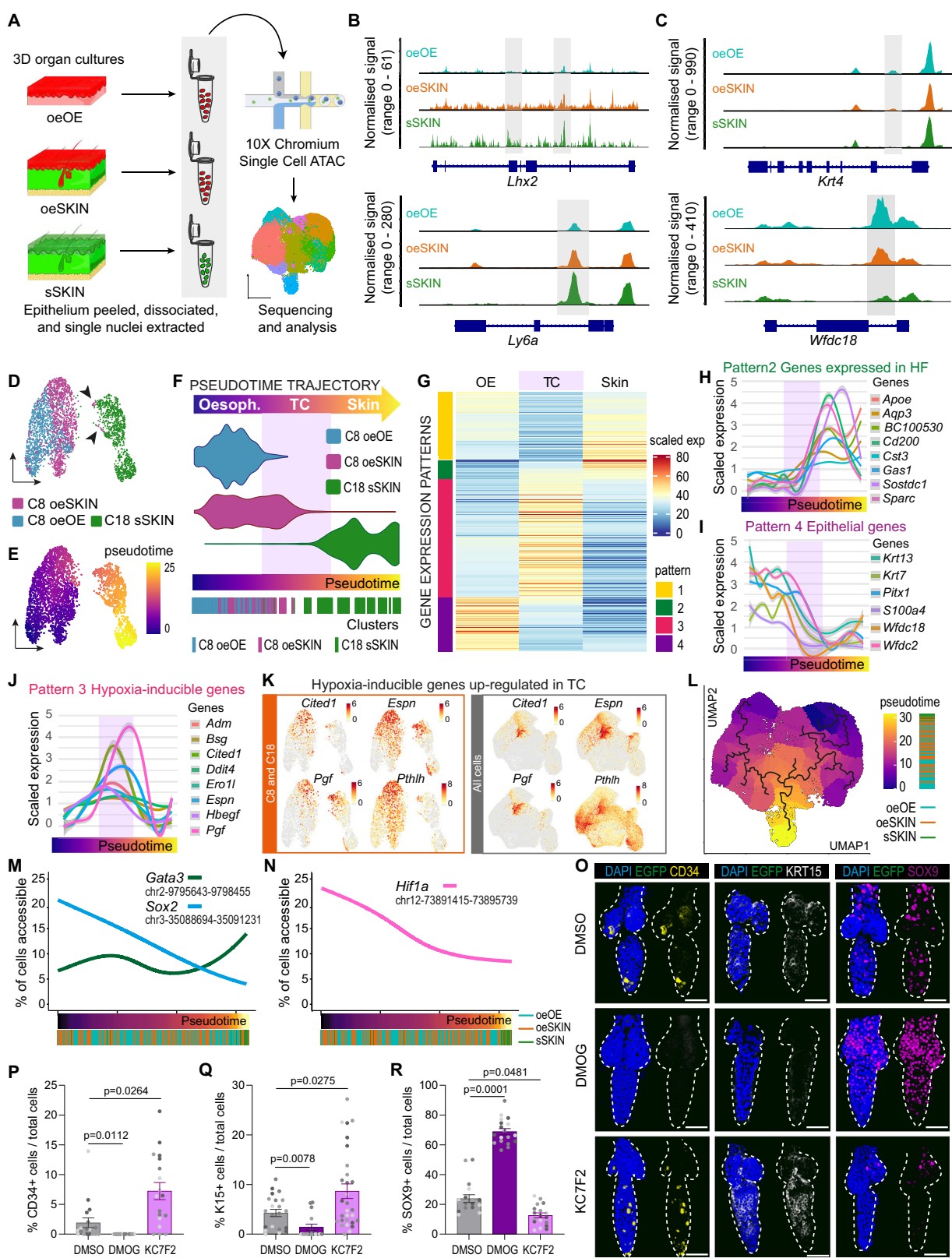

(Supplementary Fig. 4H, I). Indeed, when growing OE under impaired re-epithelialisation conditions, the OE formed **oe**HFs that contained a significantly higher number of CD34+ cells compared to our standard heterotypic approach (Supplementary Fig. 4J, K). These results reveal that, by partially restricting the re-epithelialisation capacity of epithelial cells, the fate conversion process becomes more efficient.

Altogether our data shows that, in line with other studies on epithelial cell fate plasticity[9,68], the oesophageal-to-skin fate transition is also associated with cells displaying a regenerative signature. However, we further demonstrate that, although tissue regeneration is known to promote epithelial plasticity[9,10], it also bears intrinsic fate barriers preserving cell identity and restricting re-specification events.

**Fig. 5 | Oesophageal cells transitioning towards skin identity show a marked hypoxic signature. A** Overview of the scATAC-seq workflow (10x Genomics). Epithelial **oe**OE, **oe**SKIN and **s**SKIN cells nuclei were isolated at day 10, libraries prepared (x10 Chromium) and sequenced. Created in BioRender. Alcolea, M. (2025) https://BioRender.com/p4w9gy0. **B**, **C** Track plots showing chromatin accessibility for skin-related (**B**), and oesophageal-related (**C**) loci across sample types. Peaks with differential accessibility are shaded. **D** UMAP of C18/C8 cells after re-clustering. Colours show original clusters and sample origin. **E** Pseudotime trajectory for C18/C8 cells on UMAP. Colour bar denotes pseudotime axis. **F** Violin plots showing distribution of cells along the pseudotime trajectory for each sample type, split into Oesophageal identity, Transitioning cells (TC; purple shade) and Skin identity. Rug plots show individual cells along the trajectory. **G** Heatmap of genes defining 4 main expression patterns along pseudotime trajectory. Data expressed as log-transformed normalised expression levels, averaged across all basal cells and scaled to a range [0–100]. **H**–**J** Expression of representative genes from Pattern 2 (skin, **H**), 4 (oesophageal, **I**) and 3 (transitioning, **J**). Data presented as auto-scaled, log2-transformed normalised, smoothed (generalised additive model). Shaded grey bands in line plots indicate 95% confidence intervals. **K** UMAPs of hypoxia-inducible genes (pattern 3) up-regulated in transitioning cells (TC). Left, C8 + C18 UMAP reclustering; right, UMAP including all clusters. Colour bars indicate log2-transformed normalised expression levels. **L** scATAC-seq pseudotime trajectory on UMAP with rug plot showing pseudotime order of individual cells and coloured by sample origin. **M**, **N** Line plots of pseudotime-dependent accessibility for skin and oesophagus related transcription factors (**M**) and HIF1a (**N**) loci with accessibility-driven rug plots, coloured by sample. **O** 3D confocal side views of CD34, KRT15 and SOX9 in **oe**HF after HIF1a activator (DMOG) or HIF1a inhibitor (KC7F2) treatment. Vehicle = DMSO. **P**–**R** Quantification of CD34+ (**P**), KRT15+ (**Q**) and SOX9+ (**R**) cells per **oe**HF (% vs DAPI + ). ≥ 1600 cells from ≥ 18 **oe**HFs quantified; n = 4 mice. Points show individual measurements (greyscale = biological replicate). Data expressed as mean ± SD. Kruskal-Wallis with multiple comparisons. **Scale bars** 50μm. **Fluorescent labels**. Blue, DAPI; Yellow, CD34; Greyscale, KRT15; Magenta, SOX9. **Mouse lines**. Constitutive *H2B-EGFP* mice were used as source of tail skin. *mTmG* (scATAC-seq) and C57BL/6 wild-type (HIF1a treatments) mice were the source of oesophageal tissue. Dashed lines delimit hair follicles. See Supplementary Fig. 5. Source data are provided as a Source Data file.

We propose that upon stress, epithelial tissues establish a trade-off response that balances plasticity and fate-keeping to ensure the adequate repair of the tissue. Tissue regeneration may, by default, prevent free access to alternative identities, requiring additional regulatory processes to fine-tune cell fate plasticity and efficiently repair tissue damage.

## Oesophageal cells transitioning towards skin identity present a marked hypoxic signature

With the aim to define the specific mechanisms dictating the permissibility of OE cells to signals instructing alternative fates, we focused on the transition between C8 and C18 marking the switch from oesophageal to epidermal identity at D10 (Fig. 5D). The new re-clustered UMAP segregated cells by tissue of origin, showing cells of different origin at the UMAP boundaries. Pseudotime analysis further revealed the trajectory along the cell fate conversion process; with cells of different identity plotting at opposite ends of the pseudotime path (i.e., **s**SKIN-C18 and **oe**OE-C8), and transitioning **oe**SKIN cells arranged between the two (**oe**SKIN-C8; Fig. 5D–F). To further validate this fate trajectory, and avoid any potential confounding effects as a result of including cells derived from different tissues, the pseudotime analysis was repeated only including cells of oesophageal origin this time (Supplementary Fig. 5A–C). The UMAP representation of the oesophageal pseudotime also showed a clear segregation between OE cells exposed to their own stroma (**oe**OE, blue; Supplementary Fig. 5A–C), and those exposed to the ectopic stroma of the skin (**oe**SKIN, magenta; Supplementary Fig. 5A–C), reproducing the trajectory obtained when the 3 experimental conditions were included (Fig. 5D–F).

Next, we examined the transcriptional profile along the three main in vitro states defined by the pseudotime trajectory, namely **Oesophagus, Transitioning Cells** (TC) and **Skin** (Fig. 5F; see Methods). We identified four major patterns defining changes in the gene expression profile across states (Fig. 5G and Supplementary Fig. 5D). Pattern (P) 2 marked the switch in cell identity by showing a sustained increase in the expression of typical HF genes (*Cd200, Apoe, Sparc, BC100530, Gas1, Sostdc, Cst3, Aqp3*)[69–72] (Fig. 5H). In contrast, P4 revealed a progressive reduction in those epithelial genes found to be enriched in oesophageal cells (**oe**OE; *Krt7, Krt13, Pitx1, S100a4, Wfdc18, Wfdc2*; Fig. 5I)[73]. P1 and P3 represented the most relevant patterns, containing genes being specifically modulated in TCs undergoing cell fate conversion (Fig. 5J and Supplementary Fig. 5E–H). Downregulated genes in P1 were related to processes associated with epithelial differentiation[74], including protein biogenesis (*Ddx21, Gnl3, Ncl, Nop56, Nop58, Rexo2*)[75] and oxidative metabolism (*Mgst1, Mgst2, Prdx6*)[76] (Supplementary Fig. 5E). P3, instead, contained genes upregulated in TCs. Gene enrichment analysis for these genes revealed processes known to be associated with cell fate plasticity[77,78] (Fig. 5J and Supplementary Fig. 5F–H). This included terms defining an active tissue remodelling signature i.e., wound healing, invasion, EMT and tumour progression (*Hbegf, Serpine1, Bsg, Plod2, Pthlh, Cited1*)[79–84]. Other genes denoted the active regenerative state of TCs, including those associated with ECM reorganisation (*Adam8, Crim1, Itgb1, Mmp3, Mmp9, Mmp13, Pthlh*)[85,86] and glycolytic metabolism (*Aldoc, Gapdh, Gpi1, Pgam1, Pkm, Tpi1*) (Supplementary Fig. 5F–H). Interestingly, one of the most significant transcriptional features defining transitioning cells was their marked hypoxic profile, expressing particularly high levels of genes regulated by HIF1a (*Adm, Bsg, Cited1, Ddit4, Ero1l, Espn, Hbegf, Pgf, Bnip3, Cav1, Egln3, P4hb, Prelid2, Ptges, Slc2a1, Vegfa*; Fig. 5J, K)[87–91]. This emerged as a common hallmark, shown to underlie all cells in C8, that was notoriously more accentuated in cells transitioning towards skin identity (TC; Fig. 5J, K and Supplementary Fig. 5G–I; Source Data). In keeping with these findings, we observed that chromatin regions associated with identity switch loci, including *Hif1a*, present a more open state in **oe**SKIN cells at the transition between oesophagus and skin (along the scATAC-seq pseudotime trajectory; Fig. 5L–N), supporting its potential involvement in the lineage conversion process.

To histologically characterise the HIF1a signature observed within the regenerative cluster C8, tissue wholemounts derived from heterotypic cultures were immunostained against HIF1a (Supplementary Fig. 5J, K). The results showed that, indeed, HIF1a expression was associated with an active regenerative state, revealing a marked HIF1a expression early during tissue re-epithelialisation at D3, independent of whether the oesophagus was grown over their own or ectopic stroma. At D10, however, HIF1a levels dropped in the maturing in vitro oesophagus, while remained persistently high in oesophageal-derived skin (**s**SKIN), including **oe**HFs, suggesting a persistent regenerative state in **oe**SKIN samples.

In light of our scRNA-seq analysis, we conclude that oesophageal cells transitioning towards skin identity remain in an unresolved regenerative state marked by a sustained hypoxic signature, denoted by the overexpression of HIF1a downstream target genes.

## Inhibition of the HIF1a-SOX9 axis lifts the lineage conversion barrier

Next, we set out to investigate the impact of HIF1a on the lineage conversion process. HIF1a downregulation has been shown to be necessary for the adequate resolution of skin wounds[92,93]. Thus, we hypothesised that its persistent activation in transitioning cells may be locking cells in a regenerative state, limiting their ability to respond to ectopic niche signals and preventing the identity switch. To explore this possibility, we treated heterotypic cultures with pharmacological compounds known to inhibit (KC7F2) or activate (DMOG) HIF1a[94,95] (Supplementary Fig. 6A–C). Remarkably, HIF1a inhibition favoured

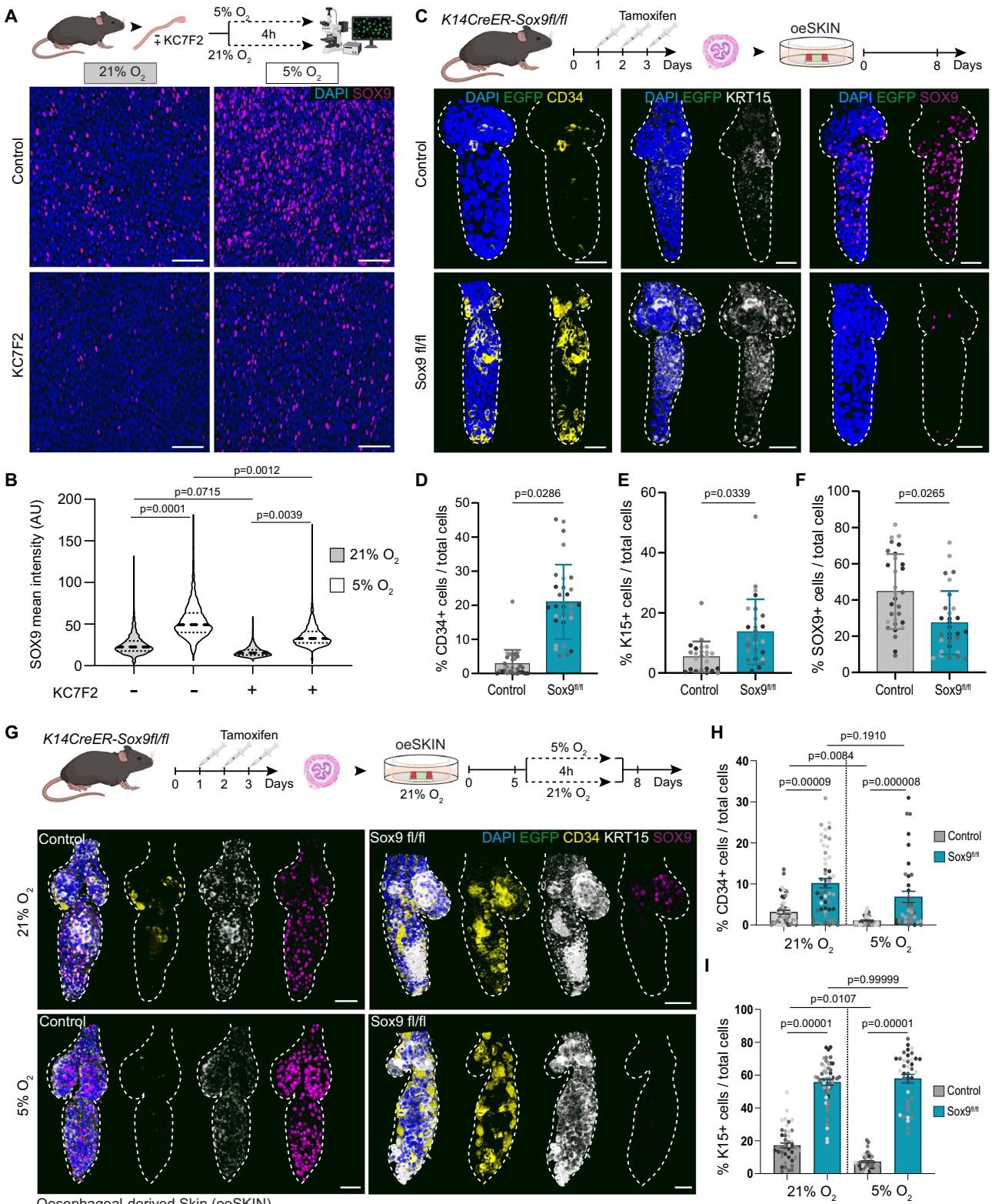

Oesophageal-derived Skin (oeSKIN)

differentiation towards the skin lineage, showing a significant increase in the percentage (%) of CD34+ and KRT15+ cells in the newly formed **oe**HFs (Fig. 5O–Q). Conversely, activation of HIF1a pathway resulted in the opposite effect, leading to a significant reduction in CD34+ and KRT15+ cells per **oe**HF (Fig. 5O–Q). These results revealed that HIF1a inhibition enabled an increase in the number of cells expressing HF markers (CD34 and KRT15) per **oe**HFs. We concluded that the persistent regenerative state limiting the lineage conversion process was modulated, at least in part, by HIF1a pathway. Moreover, inhibition of HIF1a signalling pathway promotes the escape from the regenerative blockage and favours fate re-specification towards skin as instructed by dermal signals.

Further mechanistic insight was obtained when analysing SOX9 expression, a known HF stem cell marker in the skin essential for HF formation and maintenance[13,41–44,96]. Remarkably, oesophageal-derived HFs showed that SOX9 featured the opposite profile to

**Fig. 6 | HIF1a-SOX9 axis poses barriers to oesophageal-to-skin lineage conversion. A** Schematic, C57BL/6 wild-type oesophagi exposed to 21% or 5% oxygen conditions for 4 h, in presence or absence of HIF1a inhibitor KC7F2. Representative basal single-plane views show upregulation of SOX9 under hypoxia, attenuated by KC7F2. **B** Quantification of SOX9 intensity from A (AU, arbitrary units). Dashed line, median; dotted lines, quartiles. 1500–2500 cells from 3 mice. Analysis using one-way ANOVA with multiple comparisons. **C** Schematic, $Krt14^{CreERT}$/ $Sox9^{fl/fl}$ mice were injected with tamoxifen (TAM) to delete Sox9 in epithelial cells. Oesophagi were collected 72 h after the first TAM administration and cultured over EGFP+ dermis for 8 days. 3D confocal side views showing CD34, KRT15 and SOX9 expression in **oe**HF from uninduced (top) and induced (bottom) mice. **D–F** Quantification of CD34+ (**D**), KRT15+ (**E**) and SOX9+ (**F**) cells per **oe**HF (% vs DAPI). ≥ 6400 cells from 30 **oe**HFs, 4 mice. The points show individual measurements (greyscale = biological replicates). Data expressed as mean ± SD. Analysis using two-tailed Mann Whitney test. **G** Schematic, $Krt14^{CreERT}$/ $Sox9^{fl/fl}$ mice were induced with tamoxifen as in (**C**). Heterotypic cultures at day 5 were exposed to 5% or 21% oxygen for 4 h, and returned to normoxia until day 8. 3D confocal side views of uninduced (left) and induced (right) **oe**HF. Normoxia (top); hypoxia (bottom). **H,I** Quantification of CD34+ (**H**) and KRT15+ (**I**) cells per **oe**HF (% vs DAPI+). 10,000–15,000 cells from 40–53 **oe**HF, 4 mice. The points show individual measurements (greyscale = biological replicates). Data expressed as mean ± SEM. Analysis using Kruskal-Wallis with Dunn's multiple comparisons. **Scale bars**. 50μm (**A**, **E**, **G**). **Fluorescent labels**. Blue, DAPI; Greyscale, KRT15; Yellow, CD34; Magenta, SOX9. **Mouse lines**. Constitutive *H2B-EGFP* mice were used as source of tail skin. C57BL/6 wild-type (**A**, **B**) and $K14^{CreERT}$/ $Sox9^{fl/fl}$ mice (**C–I**) were the source of oesophageal tissue. Dashed lines delimit hair follicles. Schematics in panels **A**, **C** and **G** created in BioRender. Alcolea, M. (2025) https://BioRender.com/6xd9u2l See also Supplementary Fig. 6. Source data are provided as a Source Data file.

that observed for other HF markers, such as CD34. SOX9 was instead positively regulated by HIF1a; being downregulated upon HIF1a inhibition and increased in response to HIF1a activation. (Fig. 5O, R). Previous work already established a connection between SOX9 and HIF1a, demonstrating the direct regulation of SOX9 expression by HIF1a in bone morphogenesis[97,98]. In line with this, SOX9 expression was also found to positively correlate with HIF1a expression in **oe**HFs (Supplementary Fig. 6D). These results suggest that SOX9 may be acting in a context/tissue dependent manner, having a different role to that described during HF development and maintenance[41,42,45,99]. To test whether SOX9 expression is HIF1a-dependent in an oesophageal context, we first performed whole organ ex vivo treatments[36], exposing the oesophagi to hypoxic conditions and/or KC7F2 (HIF1a inhibitor). The results showed that SOX9 expression was indeed induced by hypoxic conditions and that this induction could be prevented by inhibiting HIF1a (Fig. 6A,B). The potential co-regulation of these two transcription factors was supported by the increased HIF1a expression observed upon genetic inactivation of *Sox9* expression in epithelial **oe**HF cells ($Krt14^{CreERT}$/$Sox9^{fl/fl}$; Supplementary Fig. 6E, F). Taken together, these intriguing results revealed that, in our model, SOX9 is regulated by HIF1a, playing a role different to that previously established in skin HFs.

Previous work had identified SOX9 as a transcription factor associated with regenerative cell fate plasticity, becoming ectopically expressed by the interfollicular epidermis in response to skin injury[13,96]. Hence, to better understand the context in which SOX9 becomes expressed in the oesophagus, we analysed SOX9 expression in oesophageal wholemounts under different physiological and non-physiological conditions. These included adult homeostasis, early postnatal development, oesophageal excisional wound healing, as well as in two different 3D organ culture systems (Epithelioids[100]) and **oe**OE re-epithelialisation assays[7]. The results revealed that, although SOX9 is typically absent in the adult homeostatic epithelium, its expression is notably upregulated in all non-homeostatic conditions tested (Supplementary Fig. 6G, H), denoting a link between an active regenerative/developmental response and SOX9 expression.

Next, we asked whether SOX9 may be acting as an effector of HIF1a in the lineage conversion process. To this end, we explored whether the ablation of *Sox9* expression reproduced the HIF1a effect on cell identity. For this, we made use of the $Krt14^{CreERT}$/$Sox9^{fl/fl}$ conditional mouse model, whose in vivo induction resulted in a significant reduction of SOX9 expression specifically in epithelial cells (Supplementary Fig. 6E). Heterotypic cultures derived from induced $Krt14^{CreERT}$/$Sox9^{fl/fl}$ oesophagi showed an increased expression of HF markers (CD34 and KRT15) in the **oe**HF (Fig. 6C–F). Interestingly, this boost in lineage conversion happened despite the increase in HIF1a levels seen upon SOX9 inhibition (Supplementary Fig. 6F), further supporting the notion that SOX9 is contributing to the lineage conversion barrier downstream of HIF1a.

Finally, to further demonstrate the mechanistic role of the HIF1a-SOX9 axis in the lineage conversion process, we examined the interplay between hypoxic signals and SOX9 expression in **oe**HFs. $Krt14^{CreERT}$/$Sox9^{fl/fl}$ derived heterotypic cultures were exposed to hypoxic conditions (5% $O_2$), in order to simultaneously inactivate SOX9 and stimulate HIF1a (Fig. 6G). We anticipated that if HIF1a and SOX9 expression were acting as independent barriers blocking the lineage conversion process, the hypoxic treatment would rescue the $Krt14^{CreERT}$/$Sox9^{fl/fl}$ phenotype, bringing HF markers back to control levels. Marker quantification showed that this was not the case, the hypoxic treatment did not restore HF markers in **oe**HFs (Fig. 6G–I), indicating that, in this context, HIF1a is acting via SOX9 to modulate the lineage conversion process. Again, these observations appeared to conflict with findings in the skin, where SOX9 is known to play a key role for the specification and maintenance of HF stem cells[41–45]. Hence, to specifically test whether the role of SOX9 was context dependent, we repeated the experiment in $Krt14^{CreERT}$/$Sox9^{fl/fl}$ derived **s**SKIN (Supplementary Fig. 6J–L). Indeed, SOX9 inactivation in in vitro skin led to the opposite effect, reducing the number of CD34+ and KRT15+ cells in the **s**SKIN (Supplementary Fig. 6J–L). HIF1a stimulation via hypoxia had no significant effect on **s**HF marker expression (Supplementary Fig. 6J–L). These results evidence the context dependent role of SOX9, driving different biological outputs in various biological processes. This is in line with its widely known involvement in numerous developmental, regenerative and pathological processes where the common denominator is tissue remodelling[13,101–103]. Overall, these data demonstrate that HIF1a and its downstream effector SOX9 establish a regenerative barrier that prevents cell plasticity during tissue repair in oesophageal cells.

### HIF1a-SOX9 axis prevents cell fate re-specification in vivo
Next, we investigated the lineage conversion process in vivo and its underlying regulation. To this end, we implemented two different grafting approaches. First, we developed a heterotypic transplantation technique that recreates our in vitro strategy. Skin punches were excised from the back of nude mice, and replaced by peeled tail dermis (expressing EGFP) overlaid with a strip of OE (expressing tdTomato; Supplementary Fig. 7A), both tissues collected from adult mice. One to two weeks following the procedure, OE cells generated proliferating HF-like structures that expressed the HF marker KRT24, and contained a central fibre resembling the HF shaft (Supplementary Fig. 7B–F). We conclude that adult OE cells are able to change their identity and form **oe**HF structures also in an in vivo context.

This heterotypic grafting approach served its purpose to characterise in vivo cell fate switching. However, grafts were ultimately lost, preventing longer-term assessment of OE-derived hair formation. To address this, we made use of a well-established in vivo HF reconstitution assay[104–106]. Dissociated cells from adult OE cells (tdTomato+) and neonate skin (EGFP+) were combined and grafted onto the back skin of nude mice using a silicon chamber (Fig. 7A).

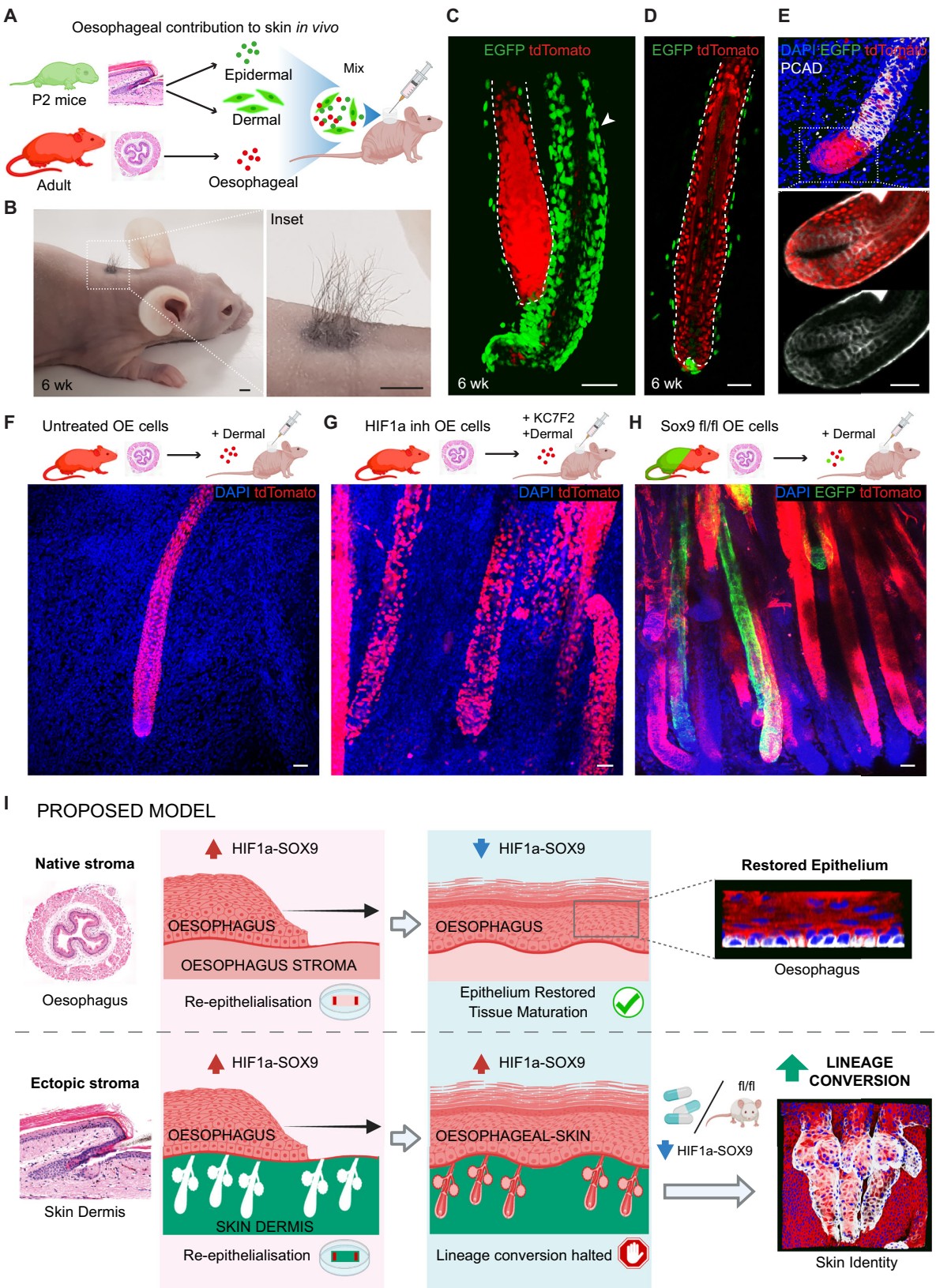

**I** PROPOSED MODEL

This resulted in the emergence of skin-derived HFs (EGFP; Supplementary Fig. 7G), as well as chimeric oesophageal/epidermal HFs (tdTomato/EGFP; Supplementary Fig. 7H) that generated functional hair shaft-like structures 6 weeks post-grafting (Fig. 7B). To improve the oesophageal HF formation efficiency, we increased the ratio of oesophageal cells by 4 times. This led to the emergence of HFs fully derived from oesophageal cells (tdTomato; Fig. 7C, D and Supplementary Fig. 7I). Oesophageal-derived HFs formed adjacent to EGFP skin-derived HFs (arrowhead, Fig. 7C and Supplementary Fig. 7I), and were surrounded by supportive skin-derived EFGP mesenchymal cells (Fig. 7C, D). Additionally, the HF marker P-cadherin was expressed in HF-like structures showing its typical in vivo expression

**Fig. 7 | HIF1a-SOX9 axis regulate oesophageal-to-skin lineage conversion in vivo. A** Schematic showing heterotypic cell grafting. Oesophageal epithelial cells ($5 \times 10^5$) from adult tdTomato mice were isolated and mixed with epidermal ($2.5 \times 10^5$) and dermal ($2.5 \times 10^6$) single-cell suspensions from postnatal day 2 (P2) *H2B-EGFP* mice. Cells were injected into a silicon chamber onto the back of a nude mouse, and allowed to reconstitute skin and grow hair for 6 weeks.
**B** Representative image showing hair outgrowth in the back skin of a nude mouse 6 weeks post-grafting (as in **A**). **C**, **D** 3D rendered side views of a full thickness skin wholemount 6 weeks post-grafting (from **B**), showing oesophageal-derived skin hair follicles. EGFP shows cells from neonatal skin origin, including newly formed skin HFs (see arrowhead in **C**) and underlying neonatal dermal cells (EGFP).
**E** Representative 3D rendered side views of confocal z-stacks, and single plane insets showing expression of HF marker PCAD (P-Cadherin) in the innermost hair matrix and outer root sheath in in vivo oesophageal-derived hair follicles (anagen).
**F–H** Heterotypic cell grafting experiments were performed as in (**A**). Donor oesophageal cells were untreated controls (**F**), KC7F2 treated to inhibit HIF1a (**G**), or derived from induced *Krt14^CreERT^/Sox9^fl/fl^* donor mice to conditionally delete *Sox9* expression (**H**). Representative 3D rendered confocal images show an increase of oesophageal-derived hair follicles generated in vivo upon HIF1a inhibition or Sox9 downregulation. **I** Suggested model. Oesophageal epithelium growing over the foreign stroma of the skin (the dermis) remains partially locked in a HIF1a-SOX9 driven regenerative state that restricts the ability of cells to change their identity towards skin. Lifting the regenerative barrier, by inhibiting the HIF1a-SOX9 axis, enhances the response of cells to niche signals instructing an alternative fate, and favours the lineage conversion process. Dashed lines indicate basement membrane. **Fluorescent labels.** Blue, DAPI; Green, EGFP; red, tdTomato; Greyscale, PCAD (in **E**). **Scale bars.** 4 mm (**B**), and 50μm (**C–H**). **Mouse lines.** Constitutive *H2B-EGFP* were used as source of skin. *nTnG* (**C–G**) and *Krt14^CreERT^/Sox9^fl/fl^*, and *mTmG* (**H**) for oesophageal tissue. Panels **A**, **F**, **G**, **H** and **I** created in BioRender. Alcolea, M. (2025) https://BioRender.com/0rewqdm and https://BioRender.com/m7pdvws See also Supplementary Fig. 7.

pattern, being primarily localised in the outer root sheath of the hair follicle (Fig. 7E)[107–109]. Overall, these results are indicative of a switch from oesophageal-to-skin identity.

Consistent with the limited lineage conversion observed in our in vitro system, the OE contribution to HF formation in vivo was highly sporadic. We identified 12 HFs events out of $14 \times 10^6$ OE cells in a total of 34 grafts. Next, we tested whether targeting the HIF1a-SOX9 axis would improve the efficiency of the lineage conversion in vivo. HF reconstitution assays where oesophageal cells were either exposed to KC7F2 (to inhibit HIF1a) or undergone conditional SOX9 inactivation revealed a significant increase in the HF forming efficiency, showing multiple oesophageal-derived HF structures per graft (Fig. 7F–H). KC7F2 treatment produced ~5.4 times more **oe**HFs than control conditions (14 HFs events out of $3 \times 10^6$ OE cells in a total of 6 grafts), while SOX9 inactivation led to ~9.2 times more **oe**HF than control conditions (55 HFs events out of $7 \times 10^6$ OE cells in a total of 14 grafts; Supplementary Fig. 7J).

Finally, 3D heterotypic cultures derived from tracheal epithelium (tdTomato) and denuded skin dermis (EGFP) revealed that the HIF1a-SOX9 axis operates as a fate plasticity barrier beyond the oesophageal epithelium. The expression of HF markers, CD34 and KRT15, was significantly increased in tracheal-derived skin upon genetic inactivation of Sox9 expression (*Krt14^CreERT^/Sox9^fl/fl^*; Supplementary Fig. 7K–M). Overall, our data shows that the oesophageal-to-skin lineage conversion process involves a series of events. These include i) the re-epithelialisation of the denuded skin dermis by oesophageal cells. Here oesophageal cells repopulate the empty epidermal stroma forming HF-like structures as they fill the empty HF sockets. ii) This process exposes oesophageal cells to the ectopic HF niche. iii) The instructive niche signals enable the lineage conversion process in competent oesophageal cells in a HIF1a-SOX9 dependent manner. We go on to demonstrate that blocking HIF1a or its downstream target SOX9 lifts the regenerative blockage and favours cell fate plasticity by promoting the transition from oesophageal to skin lineage both in vivo and in vitro (**Proposed model** Fig. 7I), and that this may represent a more universal mechanism operating beyond oesophageal epithelium.

## Discussion

It has now become widely accepted that cell fate decisions are not as predetermined as originally thought. In response to injury or environmental perturbations, adult epithelial cells present a remarkable fate plasticity allowing them to respond to changes in tissue needs[9–11]. Yet, if dysregulated, this adaptability can promote epithelial disease and cancer[9,13,68,110]. Particularly, in the context of the oesophagus, columnar metaplasia has been associated with the development of oesophageal cancer[62,111–113]. Hence, understanding the largely unclear mechanisms balancing cell fate plasticity is critical to unveil its

regenerative potential while preventing inherent tumorigenic risks[8]. Here, we develop an ex vivo regenerative assay, amenable to 3D imaging reconstruction, to investigate adult epithelial cell fate plasticity in a spatio-temporal manner. Our data demonstrates that OE cells exposed to the ectopic signals of the skin dermis are instructed to switch their identity towards epidermal fate. The inefficiency of this process, however, reveals the existence of inherent barriers to lineage conversion; with most transitioning cells being halted in an unresolved regenerative state. Mechanistically, we show that the HIF1a-SOX9 axis operates as a cell fate plasticity barrier, beyond the oesophageal epithelium, and that lifting this regenerative lock releases the fate barrier, enabling cells to respond to signals instructing alternative fate choices (**Proposed model** Fig. 7I).

Under normal homeostatic conditions, progenitor and stem cells are known to differentiate into a limited range of fates, giving rise exclusively to the cell types needed to ensure adequate tissue function and integrity[2,114]. Traditionally, different models of maintenance across tissues converged on the notion that cell fate is unidirectional, with commitment towards differentiation being irreversible[115,116]. And despite historical evidence arguing in favour of a less rigid cell fate programme[27,30,31,117,118], it has only recently become accepted that adult stem cells have the ability to alter their identity in response to extreme conditions, endowing them with the ability to dedifferentiate or transdifferentiate[10,11].

Over the years, efforts to understand the rules governing cell fate plasticity have unveiled the key role of the microenvironment in this process[10,11]. Early tissue grafting studies in chicks uncovered that epidermal cells, regardless of their region of origin, are able to form feathers or scales if instructed to do so by dermis bearing those structures[28,31]. More recently, multiple epithelial tissues have been shown to bear the potential to switch towards HF identity when exposed to the signals of the developing skin stroma[25,29]. In line with these, our results demonstrate that the signals informing the oesophageal-to-skin identity switch originate from the skin dermis. Interestingly, we show that lineage conversion events are not restricted to developing tissues, revealing that adult tissues retain both the permissibility and instructive capacity to execute changes in cell identity.

Recent research efforts have demonstrated that cell fate rewiring often involves epigenetic remodelling[10,13]. However, other than the fact that tissue damage favours plasticity, the specific regulatory processes that dictate the ability of cells to respond to signals instructing alternative fates have remained obscure. Our findings reveal that, during regeneration, epithelial cells exposed to ectopic niche signals resist the natural progression towards tissue maturation, remaining locked in a HIF1a-SOX9-dependent regenerative state that restricts their ability to change their identity towards the skin lineage. Once this regenerative blockage is resolved, re-specification

is favoured and the number of lineage conversion events builds-up. Remarkably, SOX9 function in this setting contrasts with its well-established role in HF stem cell specification and maintenance[41–44], acting downstream of HIF1a to prevent the oesophageal-to-HF fate switch. These observations reveal the context-dependent role of this cell fate master regulator[13,96,119].

As to what keeps **oe**HF cells in an unresolved regenerative state, previous work has shown that HF stem cells are maintained by a hypoxic niche[120]. This raises the possibility that this environment may be playing a role in perpetuating the hypoxic regenerative signature that marks transitioning cells.

Interestingly, studies in the skin have shown that, although HIF-1a activation is essential for initiating tissue repair, its sustained activity can impair wound healing; contributing to persistent inflammation, fibrosis, and impaired wound resolution, as seen in chronic wounds[93,121]. Hence, the HIF1a-SOX9-dependent regenerative blockage points to a remarkably simple self-regulatory process where, in response to injury, plasticity would initially be kept at bay by the same hypoxic signals that promote tissue repair in the first place. This mechanism would ensure that the fate outcome of individual cells contributing to tissue repair is not allocated until later stages of the injury response, when hypoxic barriers are lifted and the final instructive niche is established.

In an injury, often tissue compartments are affected, and cells are known to migrate out of their niche repopulating adjacent structures[9,10,12,13,20,23,63,122–124]. Therefore, it is tempting to speculate that the existence of barriers to cell conversion during tissue repair may represent a temporal switch preventing cells from prematurely changing their identity in response to transient regenerative cues. This evokes the cancer metastasis process, where epithelial cells travelling to distant organs undergo epithelial-to-mesenchymal transition, blocking their epithelial programme until the metastatic site is reached[125,126]. Intriguingly, plasticity is a well-established behaviour operating in chronic epithelial diseases, and cancer[68,127,128], suggesting that cells may be able to overcome this regenerative blockage under certain pathological conditions.

This study offers new insights into the mechanisms that enable adult epithelial cells to become competent to respond to signals instructing alternative fate choices upon tissue perturbations. Future work will be required to unveil the relevance of the HIF1a-SOX9 blockage, shown to limit cell fate re-specification during regeneration, for clinically relevant models such as wound healing and cancer, where plasticity is known to operate[8,13,15,68,129].

This study opens exciting avenues to decipher the inner workings dictating cell fate plasticity, providing a benchmark to identify the principles underlying the full regenerative capacity of epithelial cells. However, given the potential risk of abnormal tissue growth and malignancy[8,9,13,68,110], it also underscores the need for caution. Careful consideration must be given when developing strategies to either promote tissue plasticity or disrupt plastic barriers in order to minimise unintended pathological consequences. Much work is still needed to understand how to finely balance the plastic response of our epithelial tissues for therapeutic gain.

## Methods

### Mouse strains and allele induction
All mouse experiments were approved by the local ethical review committees at the University of Cambridge, and conducted according to Home Office project license PPL70/8866 and PP7037913 at the Cambridge Stem Cell Institute, Cambridge University.

To identify the tissue-of-origin in 3D heterotypic cultures the following fluorescent reporter mouse lines were used: *mTmG* (*R26mTmG*; stock #007676, Jackson Laboratory[130]) and *nTnG*

(*R26nTnG*; stock #023537, Jackson Laboratory[131]), which constitutively express *tdTomato* localised in the cell membrane and nuclei, respectively; *H2B-EGFP* (*CAG::H2B-EGFP*; kindly provided by J. Nichols[132]) with constitutive *EGFP* nuclear expression; and *Lgr5-EGFP* (*Lgr5-EGFP-IRES-creERT2*; stock #008875, Jackson Laboratory[133]) with an EGFP cassette targeted to the 3′ untranslated region of the *Lgr5* gene.

*Fucci2a* mouse line (*R26Fucci2a*, kindly provided by Ian J. Jackson[134]), which incorporates constitutive expression of cell cycle reporter proteins (G1 marked by *mCherry-hCdt1*, and S/G2/M marked by *mVenus-hGem*), was used to visualise cycling cells. For lineage tracing analysis, *R26-CreERT2* animals (stock #008463, Jackson laboratory[135]) were crossed onto *R26R-Confetti* (stock #017492, Jackson laboratory[59]) to generate inducible *R26^CreERT/ R26^Confetti* mice. The resulting mice express one of four fluorescent reporter proteins (YFP, GFP, RFP, or CFP) in sporadic cells following tamoxifen-mediated Cre recombination. For *Sox9* conditional knock out experiments, *Sox9^flox/flox* mice (obtained from MRC-Harwell, which distributes these mice on behalf of the European Mouse Mutant Archive https://www.infrafrontier.eu[136]) were crossed onto *K14-CreERT* mice (stock #005107, Jackson laboratory[137]) to generate the *K14^CreERT/ Sox9^flox/flox* enabling genetic inactivation of *Sox9* expression in epithelial cells upon tamoxifen induction. C57BL/6 J mice (strain code, 632) and nude athymic mice (strain code, 490) were purchased from Charles River, UK.

Recombination of *R26^CreERT2/ R26R^Confetti* mice was induced by a single intraperitoneal (i.p.) tamoxifen injection (3 mg per 20 g of body weight in sunflower seed oil). *K14^CreERT/ Sox9^flox/flox* received two subcutaneous and one intraperitoneal tamoxifen injections (5 mg per 20 g of body weight in sunflower seed oil) in 3 consecutive days. The oesophagi of treated animals were collected 72 h after the first TAM administration.

All experiments comprised mixed male and female animals, with no gender specific differences observed. For RNA sequencing experiments, only male animals were used in order to avoid confounding effects due to oestrous cycle. All animal cohorts were adults between 8–25 weeks of age. Mice were bred and maintained under specific-pathogen-free conditions at the Cambridge Stem Cell Institute and Gurdon Institute of the University of Cambridge. All animals were housed between 20–24 °C, 45–65% humidity, and a 12-h light-dark cycle.

### 3D Oesophageal-Dermal Heterotypic and control cultures
In order to investigate the initial mechanisms modulating adult cell fate plasticity in response to regenerative signals, we adapted our ex vivo organotypic wound healing assay[7] to co-culture adult epithelia and stroma of different origins, in this case, mouse tail skin and oesophagus. To this end, tissues were freshly dissected out and processed to separate epithelial and stromal compartments. Epithelium and stroma were obtained from the middle half of the oesophagus to avoid confounding effects due to cellular heterogeneity along the longitudinal axis of the oesophagus[7]. Oesophagi were opened longitudinally, the muscle layer removed, and the tissue flattened under a dissecting microscope. Skin and oesophageal stroma – referred to as dermis and submucosa, respectively – were carefully peeled away from their overlying epithelium with fine forceps after a 3–4 h incubation in 5 mM EDTA at 37 °C. The stroma was cut into 7 x 9 mm pieces for the dermis and 5 x 10 mm pieces for the submucosa and placed onto transparent ThinCert™ inserts (0.4 µm pore size; Greiner Bio-One Ltd; Cat#657641). Immediately afterwards, two strips of oesophageal tissue (5 x 1 mm; lacking the muscle layer) were laid on top of the denuded/peeled stroma. To generate in vitro skin controls (**s**SKIN; **s**HF, Hair Follicle; **s**IFE, Interfollicular Epidermis; **s**- prefix denotes in vitro skin), skin from the ear pinna was dissected, cut in strips (5 x 1 mm) and laid on top of the denuded tail dermis following the same procedure as with the oesophagus. Heterotypic tissue composites were then

allowed to settle in the ThinCert™ for 5 min at 37 °C to ensure attachment, and covered up to tissue level with minimal medium (mFAD; lacking added growth factors) containing one-part DMEM (4.5 g/L D-Glucose, Pyruvate, L-Glutamine), one-part DMEM/F12, supplemented with 5% foetal calf serum, 5 µg/ml insulin, 0.18 mM adenine, 5–10 µg/ml transferrin and 1% Penicillin-Streptomycin. Heterotypic constructs are viable for up to 15 days of culture[7]. Here, cultures were grown for up to 10 days at 37 °C and 5% CO2, replacing the medium on alternate days. During this period of time, the denuded/peeled stroma was re-epithelialised by oesophageal cells. Oesophageal epithelium was grown over its native stroma as a control (oeOE; prefix e- denotes oesophageal re-epithelialisation in vitro). OE grown on top of denuded dermis was considered oesophageal-derived skin (oeSKIN), and associated compartments (oeHF, Hair Follicle; oeIFE, Interfollicular Epidermis). Unless otherwise stated, oesophageal tissues were tdTomato + (mTmG or nTnG mouse lines), while tissues of skin origin were EGFP+ (from H2B-EGFP line).

At the end-point, organotypic cultures were either embedded in optimal cutting temperature compound (OCT; Fisher Scientific Ltd; 12678646) for cryosectioning, or the epithelium was whole-mounted by peeling away from the stroma following a 2-h incubation in 5mM EDTA at 37 °C.

For the limited re-epithelialisation assay (Supplementary Fig. 4H, I), heterotypic cultures were performed using peeled OE, lacking oesophageal stroma. For this, the oesophagus epithelium was peeled from the underlying submucosa following a 3-h incubation in 5 mM EDTA at 37 °C. The freshly peeled epithelium was cut into 5 × 8 mm pieces and laid on the denuded dermis (as above). For controls, 5 x 8 mm pieces of unpeeled epithelium were used instead.

For explant cultures pieces of oesophageal tissue (5 × 5 mm) were grown directly onto ThinCert™ inserts under mFAD media conditions (as above), replacing the medium on alternate days. 10 days post-culture tissues were processed as epithelial wholemounts (see below) for immunofluorescence analysis.

### Oesophageal-tongue and oesophageal-bladder heterotypic constructs

To investigate whether the exposure of the oesophageal epithelium to an ectopic stroma other than the dermis could trigger cell fate rewiring, we followed the procedure described above for oesophageal-dermis heterotypic constructs with some modifications as follows. Bladders and tongues were dissected from EGFP+ mice. The muscle layer was removed, and bladder and tongue tissues were incubated in EDTA 5 mM for 2 h and EDTA 50 mM for 90 min, respectively. The epithelial compartments were then peeled off and discarded. The bladder or tongue stroma was placed in inserts and oesophageal strips laid on top.

### Tissue wholemount preparation and immunofluorescence

Whole mounts from the mouse tail skin and the middle third of the oesophagus were obtained as follows. Tissues were dissected as indicated above and cut into pieces of approximately 5 x 8 mm, followed by a 3-4 h incubation in 5 mM EDTA at 37 °C. The epithelium was then gently peeled away from the underlying stroma and fixed in 4% paraformaldehyde (PFA) in PBS for 30 min. For immunostaining, wholemounts were incubated in blocking buffer (0.5% Bovine Serum Albumin, 0.25% Fish skin gelatin, 0.5% Triton X-100 in PBS and 10% donkey serum) for 1 h at 37 °C prior to staining with primary antibodies (diluted in blocking buffer) overnight at 4 °C (see below). This was followed by four 30-min washes with 0.2% Tween-20 in PBS. Samples were subsequently incubated with secondary antibodies (diluted in blocking buffer) for 3 h at RT. Samples were washed as above, cell nuclei counterstained with 1 µg/ml DAPI in PBS, and samples mounted in 1.52 RapiClear mounting media.

For staining of tissue cryosections, fixed samples were embedded in OCT, cut at 150µm and placed onto poly-L-lysine coated glass slides. Cryosections were stained and mounted following the whole-mount protocol (as above).

Primary antibodies used included: Caspase-3 active (Rabbit, 1:200, Abcam #ab2302), CD34 (Rat, 1:100, BD Bioscience #553731), CD49f (integrin α6, Rat, 1:200, BioLegend #313610), HIF1a (Rabbit, 1:100, Novus Biological #NB100-479), KRT4 (Mouse, 1:2000, Vector Laboratories #VP-C399), KRT10 (Rabbit, 1:200, Abcam #ab76318), KRT14 (Rabbit, 1:1000, BioLegend #905301), KRT15 (Mouse, 1:1000, Abcam #ab80522), KRT17 (Rabbit, 1:250, Cell Signalling #4543S), KRT24 (Rabbit, 1:500, Atlas #HPA022978), LHX2 (Mouse, 1:200, Santa Cruz #SC-517243), MMP9 (Goat, 1:50, R&D Systems #AF909-SP), PDGFRA (Goat, 1:200, R&D Systems #AF1062), SCA-1 (Rat, 1:100, Invitrogen #14-5981-82), SCD1 (Rat, 1:100, R&D Systems #MAB4404), SOX2 (Rat, 1:200, eBioscience #14-9811-82) and SOX9 (Rabbit, 1:1000, Millipore #AB5535).

Secondary antibodies used included: Goat IgG 647 (Millipore #AP180SA6), Goat IgG 750 (Abcam #ab175745), Mouse IgG 647 (Invitrogen #A-31571), Mouse IgG 750 (Abcam #ab175738), Rabbit IgG 488 (Invitrogen #A-21206), Rabbit IgG 555 (Invitrogen #A-31572), Rabbit IgG 647 (Invitrogen #A-31573), Rabbit IgG 750 (Abcam #ab175728), Rabbit IgG (Abcam #ab171870), Rat IgG 647 (Abcam #ab150155), and Rat IgG 750 (Abcam #ab175750). All secondary antibodies were diluted 1:500.

### Confocal imaging and analysis

Images were acquired usingeither inverted Leica SP5, or SP8 confocal microscopes (Leica Microsystems) with standard laser configuration or a Stellaris 8 FALCON FLIM microscope with a white light laser. Typical settings for z-stack image acquisition included optimal pinhole, line average 3, optimal step size, bi-directional scan with speed of 400 Hz and a resolution of 1024 × 1024 pixels. Images were acquired using a 10x, 20x or 40x objective. For the latter, a digital zoom of 3x for OE and IFE epithelium, and 1x for HFs was typically used. Confocal images were reconstructed and analysed using Volocity 5.3.3 software (PerkinElmer) and arivis 3.5.0 (Vision4D). Fluorescence intensity analysis of individual cells was performed in a minimum of 18 HF units (from a minimum of n = 3 animals). To calculate HF markers expression rate, a minimum of 150 oeHF units (from n = 3 animals) were examined on live scanning mode. Figures show representative images.

### Histology

Haematoxylin and Eosin (H&E) and Fontana-Masson (Sigma; Cat#HT200) staining were performed in 4.5µm paraffin-embedded sections by the Histology Core Service at Cambridge Stem Cell Institute. Fontana-Masson silver staining was used to visualise melanin in skin tissues in vivo (skin and denuded dermis) and in vitro (oeSKIN), based on melanin-mediated reduction of silver nitrate to metallic silver.

Oil Red O staining was performed in oesophageal-derived skin (oeSKIN) peeled wholemounts to visualise lipid accumulation. Briefly, epithelial wholemounts were fixed with 4% paraformaldehyde (PFA) in PBS for 30 min. Tissues were incubated in 60% isopropanol for 5 min at RT and immediately after, incubated in a filtered solution of 1% Oil Red O for 30 min at RT. Tissues were rinsed with 60% isopropanol, washed twice in PBS and mounted on slides. Stained sections and wholemounts were imaged using Apotome Imaging System (Zeiss).

### Dermal decellularisation assay

The decellularisation method was adapted from previously published protocols[138,139]. Briefly, freshly denuded dermis (as above) were incubated for 4 h in hypertonic solution (0.1 M NaOH, 1 M NaCl and 25 mM EDTA in DI water) to lyse cells, followed by an overnight incubation in 100 mM EDTA at 4 °C. DNA was enzymatically digested for 6 h in a solution containing 2U/ml Benzonase, 47 mM Tris, 1.4 mM MgCl$_2$,

19 mM NaCl in deionised water at pH 8-9.5. Dermises were washed overnight in 100 mM EDTA at 4 °C, followed by a 2-h incubation in a solution containing 8 mM CHAPS, 1 M NaCl and 25 mM EDTA in PBS at pH 8, and a final wash in a 100 mM EDTA solution overnight. The decellularised dermis was then thoroughly washed with PBS for 30 min and sterilised by an overnight incubation in 1 % P/S PBS at 4 °C. Unless otherwise specified, incubation steps were done in an orbital shaker at room temperature (RT). Between different solutions, dermis were equilibrated in PBS (3 washes of 10 min on each occasion).

## Oesophageal-derived HF formation efficiency

To analyse the ability of oesophageal cells to colonise the HF niche and form structures resembling HFs, 7 x 9 mm tail skin pieces from *Lgr5-EGFP* reporter mice were incubated for 30 min in 4 µg/ml DAPI in PBS, tile-scanned on a confocal microscope (Leica SP5), and the number of HFs initially present in the tissue quantified. Immediately after imaging, the quantified tail pieces were used to perform heterotypic cultures (as above). In short, skin pieces were washed in PBS and incubated in 5 mM EDTA for 4 h at 37 °C. The dermis was dissected away from the epidermis and oesophageal tissue from *mTmG* mice laid on top of the former. Heterotypic cultures were grown for 10 days, after which period the resulting tissue was fixed in full thickness in 4% PFA for 1 h. Wholemounts were then washed three times in PBS and mounted in 1.52 RapiClear mounting media for imaging. The number of oesophageal-derived HF structures (mTomato + ) were quantified using xyz rendered views of tiled confocal z-stack images. **oe**HF formation efficiency in vitro was calculated as the percentage of **oe**HF formed in vitro compared to the number of original HFs in the in vivo skin piece. A total of 3 independent samples were quantified (n = 3 animals).

## Single-cell RNA isolation and library preparation

Heterotypic 3D cultures were prepared as indicated above using oesophagus from *nTnG* mice and denuded tail skin dermis from *H2B-EGFP* mice. As controls samples, oesophagus from *nTnG* was grown over peeled oesophageal stroma from *H2B-EGFP* mice. The resulting epithelia (oesophageal control, **oe**OE; and oesophageal-derived skin, **oe**SKIN) were harvested 3 and 10 days post-culture (D3 and D10, respectively), at which point the epithelia were carefully peeled from their underlying stroma following a 50 mM EDTA incubation for 15 min at 37 °C. A single-cell suspension was obtained by rinsing the peeled epithelia with PBS, and incubating it with 0.5 mg/ml Dispase (Sigma) for 5 min. EDTA was then added to the samples at a final concentration of 5 mM, and the suspension was diluted 1/5 by adding FACS Buffer (FB; 2% heat-inactivated Foetal bovine serum (Life Technologies; 26140079), 25 mM HEPES (Life Technologies; 15630056)) in order to inhibit Dispase activity. The cell suspension was filtered through a 30µm cell strainer, and centrifuged at 300 g for 10 min at 4 °C. Cells were finally resuspended in FB containing 1U/µl RNAse Inhibitor. Single viable epithelial cells were sorted (using a BD FACSAria™ III cell sorter) by tissue of origin: tdTomato (oesophageal origin) and EGFP (in vitro skin). Sorted tdTomato cells from Batch 1 were spiked in with EGFP in vitro skin cells derived from skin organotypic cultures grown under the same conditions as the **oe**SKIN and **oe**OE as an internal control to define epidermal skin identity. Batch 2 was kept free of EGFP in vitro skin cells to avoid confounding effects (see details in Source Data). scRNA-seq libraries were generated and sequenced using 10X Genomics kits (Single Cell 3′ v3) at the CRUK-CI Genomics Core Facility of the CRUK Cambridge Institute. Libraries were processed in two different batches/dates. Batch 1 included 3 biological replicates for **oe**OE at D3, **oe**SKIN at D3, **oe**OE at D10, and **oe**SKIN at D10, rendering 12 libraries; Batch 2 included 2 biological replicates for **oe**SKIN at D10, rendering 2 libraries. Each biological replicate consisted of pooled cultures from 4-5 wells with a total of 9,000 sorted cells per sample. The libraries

were multiplexed and sequenced on 4 lanes Illumina NovaSeq6000 S2 flow cells.

## Single-cell RNA-seq analysis

Pre-processing quality checks were performed on the R2 reads using FastQC (https://www.bioinformatics.babraham.ac.uk/projects/fastqc). The data was then processed using the CellRanger software (v3.1.0) with standard alignment (genome assembly GRCm38.97 of *M musculus*) and default filtering parameters. Upper and lower bounds on the distributions of counts, features, mitochondrial and ribosomal RNA were used to remove outlier cells. Cells were included in the analysis if their sequencing depth was over 8750, the number of expressed genes was between 2000 and 8000, and the percentage of mitochondrial and ribosomal DNA was lower than 15% and 45%, respectively. Mitochondrial and ribosomal genes were subsequently removed from the matrix. As a result, a total of 107,445 cells, discarding doublets, were selected for further analysis. The raw expression matrix was normalised using sctransform[140]; and the downstream analysis was performed using the Seurat R package (v3.2.2)[141] (see Source Data). To filter out non-epithelial cells, cells expressing fibroblast or immune marker genes including *Col1a2*, *Pdgfra* or *Ptprc* were excluded from the analysis (these were cells with a normalised read count value higher than the first local minimum of the density distribution in all cells for each of the markers).

Dimensionality reductions (PCA and UMAP), as well as clustering, were performed using Seurat, with the number of unsupervised clusters being the optimal selected by the software, with default parameters. The biological replicates for each time point overlapped well with each other, confirming negligible batch effects between samples and conditions; therefore, no batch effect correction was necessary. Based on the characteristics of the dataset (stability of partitioning assessed in an incremental manner), we focused on the 1000 most abundant genes to identify the main transcriptional features for each cluster. After the gene expression analysis (identified by testing for differential gene expression using the Wilcoxon Rank Sum test on the Bonferroni-adjusted p-value, with a natural log-fold change threshold of 0.25 and a p-value threshold of 0.01), we defined 19 relevant clusters in the present dataset. Clustering stability was evaluated using a PAC analysis[142]. This resulted in a total of 75,519 cells used in the subsequent analysis.

The EGFP in vitro skin reference cells were initially identified in the UMAP space and localised to C18 by the expression of genes known to specifically discriminate EGFP in vitro skin versus tdTomato oesophageal cells. EGFP epidermal cells showed an enriched expression of the reporter gene *Egfp* and *Gt(ROSA)26Sor* (blocked in Rosa26nTnG oesophageal cells due to the targeting of the nTnG construct to the *Rosa26* locus[131]; Fig. 4D and Supplementary Fig. 3J), as well as the negligible expression of the oesophageal basal marker *Sox2*[65,66] (Fig. 4E, F). The skin origin of C18 was further confirmed by the implementation of a semi-supervised gene classifier method, which infers identity based on the transcriptomic signatures of the detected tdTomato+ and EGFP+ cells, allowing us to set C18 as a cluster of reference to discriminate conversion events. In short, to avoid high similarity regions between *EGFP* and *tdTomato* nucleotide sequences, we selected, based on a ClustalOmega alignment[143], 3 and 5 specific sub-transcripts for each (see Source Data), with less than 20% identity between the two markers. Using STAR aligner (v2.5.2a)[144], with relaxed mapped length parameters, we aligned the reads across all samples to the selected sub-transcripts; the expression levels on the sub-transcripts were defined as the algebraic sum of incident UMIs. Next, we selected ~70 cells with the highest, independent expression of red or green markers, respectively; these two sets were used to train a classifier, part of a semi-supervised model that extends the transcriptomic signatures of the EGFP and tdTomato cells to all unlabelled cells i.e., cells with no or little quantification of the fluorescent markers.

The discriminative genes were selected using a random forest model (the top 50 genes with the highest variation of gini index) and standard statistical tests i.e., t-test, Kolmogorov-Smirnov and Mann–Whitney U; genes with adjusted p-value (on a Benjamini-Hochberg multiple testing correction) lower than 0.05 were selected as discriminative. The final set of markers, used subsequently for inferring labels, was obtained by intersecting the two approaches. For each unlabelled cell, we predicted the label by correlating the selected markers to true tdTomato+ and EGFP+ cells used for the original classifier. Two different correlation-based distance approaches (Pearson and Spearman) yielded comparable results. The two distributions of distances were compared using a minimum IQR approach i.e., if the distance between the tdTomato and EGFP distributions on the proximal quartiles was larger than the average standard deviations, then the cell was labelled according to the distribution closer to 0, otherwise the cell was labelled "unknown".

After identifying cluster 18 as in vitro skin EGFP expressing cells (Fig. 4E **and** Supplementary Fig. 3K, L), the remaining clusters were manually annotated based on gene expression of known cell cycle, epithelial basal and differentiated marker genes (Fig. 4G, Supplementary Fig. 3L, M and Source Data). We then focused on differences in gene expression of the basal cells in the three major transcriptional patterns (**D3**, **oe**OE-D10 and **oe**SKIN-D10 as in Fig. 4H, I). To this end, clusters of annotated basal cells were assigned to one branch if >75% of its cells belonged to the correspondent sample origin. Clusters 3 and 8 had an equivalent contribution from the three sample types and therefore cells in these clusters were split in the three branches depending on their origin (Source Data). Next, we identified differentially expressed genes (DEGs) in pair-wise comparisons between branches. The total number of DEGs was 567. We then scaled the expression value for each gene in the [0,100] range using an affine transformation and summarised the scaled expression levels in a clustered heatmap. The genes were clustered using hierarchical clustering with 1 minus Pearson correlation as the distance measure, which allowed us to identify 6 major patterns: Pattern 1, 50 genes; Pattern 2, 66 genes; Pattern 3, 133 genes; Pattern 4, 54 genes; Pattern 5, 32 genes; Pattern 6, 232 genes (Supplementary Fig. 4D and Source Data). We performed gene enrichment analysis of the resulting 6 patterns using g:Profiler[145] (https://biit.cs.ut.ee/gprofiler/gost; R package (v0.1.8)), against the standard GO terms, and the KEGG[146] and Reactome[147] pathway databases (Source Data). The background set comprised all genes expressed in the dataset.

The expression profile of skin-related genes pointed to a region of the UMAP where two clusters of different identity merged. In vitro skin cluster C18 of epidermal origin and C8 of oesophageal origin marked the area where cells undergoing fate conversion resided. First, we investigated the hallmark genes of oesophageal C8. To this end, we performed a differential gene expression analysis between C8 and each of the remaining basal cell clusters. Next, we performed a gene enrichment analysis using those genes that were differentially expressed uniquely in C8 (Source Data). To explore whether the cell population defined by C8 had an equivalent population in vivo, we integrated our data with our in vivo scRNA-seq dataset[36] (ArrayExpress EMTAB-8662) using the Harmony single cell data integration package (v0.1.0) with default parameters[148].

We then focused on identifying cell fate conversion events which appeared within cluster 8 and cluster 18 (**s**SKIN) (Fig. 4N). To dissect the transcriptional evolution of the identity switch events identified within C8 and C18, we re-clustered cells from C18 and C8 in 10-day samples. The new UMAP distribution segregated cells by the tissue, with **oe**SKIN showing a partial overlap (Fig. 5D). To identify genes changing along the cell fate conversion trajectory, we performed a pseudotime analysis, using the monocle3 R package (v0.2.3.0)[149]. Since C18 in vitro skin cells served as a reference for the skin identity switch,

these were, therefore, fixed as the endpoint of the pseudotime trajectory. Pseudotime analysis ordered cells along the fate conversion trajectory; with cells of different origin plotting at opposite ends of the differentiation path (i.e., C18 **s**SKIN and C8 **oe**OE), and **oe**SKIN cells transitioning between the two (Fig. 5F). To further discern the differences between Oesophagus and TC cells, we repeated the analysis focusing on C8. We re-clustered the C8 cells and performed a pseudotime analysis on the resulting UMAP (Supplementary Fig. 5A, B). For consistency, we selected the start-point identified in the C8-C18 combined pseudotime as a starting point for this pseudotime analysis. The ordering of cells closely aligned with that found in the C8-C18 pseudotime, showing a notable separation between oesophageal control cells (C8 **oe**OE) and oesophageal-derived skin cells (**oe**SKIN). TCs remained separated from the rest at the end of the trajectory (Supplementary Fig. 5C). To examine in more detail the changes occurring in the Transitioning cells (TC), we defined three equidistant groups along the pseudotime axis: Oesophagus, TC, and Skin and identified the DEGs (Differentially Expressed Genes) for each pair-wise comparison (1,173 genes; Source Data). To focus on the exclusive features that drive the TC away from their original identity (Oesophagus), we selected the genes that had > 0.2 logFC between the Oesophagus and the TC (357 genes). We then scaled the expression value for each gene in the [0,100] range using an affine transformation and summarised the scaled expression levels in a clustered heatmap using the Pearson correlation for the clustering to reveal patterns of expression. The nine unsupervised clusters were then grouped in four distinctive patterns of expression: Pattern 1, 94 genes; Pattern 2, 26 genes; Pattern 3, 162 genes; Pattern 4, 75 genes (Fig. 5G and Supplementary Fig. 5D; Source Data). We run gene enrichment analysis for each of the observed patterns using g:Profiler[145], against the standard GO terms, the KEGG[146] and Reactome[147] pathway databases, and TRANSFAC[150] as transcription-factor-binding motifs prediction database (Source Data). The background set comprised of all genes expressed in clusters 8 and 18. Terms and genes were manually curated in Supplementary Fig. 5H, in accordance with the biological context of this work. We then displayed relevant DEGs resulting from the analysis along the pseudotime trajectory (Fig. 5H–J and Supplementary Fig. 5E–G); for each gene, auto-scaled expression was plotted using a generalised additive model for smoothing.

## Single-nuclei isolation and library preparation for ATAC-seq
Epithelia (oesophageal control, **oe**OE; oesophageal-derived skin, **oe**SKIN; and skin control, **s**SKIN) were harvested 10 days post-culture, at which point the epithelia were carefully peeled from their underlying stroma following a 50 mM EDTA incubation for 15 min at 37 °C. The peeled epithelia were rinsed with PBS, and incubated with 0.5 mg/ml Dispase (Sigma) for 5 min. EDTA was then added to the samples at a final concentration of 5 mM, and the suspension was diluted 1/5 by adding PBS + 0.04% BSA buffer. A cell suspension was obtained by filtering through a 30um cell strainer, centrifuging at 300 g for 5 min at 4 degrees and resuspending in PBA + 0.04% BSA. After assessing viability and cell number, nuclei were isolated following 10X CG000169 protocol. Libraries were prepared following the manufacturer's instructions for Chromium Next GEM Single Cell ATAC Kit v2, multiplexed, and sequenced in one lane Illumina NovaSeq X Plus 25B flow cell.

## Single-cell ATAC-seq processing and analysis
Single-cell ATAC-seq libraries were processed using cellranger-atac count (v2.2.0)[151], aligning reads to the reference genome (mm10 2024-A) and generating fragment files along with filtered peak-barcode matrices. Downstream analysis was conducted in R (v4.4.3) using the Signac (v1.14.0)[152] and Seurat (v5.3.0) packages. Cells were filtered based on quality control metrics, including the number of fragments in peaks and number of features to remove outliers. Only cells with ≥40%

of fragments in peak regions and a Transcription Start Site (TSS) enrichment score >1 were retained, resulting in a total of 32,499 high-quality cells for downstream analysis.

Chromatin accessibility was normalised using Term Frequency–Inverse Document Frequency (TF-IDF) transformation, followed by Singular Value Decomposition (SVD) on the peak-cell matrix. Dimensionality reduction was performed via Latent Semantic Indexing (LSI), and top LSI components (excluding the first) were used to construct a Shared Nearest Neighbour (SNN) graph. Louvain clustering identified distinct cell populations, and Uniform Manifold Approximation and Projection (UMAP) was used for visualisation in two dimensions.

To visualise aggregate chromatin accessibility at selected loci, coverage plots were generated using the CoveragePlot function in Signac, displaying fragment density and annotated gene structures. Trajectory inference was performed using Cicero (v1.3.9)[153], which was used to construct an ATAC-specific cell_data_set object from the peak-cell matrix. Monocle3 (v1.4.26) functions were then applied to learn the trajectory graph and order cells in pseudotime based on chromatin accessibility dynamics. Pseudotime-dependent changes in accessibility at individual peaks were visualised using the plot_accessibility_in_pseudotime function.

Differentially accessible (DA) chromatin regions were identified across samples using the FindAllMarkers function in Seurat with a logistic regression test. Peaks were considered significant if they exhibited a log2 fold-change greater than 0.5 and an adjusted p-value (p_val_adj) <0.05. To prioritise region-specific differences, the top 100 DA peaks per sample were selected based on the highest average log2 fold-change.

Average chromatin accessibility for the selected peaks was computed per sample using the AverageExpression function in Seurat. Normalised accessibility values were used for heatmap visualisation, preserving absolute differences in chromatin accessibility between peaks and samples. Heatmap colour scaling was based on quantiles (1st, 25th, 50th, 75th, and 99th percentiles) of the accessibility distribution.

## HIF1a modulation treatments
Heterotypic cultures were treated with the HIF1a translation inhibitor KC7F2 (10 µM; SML1043, Merck) or the HIF1a protein stabiliser DMOG (1 mM;71210, Cayman) in mFAD medium (as above) during a period of 8 days. Control cultures were treated with DMSO vehicle alone in mFAD medium. Cultures were grown in a normoxic humidified incubator at 37°C, 5% CO2, and medium was changed on alternate days.

Experiments modulating HIF1a signalling pathway via hypoxia were performed by exposing the oesophageal tissue/cultures to low oxygen conditions for 4 h in a hypoxic humified incubator at 37 °C, 5% $CO_2$, 5% $O_2$.

## Oesophageal biopsy wounding
Oesophageal wounding was performed by controlled micro-endoscopic biopsying of the mouse oesophagus as previously described[7]. Briefly, C57BL/6 J mice undergoing endoscopy were anesthetised using a combination of 100 mg/kg Ketamine (Pfizer Animal Health) and 10 mg/kg Xylazine administered intraperitoneally. Animals without a complete loss of righting reflex after anesthetic induction were topped up with inhaled isoflurane prior to intervention. A 9.5Fr diagnostic miniature endoscope with a 3Fr instrument channel was used in conjunction with an AIDA COM II image capture system for visualisation (Karl Storz GmbH). 3Fr diameter biopsy forceps with double action jaws (Karl Storz GmBH) were used to create one superficial wound in the middle third of the mouse oesophagus of between 0.4 – 0.9 mm diameter. Anesthesia was reversed using Atipamezole (Pfizer Animal Health) given at 1 mg/kg subcutaneously at least 20 min after induction. Animals were euthanised 24 h after wounding.

## Oesophageal-dermal heterotypic in vivo grafting assay
Heterotypic oesophageal-dermal tissue composites were prepared for in vivo grafting using the strategy described for our in vitro assay. Dermis was obtained from *H2B-EGFP* animals and oesophageal tissue from *mTmG* or *nTnG* mice, unless otherwise stated. To reduce tissue dehydration prior to grafting, oesophageal explants were laid on the dermis and kept in a humidified environment until transplantation. Female nude athymic mice were used as recipients. In a typical experiment, recipient mice anesthetised by isoflurane inhalation underwent a single excision on the dorsal shoulder using a sterile 8 mm diameter biopsy punch. Immediately afterwards, a heterotypic graft (8 mm diameter) was fitted and adhered to the surrounding tissue with Vetbond tissue adhesive (3 M). The area was further sutured using steri-strip wound closure strips (3 M) and covered with Tegaderm dressing film (3 M) to protect the tissue implant. Animals were monitored thereafter, and skin tissue was harvested up to 4 weeks post-transplant.

## In vivo hair reconstitution assay
Hair reconstitution studies were adapted from previously described protocols[105]. In short, back skin was obtained from postnatal day 2 (P2) *H2B-EGFP* mice and placed epidermal side up, floating in 10 ml 0.25% trypsin overnight at 4 °C. After washing with PBS, the epithelium was peeled off using fine forceps and thoroughly minced with a scalpel. The epidermal cell suspension was repeatedly pipetted to disaggregate clumps, filtered through a 30µm strainer and span down at 500 g for 8 min at RT. Epidermal cells were resuspended in cFAD medium, containing one-part DMEM (4.5 g/L D-Glucose, Pyruvate, L-Glutamine), one-part DMEM/F12, supplemented with 5% foetal calf serum, 5 µg/ml insulin, 5-10 µg/ml transferrin, 0.5 µg/ml hydrocortisone, 1 nM cholera enterotoxin, 10 ng/ml EGF and 1% Penicillin-Streptomycin. The neonatal dermis was separately minced and incubated in 0.25% collagenase for 25 min at 37 °C. The dermal cell suspension was then filtered through a 100µm strainer and the HF buds were pelleted by centrifugation at 100 g for 5 min at 4 °C. The supernatant was transferred to a new tube, centrifuged at 300 g for 5 min at 4 °C and resuspended in cFAD medium.

Oesophageal epithelial cells were isolated from adult *nTnG* or induced *K14^CreERT/ Sox9^flox/flox* mice. Longitudinally opened oesophagi were cut in 4 pieces and incubated in 0.5 mg/ml Dispase in PBS for 15 min at 37 °C. The epithelium was peeled, minced with a scalpel and transferred to a new tube with fresh Dispase solution. After a 5 min incubation at RT, EDTA was added at a final concentration of 5 mM to inhibit Dispase activity. Cells were mixed by pipetting and filtered through 30µm strainers. OE cell suspensions were centrifuged at 300 g for 5 min at 4 °C, and resuspended in cFAD medium. All cell suspensions were kept on ice until transplantation.

On the first set of cell transplantations, a cell suspension typically containing $5 \times 10^5$ oesophageal epithelial cells, $5 \times 10^6$ neonatal dermal cells and $10^6$ neonatal epidermal cells was prepared and surgically implanted onto a PDMS silicon chamber secured on the dorsal fascia of recipient nude mice[105], following a full-thickness punch biopsy (8 mm diameter). As these experiments only generated oesophageal/epidermal chimeric hair follicles (Supplementary Fig. 7H), we increased four times the oesophageal: epidermal cell ratio. Following assays (Fig. 7C–H) were performed by implanting a cell suspension containing $5 \times 10^5$ oesophageal epithelial cells, $2.5 \times 10^6$ neonatal dermal cells and $2.5 \times 10^5$ neonatal epidermal cells onto a PDMS silicon chamber in nude mice. In the HIF1a modulation group (Fig. 7G), HIF1a inhibitor KC7F2 (10 µM) was added to the cell suspension prior injection into the silicon chamber. Grafted mice were

monitored and tissue harvested for immunofluorescence analysis 6 weeks post-grafting.

## Statistics and reproducibility

All experiments were performed independently at least three times with similar results, unless otherwise stated. The reproducibility of all key findings was confirmed in independent experiments conducted on different days and using independent biological samples.

The number of animals used for each experiment is indicated in the figure legends ("n" in legends refers to number of independent replicates per time point and/or condition). For oesophageal-derived skin characterisation, in particular, similar results were observed across more than 50 independent experiments. No data were excluded.

Representative images shown in the figures were selected from experiments that were repeated multiple times across independent biological replicates. All figures show representative images of a minimum of 3 mice/tissue cultures. Experimental data are expressed as mean values ± SD, unless otherwise indicated.

Differences between groups/conditions were assessed by using two-tailed unpaired t-test, one-way analysis of variance (ANOVA), and Kruskal-Wallis ANOVA test as indicated in figure legends. ANOVA-based analysis was followed by Tukey's test, and Kruskal-Wallis was followed by Dunn's test for multiple comparisons. All tests were two-sided. Exact p-values are indicated in relevant figures up to ten decimal places. Statistical differences between groups were assessed using GraphPad Prism software. No statistical method was used to pre-determine sample size. Experiments were performed without randomisation or blinding.

## Reporting summary

Further information on research design is available in the Nature Portfolio Reporting Summary linked to this article.

## Data availability

Single-cell RNA and ATAC sequencing data generated in this study have been deposited in the NCBI GEO database under accession codes GSE163218, and GSE303427. All data generated in this study are provided in the Source Data file. Source data are provided with this paper.

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

## Acknowledgements

We thank members of the Alcolea's lab for comments and suggestions; Oriana Oniciuc (Core Bioinformatics group, Cambridge Stem Cell Institute) for her contribution to the scRNA-seq analysis; the staff of the University Biomedical Services Gurdon Institute; Peter Humphreys and Darran Clements in imaging core facilities at JCBC; Ian J. Jackson (Fucci2a); Hans Clevers (Rosa26Confetti); NIHR Cambridge BRC Cell Phenotyping Hub for FACS support; Cambridge Cancer Institute Genomics Facility for sequencing; the Jeffrey Cheah Biomedical Centre (JCBC) core facilities. We thank Paola Bonfanti for critical discussions on lineage conversion. This work was mainly supported by funding from the Wellcome Trust and The Royal Society (105942/Z/14/Z and 105942/Z/14/A to M.P.A), Leverhulme Trust (RPG-2023-136 to M.P.A), and LEO Foundation (LF-OC-24-001613 to M.P.A). This research was funded in whole, or in part, by the Wellcome Trust [203151/Z/16/Z, 203151/A/16/Z], the DRP Wellcome Platform award [226795/Z/22/Z], and the UKRI Medical Research Council [MC_PC_17230]. For the purpose of open access, the author has applied a CC BY public copyright licence to any Author Accepted Manuscript version arising from this submission. M.T.B. received funding from the European Union's Horizon 2020 research and innovation programme under the Marie Sklodowska-Curie grant agreement No 794664 (OESOPHAGEAL FATE). M.T.B. was also supported by the Isaac Newton Trust (Research Grants 16.24(e)), Leverhulme Trust (RPG-2023-136), and LEO Foundation (LF-OC-24-001613). P.J.G. was funded by the Wellcome 4-Year PhD Programme in Stem Cell Biology and Medicine (102160/B/13/Z). H.A. was supported by the Leverhulme Trust (RPG-2023-136 to M.P.A). G.S received support from Isaac Newton Trust (Research Grant 21.07(a)), Medical Research Council (MR/P019013/1), Worldwide Cancer Research and Guts UK (19-0192 and 23-0063); Human Frontier Science Program (LT000092/2016-L) and Basic Science Research Program (NRF-2014R1A6A3A01005675) to S.H; Wellcome Trust (206328/Z/17/Z to F.J.C.N, and B.G); B.D.S. acknowledges funding from the Royal Society (E.P. Abraham Research Professorship, RSRP\R\231004), Wellcome Trust (098357/Z/12/Z and 219478/Z/19/Z), and MRC (MR/V005405/1). F.J.C.N. and B.G. acknowledge funding from CRUK (C1163/A21762) and the Wellcome Trust (206328/Z/17/Z).

## Author contributions

M.T.B., P.J.G., J.M., F.J.C.N., and M.P.A. designed, validated, and conducted experiments. I.M. performed single-cell RNA sequencing data analysis supervised by I.I.M.; H.A. performed single-cell RNA and ATAC sequencing data analysis; biological mining of the data was performed by M.T.B., supervised by M.P.A.; M.T.B., I.M., S.H., F.J.C.N., B.G., I.I.M., and M.P.A. contributed to the experimental design for single-cell RNA sequencing; S.H., F.J.C.N., and B.G. provided advice on data analysis. B.C., G.S., and B.D.S. supervised parts of the study and provided expertise in the epithelial stem cell field. M.T.B. and M.P.A. conceived the project, supervised experiments, and wrote the manuscript with input from B.C, G.S., and B.D.S. Review & Editing of the final manuscript by all authors. Funding acquisition by M.T.B., B.G., B.D.S., and M.P.A.; M.P.A. supervised the study.

## Competing interests

The authors declare no competing interests.

## Additional information

[1]Cambridge Stem Cell Institute, Jeffrey Cheah Biomedical Centre, University of Cambridge, Puddicombe Way, Cambridge, UK. [2]Department of Physiology, Development and Neuroscience, University of Cambridge, Cambridge, UK. [3]Department of Haematology, Jeffrey Cheah Biomedical Centre, University of Cambridge, Cambridge, UK. [4]Wellcome Sanger Institute, Hinxton, UK. [5]Gurdon Institute, University of Cambridge, Cambridge, UK. [6]Department of Applied Mathematics and Theoretical Physics, Centre for Mathematical Sciences, University of Cambridge, Wilberforce Road, Cambridge, UK. [7]Present address: Cambridge Institute of Science, Altos Labs, Cambridge, UK. [8]These authors contributed equally: Ilias Moutsopoulos, Harikrishnan Ajith. ✉e-mail: mtb48@cam.ac.uk; mpa28@cam.ac.uk

