## [Transparent Peer Review file · Nature Communications]

Lifting regenerative barriers promotes epithelial cell fate plasticity supporting lineage conversion

Corresponding Author: Dr Maria Alcolea

Version 0:

Reviewer comments:

Reviewer #1

(Remarks to the Author)

This manuscript by Bejar and colleagues investigates the molecular mechanisms that restrict cell fate plasticity in adult epithelial cells, with a focus on the HIF1 α -SOX9 axis as a key barrier to esophageal-to-skin lineage conversion. Using an innovative 3D regenerative culture system, lineage tracing, and single-cell RNA sequencing (scRNA-seq), the authors demonstrate that while esophageal cells exhibit some degree of plasticity, they encounter significant barriers in fully adopting a skin epidermal identity. Importantly, the study reveals that inhibiting HIF1 α or SOX9 enhances cell fate conversion, suggesting promising avenues for regenerative medicine applications.

Strengths :

- The study challenges traditional concepts of rigid stem cell hierarchies by demonstrating that adult epithelial cells retain latent plasticity.
- The heterotypic culture system (esophageal epithelium cultured on skin dermis) provides an elegant model to study cell fate conversion.
- The use of scRNA-seq offers high-resolution insights into cell identity transitions and intermediate states.
- The concept of removing regenerative barriers has important implications for enhancing tissue repair and cell reprogramming strategies.

Overall, this is a well-conceived and experimentally robust study that advances our understanding of epithelial cell plasticity and regenerative barriers. The identification of HIF1 α /SOX9 as key restrictive factors is significant, but the manuscript would be strengthened by addressing the following comments to provide a more comprehensive understanding of the mechanisms governing cell fate plasticity in adult epithelial tissues.

Major comments :

1. A study from the group of Yann Barrandon (Claudinot et al, 2020), has already demonstrated that p63-expressing epithelial stem cells from various tissues (including esophagus) have hairy skin competence. The main novelty of the present study is that the HIF1 α -SOX9 axis is a key barrier to esophageal-to-skin lineage conversion. While this axis is well-characterized, the study would benefit from exploring additional regulatory mechanisms. A single cell multi-omics approach combining scRNA-seq with scATAC-seq would provide deeper insights into the molecular basis of lineage restriction and the epigenetic landscape changes during reprogramming. Indeed, Sox9 has been reported as a pioneer factor in mouse epidermis (Adam et al, 2015 ; Larsimont et al, 2015) and Sox9 conditional knockout in a model of esophageal cell reprogramming in vivo modifies chromatin accessibility (Vercauteren Drubbel et al, 2021)
2. It remains unclear whether lineage rewiring involves a dedifferentiation step and whether this process involves a fetal-like intermediate phenotype, as observed in regenerating stomach (Vallone et al, 2016), intestine (Nusse et al, 2018; Yui et al, 2018) and transcommitted esophageal keratinocytes (Vercauteren Drubbel et al, 2021). This deserves further investigation. Analysis of scRNA-seq data and/or multi-omics data could help addressing this question.
3. The inefficiency of lineage rewiring raises questions about whether all basal progenitors possess the intrinsic competence to transdifferentiate. To what extent are esophageal progenitors homogenous in the scRNA-seq analysis? As suggested in

the first point, scATAC-seq could help determine if certain progenitor subpopulations have an epigenetic background more permissive for transdifferentiation.

4. In the same line, recent studies have highlighted a cephalocaudal gradient of Troy expression in basal epithelial cells of mouse esophagus (Grommisch et al, 2024). In addition, some basal progenitors from the proximal part of the esophagus can differentiate into taste buds (epithelial appendages) (Vercauteren Drubbel & Beck, 2023). The competence of basal progenitors to change lineage may therefore be heterogeneously distributed in the esophagus. Discussing these regional differences would provide better context for the observations.

Additional Suggestions (discussion)

1. While molecular changes associated with lineage conversion are demonstrated, the study lacks assessment of whether converted cells are functionally equivalent to native skin cells (e.g., skin barrier function, response to mechanical stress, wound healing capacity). This should be discussed.

2. The study focuses solely on esophagus-to-skin conversion. Testing other transitions (e.g., esophagus-to-intestine) would reveal whether similar barriers exist in embryologically related tissues. This is really important in the field as esophageal-to-intestinal transdifferentiation may occur under pathological conditions (metaplasia). This is obviously beyond the scope of the manuscript, but this should be discussed.

3. The potential oncogenic implications merit further discussion. While inhibiting HIF1 α -SOX9 increases plasticity (potentially beneficial for regeneration), this could prove risky in cancer contexts. The authors should consider discussing how modulating lineage restriction barriers might affect tumor initiation or progression in epithelial tissues.

References

- Adam RC, Yang H, Rockowitz S, Larsen SB, Nikolova M, Oristian DS, Polak L, Kadaja M, Asare A, Zheng D, et al (2015) Pioneer factors govern super-enhancer dynamics in stem cell plasticity and lineage choice. *Nature* 521: 366–370
- Claudinot S, Sakabe J-I, Oshima H, Gonneau C, Mitsiadis T, Littman D, Bonfanti P, Martens G, Nicolas M, Rochat A, et al (2020) Tp63-expressing adult epithelial stem cells cross lineages boundaries revealing latent hairy skin competence. *Nat Commun* 11: 5645
- Grommisch D, Wang M, Eenjes E, Svetličič M, Deng Q, Giselsson P & Genander M (2024) Defining the contribution of Troy-positive progenitor cells to the mouse esophageal epithelium. *Dev Cell* 59: 1269-1283.e6
- Larsimont J-C, Youssef KK, Sánchez-Danés A, Sukumaran V, DeFrance M, Delatte B, Liagre M, Baatsen P, Marine J-C, Lippens S, et al (2015) Sox9 Controls Self-Renewal of Oncogene Targeted Cells and Links Tumor Initiation and Invasion. *Cell Stem Cell* 17: 60–73
- Nusse YM, Savage AK, Marangoni P, Rosendahl-Huber AKM, Landman TA, de Sauvage FJ, Locksley RM & Klein OD (2018) Parasitic helminths induce fetal-like reversion in the intestinal stem cell niche. *Nature* 559: 109–113
- Vallone VF, Leprovots M, Strollo S, Vasile G, Lefort A, Libert F, Vassart G & Garcia M-I (2016) Trop2 marks transient gastric fetal epithelium and adult regenerating cells after epithelial damage. *Development: dev*.131490
- Vercauteren Drubbel A & Beck B (2023) Single-cell transcriptomics uncovers the differentiation of a subset of murine esophageal progenitors into taste buds in vivo. *Sci Adv* 9: eadd9135
- Vercauteren Drubbel A, Pirard S, Kin S, Dassy B, Lefort A, Libert F, Nomura S & Beck B (2021) Reactivation of the Hedgehog pathway in esophageal progenitors turns on an embryonic-like program to initiate columnar metaplasia. *Cell Stem Cell*: S1934-5909(21)00152–1
- Yui S, Azzolin L, Maimets M, Pedersen MT, Fordham RP, Hansen SL, Larsen HL, Guiu J, Alves MRP, Rundsten CF, et al (2018) YAP/TAZ-Dependent Reprogramming of Colonic Epithelium Links ECM Remodeling to Tissue Regeneration. *Cell Stem Cell* 22: 35-49.e7

Reviewer #2

(Remarks to the Author)

Review of “Lifting regenerative barriers promotes epithelial cell fate plasticity supporting lineage conversion”

By Bejar et al.

Reviewer comments:

Summary:

This manuscript addresses a potentially compelling question, whether esophageal epithelial cells transdifferentiate into skin/hair-follicle lineages under specific in vitro and in vivo conditions-yet it falls short of providing robust data to substantiate its main claims. In particular, the central assertions that full hair-follicle-like structures are derived from esophageal epithelium and that the HIF1 α -SOX9 axis is a key mechanistic barrier are not convincingly supported by the limited immunostaining, insufficient marker validation, and sparse quantitative data presented. As currently written, the paper leaves critical gaps in experimental detail and does not demonstrate that the new structures are fully functional or definitively esophageal in origin, weakening confidence in the broader conclusions.

On a more personal note, I like the idea of the paper and believe it can have broad implications, especially in the context of adult epithelial plasticity and regenerative medicine. However, as currently presented, the manuscript does not sufficiently support its central claims.

Major Comments

1. Extent of Bona Fide Hair-Follicle Formation: The authors provide preliminary evidence (e.g., CD34, KRT17/24 staining, partial pigmentation) that esophageal cells may become hair-follicle-like. However, further morphological and molecular

validation (for instance, K15, or other keratins; staining for melanin, basal layer marker) would strengthen the claim that complete hair follicles are formed rather than partial or rudimentary structures or cells simply filling in the molds left upon epidermal/HF removal.

- There is no clear architectural evidence is provided that the follicles formed are organized into bulge, matrix, ORS, etc.
- Pigmentation is vaguely mentioned; Fontana-Masson staining is needed.
- If sebaceous glands are suggested, perform Oil Red O or equivalent lipid staining to validate. Also Ki67 or proliferation markers would help show proper biology of the SG and basal layer HG, bulge.
- More than anything, I would like to see clear data showing RFP cells in a GFP setting (dermal cells) with markers of HFSC as well as IFE differentiation markers. More high quality multi-color stainings are needed to be convincing. I am not certain that the HFs I see are indeed novel HFs. It appears that in some scenarios these are cells filling in the molds left in the dermis and assume a shape similar to a hair follicle. In such a case would it be surprising that they share some similarity with HF keratinocytes/HF cellular populations as well as esophageal. Wouldn't it be expected that the dermal mold/scaffold would generate such an effect?

2. Markers Confirming Epithelial and Appendage Identities: Because CD34 can also mark hematopoietic or stromal cells, relying on CD34 alone as the readout for hair-follicle stem-cell identity is limiting. Also, the cd34 staining (and others) are far from ideal. It is over exposed and appears somewhat non-specific. In general figure 2B is far from ideal. As stated above a lot more images with GFP and RFP together with markers (wholemounts and/or sections) are required. Again, the manuscript should include additional hair follicle stem cell markers (e.g., K15) to show that esophageal-derived cells fully integrate into different compartments of the follicle. Co-staining for epithelial keratins (e.g., K14) would also clarify that the same esophageal-derived cells are indeed adopting a skin/HF phenotype.

3. No evidence of function: Are the HF structures producing hair shafts? Cycling? Integrating with the skin niche? Grafting of the ex vivo system would be very informative. There is no data on whether these structures are dynamic or terminally differentiated.

4. It is confusing whether this process occurs rarely or frequently. In figures 1D it is very frequent. Also Sup1G states it is more than 90%. Next the authors state: "The limited expression of the HF stem cell marker CD34 in eHFs became even more apparent when compared to native skin (Fig. 2E), unveiling the inefficiency of the EE-to-skin lineage conversion process". Was this the foundation for the rare assumption? The CD34 stains look far from ideal and over exposed. At minimum I would stain using a panel of markers or utilize genetic tools (such as: *Igr5*, *Igr6*, K15, *Lrig*, *Sox9*) before drawing this conclusion.

Also it is key to recall that CD34+ HFSCs grown in vitro also lose cd34 (except when grown as organoids in Matrigel) but maintain their SC features. Could it be that simply the CD34 mark is not normal but that it is not indicative?

If the authors are correct and it's a rare event what are the 90% Hfs seen? Is that just cells filling in the molds that maintain the semi esophageal/HF signature? If so why not examine the effect of such a signature or the heterogeneity within the formed HFs in correlation with function/morphology/cycling/etc?

This point is a critical point of the manuscript and has to be made a lot clearer.

5. Mechanistic Evidence for HIF1 α -SOX9 Axis: The proposed HIF1 α -SOX9 axis is underdeveloped. Pharmacological and genetic manipulation (*Sox9* deletion) show shifts in CD34 expression. Still, additional immunofluorescence panels (e.g., double labeling for SOX9 and HIF1 α) and a more direct demonstration that *Sox9*-null/HIF1 α -inhibited cells adopt stable hair-follicle gene signatures is required.

6. Imaging and presentation: Multiple figures lack clarity and proper zoom-in insets (e.g., Fig 1C). Stratification is mentioned, but not clearly shown or validated.

- Some immunostainings (e.g., Supplementary Fig. 1E) should be repeated with basal and proliferation markers to better characterize the tissue structure.
- Controls are inconsistently presented—e.g., peeled vs. unpeeled dermis, esophagus-only grafts, skin-only grafts. This comparison must be rigorously controlled and quantified.

Additional specific comments organized by Figure

Introduction:

Rationale for Esophagus–Skin Model: The manuscript would benefit from a more compelling explanation of why esophageal tissue was chosen to test epithelial fate switching in a skin-dermis environment. Although the esophagus is squamous and devoid of adnexal structures, it would be helpful to articulate how this setup advances the concept of adult stem cell plasticity or potential clinical applications.

Figure 1

- Include additional details on culture duration and viability, how long the peeled dermis and esophageal strips remain viable in the 3D culture?
- Panel 1C could benefit from zoomed-in insets (e.g., progressive days of re-epithelialization) to visualize early stages of esophageal cell expansion over the dermis.
- 1C-need clearer zoomed in images of the process.
- What stages are required for the formation of the hair follicles arise from esophageal tissue? Author shows the appearance of the follicles without clear indication and discussion on how they are formed. Not clear if mCherry cells only fill out empty areas. Are the follicles in specific stage? What are the stages of hair formation and how are they similar to embryonic formation of hair follicle?
- Confirm the endogenous staining in figure 1c with epidermal markers, basal layer markers, to confirm that mCherry cells

indeed convert to epidermal cells.

- Supplementary figure 1E- cp3 and ki67—need to show the staining in the tissue couples with basal layer marker
- Need farther verification to confirm epidermal tissues, staining with different markers and zoom in images to show stratification, is there epidermal turnover? Is it functional? The author mentions the formation of squamous stratified epithelium but there are no evidence for it.
- Image 1F is not enough to show pigmentation. Perform fontana mason analysis in order to indicate pigmentation. Are there melanocyte stem cells in the follicles?
- Are the structures that appear to be the sebaceous glands, actually the sebaceous glands? Perform oil staining and zoomed in images of the cells.
- Is that the hair shaft in 1F middle and right HF?

Figure 2

- Use additional markers for hair-follicle identity including K15
- Use conventional epithelial markers (K14 for basal cells, K10 or others for differentiation) would confirm that the newly formed structures exhibit proper stratification rather than partial epithelial overgrowth.
- In figure supplementary figure 1C, positive staining of sox 2 indicate the presence of the dermal papilla in the peeled skin, is that true for every area where hair formed? Is the presence of the DP required for hair formation in this case? Is there any data regarding the specific fibroblasts populations contributing?
- You mention low efficiency of hair formation, yet in images such as supp 1 D we see hair formation via esophageal tissue that is almost all over the tissue.

Figure 3

- Please also show GFP channel in all the images in order to see the cells originated from skin stromal origin.
- The conclusion that based on cd34 marker only the regeneration depends on dermal niche is insufficient. Need to check in other stem cell populations and cell types.
- The author shows polyclonal contribution for the esophageal cells make up the hair follicle, yet it will be valuable to add if different clones give rise to different populations and their actual contribution to hair follicle formation

Figure 6

- The text states there is a SOX9–HIF1a Co-localization : “The data showed a positive correlation between SOX9 and HIF1a levels; with HIF1a expressing cells being more likely to be positive for SOX9, as measured by immunostaining (Supplementary Fig. 6C)”, but panels explicitly showing co-staining are not shown
- Need proof of concept for sox9 floxed model, are the knocked out cells the ones express HIF1a?

Figure 7

- The paper mentions a low incidence of complete HF formation in vivo. A bar chart or table quantifying “follicles per graft” across several mice would help interpret these data.
- The image in 7G does not seem to be representative to the data mentioned in the text regarding number of hair follicles per grafts.
- The low frequency of the HF formation event raises questions regarding the validity and the relevance of the mechanism

Reviewer #3

(Remarks to the Author)

Bejar et al. dissect the molecular mechanism of fate permissibility in adult cells using a 3D regenerative culture system where TdT-marked esophageal tissue is placed on epithelium-less GFP skin dermis and the esophageal epithelial (EE) cells crawl out and on to the skin dermal support. They nicely show that esophageal to skin lineage conversion of the migrating EE is polyclonal and inefficient (using CD34+) showing mixed EE and skin epithelial (SE) markers and that conversion is dermal niche cell-dependent, as removal of the LGR5-GFP dermal cells prevents CD34+ conversion in TdT+ migratory cells. scRNA seq of in vitro transitioning TdT cells indicates reactivation of the embryologically expressed HIF1a and Sox9 (Fig. S6F) among other gene expression changes. Addition of HIF1a small molecular inducers/inhibitors or K14Cre-ERT2 conditional Sox9 mutants cells allows more efficient conversion within the denuded skin microenvironment, providing evidence for an epigenetic block for fate conversion. Overall the methodology follows the lab's previous published in vitro wound healing model and standard chamber regeneration methods.

The work follows pioneering work from the Barrandon lab and others (refs. 25-31) showing p63+ non-skin epithelium retains plasticity to convert to skin in the presence of skin dermal cues, and that of the Fuchs lab showing that Sox9 can reprogram skin IFE to HF. While the present work presents an interesting observation, it is still preliminary and not at the level for this journal. The concept of plasticity is not new and the authors focus only on unidirectional plasticity of one tissue in the present work. The conversion requires additional characterization (see below) to distinguish loss of esophageal markers as well as gain of hair follicle vs. interfollicular epithelial fates. The present work provides a new mechanistic insight by showing HIF1a/SOX9 prevent efficient conversion in this system, but need to demonstrate generalizability and reversibility to enhance the overall impact of the work. In addition previous work clearly shows the role for Sox9 as a pioneer factor for hair follicle lineage (Yang PMID: 37488435 ; Kadaja PMID: 24532713; Pyczek PMID: 31867038). The present work showing SOX9 removal enhances contribution to skin HF lineage is quite surprising and contradicts much previous work and so additional controls and experimental data is required to clarify this issue prior to publication.

Major Issues:

1. Generalizability of the work. Barrandon showed cornea, bladder, thymus niches are instructive so would help to confirm this as well in other K14+; p63+ tissues, not just esophagus. Using the authors wound migration assay they could implant one or more of these other tissues using the same TdT-lineage tracing mouse allele and demonstrate the universality of the HIF1a/SOX9 universal block using DMOG/KC7F2/Sox9 fl/fl.
2. Reversibility. Similarly, Barrandon elegantly showed that plasticity is reversible (Bonfanti PMID: 20725041) and the authors should show whether skin to EE transdifferentiation occurs and is stromal dependent. Again this could be done using the same assay performed in reverse with skin epidermal cells migrating onto esophageal mucosa.
3. Stromal Dependence. The authors show dermal dependence in Fig. 3B-F. However a critical control is required and the authors should show that replacing the skin substrate with esophageal dermal cells maintains esophageal fate and prevent skin transdifferentiation, and whether prostate, bladder corneal stromal cells regulate transdifferentiation to the cognate epithelial state rather than skin.
4. Role of Pioneer Sox 9 in HF lineage. Previous work mentioned above indicates Sox9 is required for HF lineage commitment from skin IFE. The authors show that wound EE reactivates embryonic Sox9 expression and K14 CreERT2 Sox9 fl/fl enhances rather than blocks transdifferentiation into the HF compartment. This is surprising and an apparent contradiction from previous work. In particular the quantitation of Sox 9 KO CD34+ in Fig 6B-E looks similar to that in the control and quantitation from Fig 2B, arguing against a true difference. The authors should perform scRNA seq on Sox9 fl/fl TdT eHF (or in Fig. 7H) and compare it to published datasets. For example SOX9 is a major regulator of the hedgehog pathway critical for HF growth and the so if HF growth is occurring via hedgehog /WNT signaling then the results would suggest an alternative HF pioneer pathway distinct from skin IFE to skin HF. This would be of interest to the broader audience.

Minor Critiques:

5. It would be interesting to perform immunofluorescence for some key esophageal and skin transcription factors, like Sox2, Pax9, Lhx2. Whether the CD34+ cells exhibit reduced levels of esophageal transcription factors and elevated skin transcription factors, compared to the CD34- cells or interfollicular cells?
6. In Figure 4H, what are the cells in grey color? From Figure 4G, these cells are annotated as differentiated cells. Which tissue origin are they from?
7. In Figure 4I and 4J, the authors first generated transcriptional signatures for D3, eSkin-D10, and eEE-D10 cells, and then checked how many genes were overlapped in a pairwise way. Can the authors compare the similarity among these three types of cells based on whole transcriptome using PCA or Pearson's correlation analysis?
8. In the scRNA-seq, can the authors distinguish hair follicle cells versus interfollicular cells in the eSkin samples? From the data presented here, these cells should be quite distinguishable.
9. The authors used % CD34+ cells per eHF and % CD34+ HF as index for fate conversion, and found that pharmaceutical inhibiting Hif1a or knockout of Sox9 increased CD34+ percentage. How about cells in the IFE, do they exhibit more conversion towards skin IFE fate?

Minor edits:

10. HIF1 has been recently shown to be the major driver for the skin wound healing response, including vascularization and epithelial and dermal migration. It would be of interest for the authors to discuss the relationship between wound healing and lineage fidelity in light of their work. In particular clusters C8 / C18 at D10 of the assay from this and their previous paper.
11. Pg. 10. Line 8-10 Fig S4F, G) These results reveal that, by partially restricting the regenerative capacity of epithelial cells, the fate conversion process becomes more efficient. The experimental design the authors term "less regenerative" is removing EE dermis and showing that it can transdifferentiate more. The term regenerative is quite vague and should be better defined here. Moreover an alternative explanation is that EE dermis contains inhibitory signals acting on the underlying skin dermis.
12. There are numerous typographic errors.
Line 7 "maintain plasticity at check" should be "in check".
Pg 8 line 1 "...that maker expression" misspelled

Version 1:

Reviewer comments:

Reviewer #1

(Remarks to the Author)

Thank you for giving me the opportunity to re-evaluate the revised manuscript by Bejar and colleagues. I have carefully reviewed both the authors' detailed rebuttal and the revised version of the manuscript. I am satisfied that the authors have addressed my initial concerns in a comprehensive and convincing manner.

First, in response to my request for deeper mechanistic insights beyond the HIF1 α -SOX9 axis, the authors performed single-cell ATAC-seq experiments. These new data (Fig. 5A-C; Supplementary Fig. 4E-G) convincingly demonstrate chromatin

accessibility changes during lineage conversion, with +/-30% of genomic regions altered in oesophageal-derived skin compared to controls. Importantly, they highlight increased accessibility at epidermal identity loci (Lhx2) and decreased accessibility at oesophageal loci (Krt4), providing clear epigenetic support for the conclusions. In addition, the pseudotime analysis (Fig. 5E–N) strengthens the evidence that subsets of cells move toward an epidermal trajectory. This new dataset significantly enhances the mechanistic weight of the manuscript.

Second, I asked whether lineage conversion involves a fetal-like or developmental intermediate state. The authors show that their scRNA-seq analysis indeed detects developmental signatures in certain clusters, particularly in the oesophagus-to-skin transition cluster. Although the number of transitioning cells limited the resolution of this analysis, I appreciate the authors' transparency about these technical challenges and their commitment to address this in future work. I wish these data could be integrated version of the manuscript but I understand that this specific point may be the ground of a future manuscript.

Third, I raised the issue of basal progenitor heterogeneity and regional differences along the esophagus, including the potential contribution of Troy+ cells. The authors have clarified that lineage tracing demonstrates a polyclonal origin of conversion events, and they show that Tnfrsf19 (Troy) is only expressed in scattered basal cells without enrichment in the conversion cluster (Supplementary Fig. 3M). Their new UMAP analyses and discussion (pp. 8–9) strengthen the argument that no single progenitor subpopulation exclusively drives lineage conversion. They also clarified their methodological choice to use the middle portion of the esophagus and provided new data showing no differences across proximal, middle, or distal regions (Response Fig. 3). Together, these additions resolve my earlier concerns about progenitor heterogeneity and anatomical bias.

Fourth, I had suggested further exploration of functional equivalence of converted cells. While the authors appropriately acknowledge that their in vitro model is not intended to achieve complete functional identity, they have added a much more detailed characterization of the converted structures (Fig. 2; Supplementary Fig. 1L–M, 2A–J), including pigmentation, sebaceous-like lipid accumulation, and expression of epidermal markers. These data convincingly reinforce the extent of conversion achieved and are a substantial improvement over the original submission.

Finally, I asked whether the findings generalize to other epithelial contexts. The authors now include new heterotypic culture data with oesophagus, tongue, bladder, and tracheal epithelia (Supplementary Figs. 2K–M and 7K–M), and show that SOX9 acts as a barrier in tracheal epithelium as well. Although they were unable to establish oesophagus-to-intestinal conversions in their culture system, they openly discuss this limitation and highlight the translational importance of this question (pp. 18–20). The revised discussion also explicitly considers the oncogenic risks of modulating plasticity (p. 20), which I view as an essential addition.

In summary, the authors have provided extensive new data, thoughtful clarifications, and expanded discussion. Their revisions substantially strengthen the manuscript, particularly through the addition of scATAC-seq and heterotypic culture experiments. I am satisfied that all my initial concerns have been adequately addressed, and I have no further comments.

Reviewer #2

(Remarks to the Author)

The authors have done an elegant job in thoroughly addressing all of my comments. Their revisions are clear, thoughtful, and have substantially improved the manuscript.

I have no further comments, and I'm pleased to recommend this paper for publication.

Reviewer #3

(Remarks to the Author)

Bejar et al. dissect the molecular mechanism of fate permissibility in adult cells using a 3D regenerative culture system where TdT-marked esophageal tissue is placed on epithelium-less GFP skin dermis and the esophageal epithelial (EE) cells crawl out and on to the skin dermal support. In the revised version, they have expanded the plasticity observation using stroma of other stratified epithelial tissues, showing local partial cell fate acquisition that is blocked by Sox9 and HIF1. They include a more detailed single cell analysis of the fate transition and additional histological characterization of the eeHF conversion. I appreciate the authors expanded discussion of HIF and Sox9, and additional explanation for the HIF/Sox9 barriers and their relationship to committed skin lineages and wound healing. Overall the additional data in this revised version supports a general stratified epithelial signaling environment in the skin dermis that will be of general interest to journal readers.

This reviewer suggests including the intriguing negative data of the inability of skin epithelium to convert to esophageal (Response Fig. 5) and at least discussing the delayed re-epithelialization of esophageal stroma. With the data suggesting the skin stroma provides the local information to convert esophageal mucosa to skin, these additional data provide clues to the strong skin environmental determinants.

Authors' point-by-point response to reviewers (NCOMMS-25-11438)

Bejar et. al. entitled "Lifting regenerative barriers promotes epithelial cell fate plasticity supporting lineage conversion"

Text from reviewers' comments presented in blue italics, our responses to reviewers' questions in black, and quoted references to the revised manuscript text are highlighted in orange italics.

Please note, American English has been changed to British English; hence, the acronyms have been updated accordingly. Most notably, the reference to oesophageal epithelium is now referred to as OE.

REVIEWER COMMENTS

Reviewer #1 (Remarks to the Author):

This manuscript by Bejar and colleagues investigates the molecular mechanisms that restrict cell fate plasticity in adult epithelial cells, with a focus on the HIF1 α -SOX9 axis as a key barrier to esophageal-to-skin lineage conversion. Using an innovative 3D regenerative culture system, lineage tracing, and single-cell RNA sequencing (scRNA-seq), the authors demonstrate that while esophageal cells exhibit some degree of plasticity, they encounter significant barriers in fully adopting a skin epidermal identity. Importantly, the study reveals that inhibiting HIF1 α or SOX9 enhances cell fate conversion, suggesting promising avenues for regenerative medicine applications.

Strengths:

- The study challenges traditional concepts of rigid stem cell hierarchies by demonstrating that adult epithelial cells retain latent plasticity.*
- The heterotypic culture system (esophageal epithelium cultured on skin dermis) provides an elegant model to study cell fate conversion.*
- The use of scRNA-seq offers high-resolution insights into cell identity transitions and intermediate states.*
- The concept of removing regenerative barriers has important implications for enhancing tissue repair and cell reprogramming strategies.*

Overall, this is a well-conceived and experimentally robust study that advances our understanding of epithelial cell plasticity and regenerative barriers. The identification of HIF1 α /SOX9 as key restrictive factors is significant, but the manuscript would be strengthened by addressing the following comments to provide a more comprehensive understanding of the mechanisms governing cell fate plasticity in adult epithelial tissues.

Reviewer #1 - Author response: We thank the reviewer for the overall positive comments, highlighting the "innovative", "well-conceived" and "robust" nature of our work, which offers "promising avenues for regenerative medicine applications". We are also grateful to the reviewer for pointing us towards aspects of the data that needed further consideration in order to strengthen the claims of our study. We have addressed each of the comments as described below.

Major comments:

1. A study from the group of Yann Barrandon (Claudinot et al, 2020), has already demonstrated that p63-expressing epithelial stem cells from various tissues (including esophagus) have hairy skin competence. The main novelty of the present study is that the HIF1 α -SOX9 axis is a key barrier to esophageal-to-skin lineage conversion. While this axis is well-characterized, the study would benefit from exploring additional regulatory mechanisms. A single cell multi-omics approach combining scRNA-seq with scATAC-seq would provide deeper insights into the molecular basis of lineage restriction

and the epigenetic landscape changes during reprogramming. Indeed, Sox9 has been reported as a pioneer factor in mouse epidermis (Adam et al, 2015 ; Larsimont et al, 2015) and Sox9 conditional knockout in a model of esophageal cell reprogramming in vivo modifies chromatin accessibility (Vercauteren Drubbel et al, 2021)

Question 1. Reviewer #1

We thank the reviewer for acknowledging the novelty of our study and for their insightful suggestions which certainly enhance the message of our study. Following the reviewer's comments, we have now performed a single-cell ATAC sequencing experiment (scATAC-seq), including *in vitro* oesophageal-derived skin (oeSKIN), as well as *in vitro* oesophagus (oeOE) and *in vitro* skin (sSKIN) as controls. The new data and analysis can be found in the revised version of the manuscript (**Fig. 5A, B and Supplementary Fig. 4E-G**).

The UMAP projection of the scATAC-seq data recapitulates the overall scRNA-seq profile, revealing that a subset of oeSKIN cells acquires a chromatin accessibility landscape more closely resembling that of the epidermal control (sSKIN cells; **see cluster 6; Supplementary Fig. 4E-G**). This, as suggested by the scATAC-seq pseudotime analysis, indicates a shift in chromatin accessibility trajectory consistent with acquisition of epidermal identity (**Fig. 5L**).

Overall, the data demonstrate that oesophageal-derived skin cells at the culture end-point exhibited changes in chromatin accessibility at ~30% of genomic regions compared to oesophageal control (oeOE). Supporting the skin-to-oesophageal lineage conversion process, we find increased accessibility at loci associated with epidermal identity and decreased accessibility at loci linked to oesophageal identity (**Fig. 5B, C and Supplementary Fig. 4G**). In particular, we observe that chromatin regions associated with identity switch loci, including *Hif1a* (plasticity barrier-related), present a more open state in oeSKIN cells at the transition between oesophagus and skin (along the scATAC-seq pseudotime trajectory; **Fig. 5M,N**), further reinforcing our overall claim.

Accordingly, in line with our histological characterisation, as well as existing scRNA-seq data, the scATAC-seq analysis also highlights the inefficiency of the lineage conversion process, with only a limited subset of oesophageal cells showing a shift in chromatin accessibility consistent with epidermal identity (**see cluster 6; Supplementary Fig. 5E-G; Fig. 5L**). This reinforces the main message of our study highlighting the existence of inherent barriers to cell fate plasticity. However, this inefficiency also comes with technical limitations, rendering very few transitioning cells available for in-depth molecular analysis; an aspect we have found challenging when performing the current scATAC-seq analysis. Despite this, we hope the reviewer agrees that our new data robustly supports the lineage conversion process and reinforces our mechanistic observations.

2. It remains unclear whether lineage rewiring involves a dedifferentiation step and whether this process involves a fetal-like intermediate phenotype, as observed in regenerating stomach (Vallone et al, 2016), intestine (Nusse et al, 2018; Yui et al, 2018) and transcommitted esophageal keratinocytes (Vercauteren Drubbel et al, 2021). This deserves further investigation. Analysis of scRNA-seq data and/or multi-omics data could help addressing this question.

Question 2. Reviewer #1

We fully appreciate the reviewer's comment. Indeed, during our initial scRNA-seq analysis we had spotted a foetal-like/developmental signature in our *in vitro* re-epithelisation assay. In particular, we

observed that, as anticipated for a regenerative *ex vivo* model, basal cells show a heterogeneous foetal-like/developmental signature; with some cell clusters presenting a more prominent developmental signature than others (see below **Response Fig. 1**). Interestingly, Cluster 8, which contains oesophageal-to-skin transitioning cells, emerged as a cluster with a variable developmental signature, depending on study used as reference signature¹⁻¹⁰ (see below **Response Fig. 1**). We presume this variability may be linked to technical limitations associated with the reduced number of cells found to undergo lineage conversion in our system. Since we agree on the importance of this point, we are currently working on scaling up our approach to be able to dissect this interesting point in-depth as part of a follow up project. This represents future efforts that are currently undergoing in my laboratory. We hope the reviewer agrees with us that this does not preclude or impact the observations presented in the current manuscript.

Response Figure 1: Violin plots showing the distribution of previously published gene signatures scores across basal cell clusters in our single-cell RNA-seq dataset (as in Fig. 4B). Signature scores were calculated per cell using Seurat's AddModuleScore function, which normalises for expression levels using matched control genes to provide a relative measure of gene set activity.

Embryonic/developmental oesophageal signatures were derived from Yang et al. (2025)⁴, Kumar et al. (2024)³, Magaletta et al. (2022)¹, McGinn et al. (2021)⁵, and Kuwahara et al. (2020)². Foetal-like transitioning cell signatures were obtained from Chen et al. (2023, intestine)⁶, Vercauteren et al. (2021, oesophagus)⁹, Yui et al. (2018, intestine)¹⁰, Nusse et al. (2018, intestine)⁷, and Vallone et al. (2016, stomach)⁸.

3. The inefficiency of lineage rewiring raises questions about whether all basal progenitors possess the intrinsic competence to transdifferentiate. To what extent are esophageal progenitors homogenous in the scRNA-seq analysis? As suggested in the first point, scATAC-seq could help determine if certain progenitor subpopulations have an epigenetic background more permissive for transdifferentiation.

Question 3. Reviewer #1 Please consider this response in conjunction with **Question 4. Reviewer #1**, as both are related.

We agree with Reviewer #1 that, given the inefficiency of the lineage conversion process and the emerging recognition of heterogeneity within the oesophageal basal cell population^{9,11,12}, the possibility that oeHFs emerge from a specific cellular subpopulation warrants further discussion. We apologise for the lack of clarity; this was the intended purpose of **sections 3 and 4** of results (lineage tracing sections). Using unbiased lineage tracing of epithelial oesophageal cells, we conclude that the change in identity depends on niche signals from the skin dermis, rather than on the oesophageal cell of origin (see revised manuscript for further details; **section 4** of results).

Although, we cannot rule out the involvement of diverse equipotent basal cell populations (known or yet to be described). Our scRNA-seq and newly added scATAC-seq data do not indicate any apparent basal heterogeneity. However, low-abundance populations like may not be readily detectable unless they form a distinct and cohesive cluster with a unique transcriptional and/or chromatin accessibility profile.

As rightly indicated by the reviewer in question 5, a recent paper (Grommisch et al., 2024¹¹) has described a Troy⁺ basal cell subpopulation in the mouse oesophagus that shows a proximo-distal gradient of expression. The authors showed that this population restricts progenitor proliferation and enables commitment to differentiation. To address the reviewer's point, we confirm that we also find *Tnfrsf19* (Troy) to be expressed in scattered cells across the basal population both *in vivo* (postnatal development and adult samples; McGinn et al., 2021)⁵, as well as in our heterotypic culture model (see scRNA-seq UMAPs below **Response Fig. 2**). Consistent with the findings by Grommisch et al., Troy⁺ cells represent a small disperse fraction of basal cells, failing to form a cohesive cluster. We also observe a modest enrichment in the resting basal population transitioning towards differentiation (**Response Fig. 2**). Notably, the lineage conversion cluster (C8) associated with the oesophageal identity switch does not express *Tnfrsf19*. While we cannot entirely exclude the contribution of Troy⁺ cells, the low frequency of *Tnfrsf19*-expressing cells in our system and the polyclonal nature of eHF formation (as shown by our lineage tracing analysis, **Figs. 3 and 4**) make it improbable that Troy⁺ cells are the sole contributors to lineage conversion.

In summary, while multiple basal cell pools may contribute to lineage conversion, our lineage tracing data suggest that no single basal subpopulation is solely responsible of this process, with oeHF formation being a widespread and polyclonal process.

We have now included in the revised manuscript UMAP plots showing *Tnfrsf19* expression (**Supplementary Fig. 3M**) and have added a reference in the revised manuscript to acknowledge the potential contribution of Troy⁺ and other basal cell populations to the lineage conversion process. The text reads:

“Given the inefficiency of the process and the emerging recognition of heterogeneity within the oesophageal basal cell population^{9,11,12}, it is important to consider the dynamics of basal cell

contribution to the lineage conversion process. To determine whether the basis of the inefficient OE-to-skin fate re-specification was associated with restricted competency of a heterogeneous cell population, we analysed the KRT15 and CD34 expression within oeHF confetti labelled clones.”... (p8; l18-23)

...”While we cannot discard that multiple oesophageal basal cell pools may contribute to lineage conversion, our lineage tracing data suggest no single basal subpopulation is solely responsible for this process.”.... (p9; l11-12).

Redacted

Grommisch et al., 2024

McGinn et al., 2021

Bejar et.al (manuscript)

Response Figure 2: Cell distribution in the UMAP dimensionality reduction space showing *Tnfrsf19* (*TROY*) expression as per Grommisch et al., 2024¹¹ (adult oesophagus), McGinn et al., 2021⁵ (Postnatal and adult oesophagus), and Bejar et.al. (current manuscript, **Supplementary Fig. 3M**) from left to right.

4. In the same line, recent studies have highlighted a cephalocaudal gradient of Troy expression in basal epithelial cells of mouse esophagus (Grommisch et al, 2024). In addition, some basal progenitors from the proximal part of the esophagus can differentiate into taste buds (epithelial appendages) (Vercauteren Drubbel & Beck, 2023). The competence of basal progenitors to change lineage may therefore be heterogeneously distributed in the esophagus. Discussing these regional differences would provide better context for the observations.

Question 4. Reviewer #1 Please consider this response in conjunction with **Question 3. Reviewer #1**, as both are related.

We thank the reviewer for raising this point and apologise for the lack of clarity. In our study we use the middle half of the oesophagus rather than the entire oesophagus, we have clarified this in the revised manuscript (refer to **Results section 1** and **Methods section; 3D Oesophageal-Dermal Heterotypic and control cultures**). Given the obvious anatomical and architectural differences, as well as emerging cellular heterogeneity^{11,12} between the main body of the oesophagus and the most proximal and distal areas, we focused our work on the main body of the oesophagus to minimise any potential cephalocaudal complexity. See below **Response Fig. 3** for the details on oesophageal preprocessing; the most proximal and distal parts of the oesophagus are cut out prior any further downstream assay, including heterotypic cultures. This allows us to refer back to our original study on oesophageal cell fate dynamics where a similar approach was used¹³.

However, given the importance of this point, we have repeated our heterotypic cultures further separating tissue samples (middle half of the oesophagus) by their anatomical position along the longitudinal axis of the oesophagus (proximal, middle, distal, see below **Response Fig. 3**). The aim here is to determine whether our original observations may have been hindered by artefacts related to the longitudinal origin of the samples used. Interestingly, we did not see any differences either in the re-epithelialisation efficiency or lineage conversion process, as assessed by the emergence of HF markers, CD34 and KRT15 (**Response Fig. 3**).

Response Figure 3: (A) Representative images oesophageal tissue pre-processing method. Upon dissection, the most proximal and distal quarters of the oesophagus were discarded. Only the middle half of the oesophagus was used to generate 3D organ cultures on denuded oesophageal or skin stroma (dermis).

(B) Image showing samples used for new regional heterotypic cultures assays. Oesophagi were further cut into defined proximal, middle and distal regions, and used to assess regional differences in our heterotypic organ cultures.

(C) Re-epithelialisation of dermis after 8 days of culture, expressed as percentage of area covered.

(D-E) Quantification of HF marker expression in oesophageal-derived HF derived from different regions along the longitudinal axis of the oesophagus. CD34 and KRT15 intensity showed no significant differences across samples. (n=3 biological replicates, 3 technical replicates each).

Additional Suggestions (discussion)

1. While molecular changes associated with lineage conversion are demonstrated, the study lacks assessment of whether converted cells are functionally equivalent to native skin cells (e.g., skin barrier function, response to mechanical stress, wound healing capacity). This should be discussed.

Minor point 1. Reviewer #1

We fully appreciate the reviewer's point. In response to this point, we have expanded our analysis with a more comprehensive characterisation of oesophageal-derived hair follicles (oeHFs). Specifically, we have included staining of oeSKIN wholemounts for various epidermal markers, such as LHX2, KRT15, KRT10, SCD1, and SCA-1 (Fig. 2 and Supplementary Fig. 2A-J), which further support the conversion from oesophageal to skin lineage. Additionally, we utilised Oil Red O staining to highlight lipid accumulation in oesophageal cells (akin of sebaceous glands; Supplementary Fig. 1M), and Fontana Masson staining to track melanin incorporation within the oeHFs (Supplementary Fig. 1L). These new data strengthen the evidence for the lineage conversion process. The text has been amended in line with the new data provided:

“Remarkably, oesophageal-derived hair follicles oeHFs (red) showed features reminiscent of the in vivo appendages. These included pigmented fibre-like structures (Fig. 1G and Supplementary Fig. 1K) positive for Fontana-Masson stain, which labels melanin (Supplementary Fig. 1L). Oesophageal-derived HFs were also positive for Oil Red O staining, indicating the accumulation of sebum in the proximal area where sebaceous glands are typically found (Supplementary Fig. 1M).”(p5, l24-28).

Finally, while our data provide compelling evidence for early lineage conversion events, we wish to reiterate that the central novelty of this study lies in demonstrating the presence of lineage barriers that inherently constrain inappropriate lineage conversion (regulated, at least in part, via the HIF1a-SOX9 axis). In this context, we do not anticipate observing a complete identity switch in our *in vitro* model. And, hence, attempting a direct functional comparison between oesophageal-derived HFs and skin would be beyond reach in the context of our observations. We hope the reviewer will be satisfied by the deeper marker oeHF characterisation provided in the revised manuscript.

2. The study focuses solely on esophagus-to-skin conversion. Testing other transitions (e.g., esophagus-to-intestine) would reveal whether similar barriers exist in embryologically related tissues. This is really important in the field as esophageal-to-intestinal transdifferentiation may occur under pathological conditions (metaplasia). This is obviously beyond the scope of the manuscript, but this should be discussed.

Minor point 2. Reviewer #1

We thank the reviewer for their insightful comment. Following the reviewer's recommendation, we have now included heterotypic cultures combining epithelium and stromal from additional epithelial tissues. These include oesophageal cells exposed to the ectopic stroma of the tongue and bladder (Supplementary Fig. 2K-M), tracheal epithelium exposed to denuded dermis (Supplementary Fig. 7K-M), as well as relevant controls. As anticipated, epithelial cells expressed ectopic markers as instructed by the stroma they were exposed to; KRT8 for denuded tongue and bladder stroma (Supplementary Fig. 2K-M), CD34 and KRT15 for denuded dermis (Supplementary Fig. 7K-M). We have further shown that the SOX9 lineage conversion barrier also operates in the tracheal epithelium (Supplementary Fig. 7L, M), revealing the relevance of our mechanistic insights beyond the oesophageal epithelium.

Response to reviewers' NCOMMS-25-11438

Regarding the oesophagus-to-intestine transition, we have indeed attempted to perform our heterotypic cultures by growing oesophageal epithelium over the stroma of glandular tissues, including the stomach and the intestine. However, we found that our *in vitro* assay does not allow for epithelial cells to repopulate glandular tissue stroma. We have tested a number of different culture media conditions, including our standard conditions as well as intestinal and stomach organoid media. However, epithelial cells were not able to efficiently grow out of the original tissue explant under any of the conditions tested. Given its translational relevance, follow-up work, beyond this study, will focus on optimizing culture conditions to address this important question.

We have revised the manuscript to highlight the translational relevance of understanding epithelial cell fate plasticity in the context of oesophageal metaplasia. The revised text now reads: *“It has now become widely accepted that cell fate decisions are not as predetermined as originally thought. In response to injury or environmental perturbations, adult epithelial cells present a remarkable fate plasticity allowing them to respond to changes in tissue needs¹⁴⁻¹⁶. Yet, if dysregulated, this adaptability can promote epithelial disease and cancer^{15,17-19}. Particularly, in the context of the oesophagus, columnar metaplasia has been associated with the development of oesophageal cancer^{9,20-22}. Hence, understanding the largely unclear mechanisms balancing cell fate plasticity is critical to unveil its regenerative potential while preventing inherent tumorigenic risks²³.”* (p18; l2-8).

3. The potential oncogenic implications merit further discussion. While inhibiting HIF1 α -SOX9 increases plasticity (potentially beneficial for regeneration), this could prove risky in cancer contexts. The authors should consider discussing how modulating lineage restriction barriers might affect tumor initiation or progression in epithelial tissues.

Minor point 2. Reviewer #1

We completely agree with the reviewer and thank them for this word of caution. Understanding the mechanisms modulating cell fate plasticity indeed holds great promise to promote tissue repair and unleash the true regenerative nature of mammalian tissues. However, it does come at a cost. The activation of plasticity cell fate programs or the disruption of lineage barriers carries inherent risks, including aberrant tissue growth and malignant transformation^{15,17-19,23}. Hence, although diving into the mechanisms modulating cell fate plasticity are of utmost importance, they should underscore the need for caution. A major challenge lies ahead of us to precisely understanding how to fine-tune epithelial plasticity; maximizing its reparative potential while minimizing unintended pathological consequences. We have highlighted this in the revised version of the manuscript, which now reads: *“This study opens exciting new avenues to decipher the inner workings dictating cell fate plasticity, providing a benchmark to identify the principles underlying the full regenerative capacity of epithelial cells. However, given the potential risk of abnormal tissue growth and malignancy^{15,17-19,23}, it also underscores the need for caution. Careful consideration must be given when developing strategies to either promote tissue plasticity or disrupt plastic barriers in order to minimise unintended pathological consequences. Much work is still needed to understand how to finely balance the plastic response of our epithelial tissues or therapeutic gain.”* (p20; l4-10).

References

Adam RC, Yang H, Rockowitz S, Larsen SB, Nikolova M, Oristian DS, Polak L, Kadaja M, Asare A, Zheng D, et al (2015) Pioneer factors govern super-enhancer dynamics in stem cell plasticity and lineage choice. Nature 521: 366–370

Response to reviewers' NCOMMS-25-11438

Claudinot S, Sakabe J-I, Oshima H, Gonneau C, Mitsiadis T, Littman D, Bonfanti P, Martens G, Nicolas M, Rochat A, et al (2020) Tp63-expressing adult epithelial stem cells cross lineages boundaries revealing latent hairy skin competence. Nat Commun 11: 5645

Grommisch D, Wang M, Eenjes E, Svetličič M, Deng Q, Giselsson P & Genander M (2024) Defining the contribution of Troy-positive progenitor cells to the mouse esophageal epithelium. Dev Cell 59: 1269-1283.e6

Larsimont J-C, Youssef KK, Sánchez-Danés A, Sukumaran V, Defrance M, Delatte B, Liagre M, Baatsen P, Marine J-C, Lippens S, et al (2015) Sox9 Controls Self-Renewal of Oncogene Targeted Cells and Links Tumor Initiation and Invasion. Cell Stem Cell 17: 60–73

Nusse YM, Savage AK, Marangoni P, Rosendahl-Huber AKM, Landman TA, de Sauvage FJ, Locksley RM & Klein OD (2018) Parasitic helminths induce fetal-like reversion in the intestinal stem cell niche. Nature 559: 109–113

Vallone VF, Leprovots M, Strollo S, Vasile G, Lefort A, Libert F, Vassart G & Garcia M-I (2016) Trop2 marks transient gastric fetal epithelium and adult regenerating cells after epithelial damage. Development: dev.131490

Vercauteren Drubbel A & Beck B (2023) Single-cell transcriptomics uncovers the differentiation of a subset of murine esophageal progenitors into taste buds in vivo. Sci Adv 9: eadd9135

Vercauteren Drubbel A, Pirard S, Kin S, Dassý B, Lefort A, Libert F, Nomura S & Beck B (2021) Reactivation of the Hedgehog pathway in esophageal progenitors turns on an embryonic-like program to initiate columnar metaplasia. Cell Stem Cell: S1934-5909(21)00152–1

Yui S, Azzolin L, Maimets M, Pedersen MT, Fordham RP, Hansen SL, Larsen HL, Guiu J, Alves MRP, Rundsten CF, et al (2018) YAP/TAZ-Dependent Reprogramming of Colonic Epithelium Links ECM Remodeling to Tissue Regeneration. Cell Stem Cell 22: 35-49.e7

Reviewer #2 (Remarks to the Author):

Review of “Lifting regenerative barriers promotes epithelial cell fate plasticity supporting lineage conversion”

By Bejar et al.

Reviewer comments:

Summery:

This manuscript addresses a potentially compelling question, whether esophageal epithelial cells transdifferentiate into skin/hair-follicle lineages under specific in vitro and in vivo conditions-yet it falls short of providing robust data to substantiate its main claims. In particular, the central assertions that full hair-follicle-like structures are derived from esophageal epithelium and that the HIF1 α -SOX9 axis is a key mechanistic barrier are not convincingly supported by the limited immunostaining, insufficient marker validation, and sparse quantitative data presented. As currently written, the paper leaves critical gaps in experimental detail and does not demonstrate that the new structures are fully functional or definitively esophageal in origin, weakening confidence in the broader conclusions.

On a more personal note, I like the idea of the paper and believe it can have broad implications, especially in the context of adult epithelial plasticity and regenerative medicine. However, as currently presented, the manuscript does not sufficiently support its central claims.

Reviewer #2 - Author response: We are most grateful to the reviewer for appreciating the “*compelling question*” addressed in our study and recognizing the “*broad implications*” of our work “*especially in the context of adult epithelial plasticity and regenerative medicine*”.

In light of the reviewer’s detailed and insightful assessment of our study, we have implemented the proposed recommendations, which further enhance the robustness of our results and strengthen the manuscript.

Major Comments

1. Extent of Bona Fide Hair-Follicle Formation: The authors provide preliminary evidence (e.g., CD34, KRT17/24 staining, partial pigmentation) that esophageal cells may become hair-follicle-like. However, further morphological and molecular validation (for instance, K15, or other keratins; staining for melanin, basal layer maker) would strengthen the claim that complete hair follicles are formed rather than partial or rudimentary structures or cells simply filling in the molds left upon epidermal/HF removal.

- There is no clear architectural evidence is provided that the follicles formed are organized into bulge, matrix, ORS, etc.*
- Pigmentation is vaguely mentioned; Fontana-Masson staining is needed.*
- If sebaceous glands are suggested, perform Oil Red O or equivalent lipid staining to validate. Also Ki67 or proliferation markers would help show proper biology of the SG and basal layer HG, bulge.*
- More than anything, I would like to see clear data showing RFP cells in a GFP setting (dermal cells) with markers of HFSC as well as IFE differentiation markers. More high quality multi-color stainings are needed to be convincing. I am not certain that the HFs I see are indeed novel HFs. It appears that in some scenarios these are cells filling in the molds left in the dermis and assume a shape similar to a hair follicle. In such a case would it be surprising that they share some similarity with HF keratinocytes/HF cellular populations as well as esophageal. Wouldn't it be expected that the dermal mold/scaffold would generate such an effect?*

Question 1. Reviewer #2

We thank the reviewer and acknowledge the importance of their comments and suggestions. As requested, the revised manuscript now includes a deeper characterisation of oesophageal derived hair follicles (**oeHFs**), including additional markers that reinforce the oesophageal-to-skin lineage conversion notion. New data is summarised below.

- a) To further support the **oesophageal switch towards HF identity**, we have included immunofluorescent images showing additional epidermal markers in oesophageal-derived skin (**oeSKIN**), including LHX2, KRT15, KRT10, SCD1 and SCA-1 (**Fig. 2 and Supplementary Figs. 2A-E**).

To strengthen the oesophageal-to-skin identity switch, we have also included a more comprehensive marker analysis showing the downregulation of key oesophageal markers, including Krt4 and Sox2, within oesophageal-derived skin (**oeSKIN**; **Supplementary Figs. 2F-H**). To add additional depth to this point, we have made use of our scRNA-seq data to identify differentially expressed genes (DEG) between oesophagus and skin, and used them to illustrate the cell fate transition. As anticipated, the data show that genes enriched in **oeOE** versus **sSKIN** are expressed at lower levels in oesophageal-derived skin (**oeSKIN**; **Figs. 4L-N**).

Additionally, regarding the request of more markers, we would like to note that there is a limited repertoire of markers known to be specific to skin, i.e. not expressed in the

oesophagus. Despite this, we hope the reviewer agrees that our newly added data robustly supports the lineage conversion process.

- b) Single-cell ATAC sequencing experiment (scATAC-seq); we have performed scATAC-seq, including *in vitro* oesophageal-derived skin (**oeSKIN**), as well as *in vitro* oesophagus (**oeOE**) and *in vitro* skin (**sSKIN**) as controls. The new data and analysis can be found in the revised version of the manuscript (**Fig. 5A and Supplementary Fig. 4E-G**).

The UMAP projection of the scATAC-seq data recapitulates the overall scRNA-seq profile, revealing that a subset of **oeSKIN** cells acquires a chromatin accessibility landscape more closely resembling that of the epidermal control (**sSKIN** cells; **see cluster 6; Supplementary Fig. 4E-G**). This, as suggested by the scATAC-seq pseudotime analysis, indicates a shift in chromatin accessibility trajectory consistent with acquisition of epidermal identity (**Fig. 5L**).

- c) In relation to **oeHF** pigmentation, the Fontana Mason staining now included in the revised manuscript shows that the denuded dermis retains melanin after peeling, which may be taken up by oesophageal keratinocytes during **oeHF** formation (**Supplementary Fig. 1L**). We further support this notion by showing how pigmented structures in **Fig. 1G** and **Supplementary Fig. 1K** are indeed of oesophageal origin (i.e. express Tomato reporter).
- d) To confirm the conversion towards a sebaceous gland identity we show that, besides the acquisition of the sebaceous gland morphology at the upper region, **oeHFs** also contain cells expressing the sebaceous gland marker stearyl-CoA desaturase 1 (**SCD1; Figs. 2B and Supplementary Fig. 2A**), and can accumulate lipid droplets (as shown by Oil Red O staining in **Supplementary Fig. 1M**).
- e) In relation to the request of images showing **GFP and RFP together with markers**, it's important to mention that, to enable good antibody penetration and 3D confocal reconstruction of the tissue, **oeSKIN** wholemounts are peeled (dermal composite was removed prior immunostaining). Hence, epithelial stainings in most cases only show the epithelial compartment post-wholemount peeling. Please note that EGFP confocal channel (green channel) is always ON (as stated in the figure caption). Hence, the lack of EGFP signal merely shows the lack of EGFP skin-derived cells in the **oeSKIN** wholemounts. Where possible images showing tdTomato (OE) and EGFP (dermal) composites are shown (**Figs. 1C, E and Supplementary Figs. 1F, 2D and 7E-I**).

As per **oeHF** architecture, we agree with the reviewer that emerging **oeHFs** do not yet show a fully mature HF morphology and marker compartmentalisation. However, we have not been surprised by this observation given the current challenges associated with recreating hair follicle-like structures in *in vitro* systems²⁴⁻²⁸. As far as we are aware, the only approach able to fully reproduce this *in vitro* has been achieved by recapitulating the HF developmental programme in iPSCs derived hair follicle organoids^{26,27}. Otherwise, studies have traditionally relied on *in vivo* grafting assays to show the emergence of relevant markers in lineage conversion models underscoring the challenge of hair follicle marker detection in *in vitro* systems^{29,30}. To further support the notion that this is a feature associated with *ex vivo* conditions, we have now included additional controls, including skin-derived *in vitro* HFs (**sHF**) in relevant figures, which show that the unusual morphology and ectopic marker expression are found in *in vitro* reconstituted HFs regardless of their skin or oesophageal origin. We have alluded to this important point in the revised manuscript: *"Of note, oeHFs showed atypical morphology and ectopic expression, with markers localised outside their anatomical position within in vivo skin HFs. This, however, was observed to be a recurrent feature of ex vivo reconstituted HFs regardless of their skin or oesophageal origin (Compare sSKIN and oeSKIN versus Skin in (Fig. 2B and Supplementary Fig.*

2A) *in line with previous reports^{26,27,31-33}. Reassuringly, oesophageal control samples in vivo (OE) and in vitro (oeOE) confirmed the lineage specificity of these skin markers (Supplementary Figs 2B, C)³⁴.*" (p6; l15-22).

Finally, despite the compelling evidence pointing towards early lineage conversion events, as provided by existing and newly presented experiments, we would like to reinforce the point that the novelty of this study centres on the existence of lineage barriers, which naturally limit inadequate lineage conversion events. Hence, under this premise, we would not anticipate a complete identity switch in our *in vitro* model system. On the contrary, we believe that this highly reproducible and scalable system of partial lineage conversion offers an unprecedented model where to start elucidating the largely unknown mechanisms regulating the transition states associated with cell fate plasticity.

2. Markers Confirming Epithelial and Appendage Identities: Because CD34 can also mark hematopoietic or stromal cells, relying on CD34 alone as the readout for hair-follicle stem-cell identity is limiting. Also, the cd34 staining (and others) are far from ideal. It is over exposed and appears somewhat non-specific. In general figure 2B is far from ideal. As stated above a lot more images with GFP and RFP together with markers (wholamounts and/or sections) are required. Again, the manuscript should include additional hair follicle stem cell markers (e.g., K15) to show that esophageal-derived cells fully integrate into different compartments of the follicle. Co-staining for epithelial keratins (e.g., K14) would also clarify that the same esophageal-derived cells are indeed adopting a skin/HF phenotype.

Question 2. Reviewer #2

We acknowledge the reviewer's point regarding HF markers. To address this comment, we have now added a deeper characterisation of oesophageal derived hair follicles (**oeHFs**). New data includes:

- a) **CD34**- We acknowledge the fact that CD34 may potentially come from hematopoietic or stromal cells. For this reason, we originally included the co-staining of KRT14 (basal epithelial marker), PDGFRa (fibroblast markers) and CD34 (HF putative marker) in **oeSKIN**, see **Supplementary Fig. 2E**. Our results show that CD34 expression indeed colocalises with KRT14 in oesophageal epithelial cells within **oeHFs**. And, hence, CD34 expression in **oeHFs** is not the result of stromal contamination.
- b) **Keratin 15**- Additionally, to address the reviewer's concerns regarding the fidelity of CD34 as HF marker, we have now included in relevant experiments (including HIF1a-SOX9 functional assays), not only CD34, but also KRT15, as HF markers (**Figs. 2, 3E, 5O-R and 6, Supplementary Figs. 2A-C, 3A-C and 6J-L**). Our new data further supports the inefficiency of the lineage conversion process, as shown the KRT15 expression quantification, and demonstrates that this is not limited to CD34 expression (**Figs. 2D, E**). Overall, these new results reinforce our claims regarding the lineage conversion process, as well as the proposed regulatory mechanism via the HIF1a-SOX9 axis.
- c) **Hair reconstitution assays**- Additionally, we would like to reinforce that existing functional experiments also show that, in *in vivo* hair reconstitution assays, **oeHFs** were entirely derived from oesophageal cells (exclusively tomato+ **oeHFs**; **Fig. 7A-H and Supplementary Fig. 7A-I**). And, what is more, *in vivo* **oeHF** formation may be modulated by targeting the HIF1a-SOX9 axis. Both KC7F2 treatment, which inhibits HIF1a, and genetic SOX9 loss of function experiments led to an increased formation of oesophageal derived HFs in grafting experiments using primary cells (**Fig 7F-H**). We believe that these results robustly support the existence of

lineage conversion barriers, and show that the lineage conversion in this context is a regulated process and not a mere experimental artefact.

- d) **HF Markers (as indicated above)**- Staining of oeSKIN wholemounts for additional markers supporting the conversion from oesophageal to skin lineage (**Fig. 2 and Supplementary Figs. 1L and 2A-E**).

3. No evidence of function: Are the HF structures producing hair shafts? Cycling? Integrating with the skin niche? Grafting of the ex vivo system would be very informative. There is no data on whether these structures are dynamic or terminally differentiated.

Question 3. Reviewer #2

We thank the reviewer for their comment and fully acknowledge the importance of assessing hair follicle (HF) cycling in reconstitution assays. Understandably, the current heterotypic approach presents limitations, as do most existing reconstitution systems. To date, and to the best of our knowledge, even state-of-the-art iPSC-derived HF organoids have not yet been shown to cycle^{26,27}. Hence, it seems rather unremarkable that we have been unable to follow hair cycling in our 3D oesophageal-derived skin cultures.

Nonetheless, we do show that epithelial cells in oeHFs proliferate *in vitro* (**Supplementary Figs. 1E, F**) and contribute to rudimentary hair shafts in *in vivo* skin reconstitution assays (**Fig 7**), thereby demonstrating the ability of oeHFs to generate functional hair structures (to a certain extent). Whether such *ex vivo*-generated HFs are capable of undergoing long-term cycling remains a major unresolved challenge in the field.

4. It is confusing whether this process occurs rarely or frequently. In figures 1D it is very frequent. Also Sup1G states it is more than 90%. Next the authors state: "The limited expression of the HF stem cell marker CD34 in eHFs became even more apparent when compared to native skin (Fig. 2E), unveiling the inefficiency of the EE-to-skin lineage conversion process". Was this the foundation for the rare assumption? The CD34 stains look far from ideal and over exposed. At minimum I would stain using a panel of markers or utilize genetic tools (such as: lgr5, lgr6, K15, Lrig, Sox9) before drawing this conclusion.

Also it is key to recall that CD34+ HFSCs grown in vitro also lose cd34 (except when grown as organoids in Matrigel) but maintain their SC features. Could it be that simply the CD34 mark is not normal but that it is not indicative?

If the authors are correct and it's a rare event what are the 90% Hfs seen? Is that just cells filling in the molds the maintain the semi eosopehsgal/HF signature? If so why not examine the effect of such a signature or the heterogeneity within the formed HFs in correlation with function/morphology/cycling/etc?

This point is a critical point of the manuscript and has to be made a lot clearer.

Question 4. Reviewer #2

We agree with the reviewer that, in retrospective, the point referring to the "efficiency" of the process was not properly conveyed and appears rather misleading. The statement referring to 90% efficiency (**Supplementary Fig. 1J**) indeed refers to re-epithelialisation or, as the reviewer refers to "cells filling the molds" of pre-existing HFs. We have now amended the text to make a clear distinction between the two statements. This has no connection to the lineage conversion process *per se*, and merely acknowledges that cells are able to efficiently repopulate the denuded dermis, hence this should not

represent a limitation for cells to receive instructive cues from the stroma. See amended text. The text reads: *“Re-epithelialisation of the denuded skin and HF niches by oesophageal cells in heterotypic cultures represented a highly efficient process, with $91.4 \pm 1.8\%$ of the original EGFP+ HFs being replaced by oesophageal-derived HFs-like structures (tdTomato; red) at day (D) 10 post-culture (Supplementary Fig. 1J).”* (p5; l20-23).

The poor efficiency of the lineage conversion process, instead, was initially based on CD34 expression. However, we completely take the reviewers point regarding the fact that CD34 expression can be significantly impacted by experimental conditions. Hence, to support our claim that the lineage conversion process is inefficient, we have now also included the quantification of KRT15 expression in oesophageal-derived HFs (Figs 2D, E). Regarding the expression of CD34 *in vitro*, we can only elaborate on the fact that in our 3D culture system cells are grown directly over alive stroma in order to more closely mimic *in vivo* conditions.

Regarding the heterogeneous competency of oeHFs to acquire certain HF markers, we believe this is covered, at least in part, by the epidermal marker characterisation (extended now in the new version of the manuscript; Fig 2 and Supplementary Fig. 2). Our data shows that certain epidermal markers such as KRT24 and KRT17, become widely expressed. While others such as CD34, KRT15, and KRT10 expression is only acquired by certain oeSKIN cells (Fig. 2D). We have now added a text remark for clarity (See amended text below). In relation to the characterisation of this heterogeneity, as shown by our lineage conversion trajectory analysis (scRNA-seq and scATAC-seq; Figs. 4 and 5; Supplementary Figs. 4 and 5) as well as mechanistic studies (Figs. 6 and 7; Supplementary Figs. 6 and 7), we associate this to the existence of barriers to lineage conversion. In particular, we define the HIF1a-SOX9 axis as a critical regulatory barrier. To further support his notion, we have now included new mechanistic data (Fig. 5O-R, Fig. 6 and Supplementary Fig. 6J-L). For further details, please refer to our answer to the next point (Question 5. Reviewer #2).

Amended text:

“Despite tangible evidence proving the ability of oesophageal cells undergo environmental reprogramming and switch towards skin identity oeHF, further inspection of the oeHF data revealed the limited nature of the fate conversion process. In particular, the expression of HF markers was highly variable across oeHFs appendages, ranging from widespread expression in the case of KRT17 and KRT24 (~78% and ~91% oeHFs expressing these markers, respectively), to only a reduced proportion of eHFs (~28%) expressing CD34 (Fig. 2D). The expression was similarly variable within individual oeHFs, with the number of cells expressing CD34 and KRT15 representing just a minor proportion when compared to native skin (Fig. 2E). These results unveiled the inefficiency of the OE-to-skin lineage conversion process.” (p7; l9-17).

5. Mechanistic Evidence for HIF1a–SOX9 Axis: The proposed HIF1a–SOX9 axis is underdeveloped. Pharmacological and genetic manipulation (Sox9 deletion) show shifts in CD34 expression. Still, additional immunofluorescence panels (e.g., double labeling for SOX9 and HIF1a) and a more direct demonstration that Sox9-null/HIF1a-inhibited cells adopt stable hair-follicle gene signatures is required.

Question 5. Reviewer #2

We apologise for the lack of depth in our mechanistic validation of the HIF1a-SOX9 axis. Please find below new experiments, included in the revised manuscript, further demonstrating the role of the HIF1a-SOX9 axis as a barrier in the lineage conversion process:

- a) **Hypoxic cues induce SOX9 expression to establish the lineage conversion barrier**– We have included additional experiments that add functional depth to the proposed mechanism. To directly test the interplay between hypoxic signals and SOX9 during the lineage conversion process, *Krt14^{CreERT};Sox9^{fl/fl}* derived heterotypic cultures were exposed to a hypoxic treatment (4h) to simultaneously block SOX9 expression and stimulate HIF1a (**Fig. 6G-I**). We anticipated that if HIF1a and SOX9 expression were acting as independent barriers blocking the lineage conversion process, the hypoxic treatment would rescue the *Sox9^{fl/fl}* phenotype, bringing HF markers back to control levels. However, this was not the case, the hypoxic treatment did not restore HF markers (**Fig. 6G-I**), indicating that, in this context, HIF1a is acting via SOX9.

In line with these results, we now also include experiments showing that SOX9 expression is indeed directly regulated by hypoxic signals. New data from hypoxic *ex vivo* oesophagi reveal an increase in SOX9 expression that can be rescued by treating them with the HIF1a inhibitor KC7F2 (**Fig. 6A,B**), indicating that SOX9 is indeed downstream HIF1a.

Overall, these new observations further demonstrate that the cell identify barrier is exerted via the HIF1a-SOX9 axis and that SOX9 is downstream HIF1a, as originally proposed.

- b) **SOX9 promotes HF identity in the skin**– Interestingly, and in line with the reviewer's comment regarding the discrepant role of SOX9 in the skin, new experiments on sSKIN (*in vitro* skin-derived from epidermis) reassuringly showed the opposite trend to oeSKIN (oesophageal derived skin); with *Sox9^{fl/fl}* epidermal cells expressing less CD34 and KRT15 than controls (**Supplementary Fig. 6J-L**). Accordingly, *in vivo* oeHFs derived from *Krt14^{CreERT};SOX9^{fl/fl}* hair reconstitution assays were not viable long-term, suggesting that once the HF identity is established, SOX9 expression is required for oeHF maintenance. These observations reveal that SOX9 is having the opposite effect in endogenous skin cells and oesophageal cells, and demonstrate the context specific role of SOX9. This is not an entirely surprising observation, given that SOX9 is a transcription factor that does not merely modulate HF morphogenesis and maintenance. SOX9 has been shown to be a pioneering factor, acting as a master regulator that governs cell fate during development, injury and cancer in multiple tissues³⁵⁻⁴¹, often showing context-dependent roles in epithelial biology.

All the results described above have now been included and are discussed in an extended revised version of the manuscript, our observations support a role of the HIF1a-SOX9 axis as a barrier to lineage conversion.

6. Imaging and presentation: Multiple figures lack clarity and proper zoom-in insets (e.g., Fig 1C). Stratification is mentioned, but not clearly shown or validated.

• Some immunostainings (e.g., Supplementary Fig. 1E) should be repeated with basal and proliferation markers to better characterize the tissue structure.

• Controls are inconsistently presented—e.g., peeled vs. unpeeled dermis, esophagus-only grafts, skin-only grafts. This comparison must be rigorously controlled and quantified.

Question 6. Reviewer #2

Thank you for this suggestion. Images showing stratification and zoomed in insets have been added to Figure 1 as requested. Apologies for missing to depict this feature in the original version of the manuscript.

Additionally, we have significantly reconfigured figures in order to enhance clarity across the revised version of the manuscript, including new controls. Please note that most antibodies used required peeling (removal of the underlying stroma) in order to achieve adequate penetration and staining of epithelial wholemounts. Indeed, wholemount immunostaining was a necessary technique to study

Response to reviewers' NCOMMS-25-11438

oesophageal-to-skin lineage conversion, given the limited number of fate switch events found in heterotypic cultures, which, in itself, represents part of the main message of the presented work.

Additional specific comments organized by Figure

Introduction:

Rationale for Esophagus–Skin Model: The manuscript would benefit from a more compelling explanation of why esophageal tissue was chosen to test epithelial fate switching in a skin-dermis environment. Although the esophagus is squamous and devoid of adnexal structures, it would be helpful to articulate how this setup advances the concept of adult stem cell plasticity or potential clinical applications.

General comment. Reviewer #2

We agree with the reviewer that highlighting the relevance of studying epithelial cells fate plasticity in the oesophagus is important, particularly given that columnar metaplasia (transdifferentiation) in the oesophagus has been associated with clinical complications, including oesophageal cancer^{9,20-23}. We have revised the manuscript to highlight the translational relevance of understanding epithelial cell fate plasticity in the context of the oesophagus. The revised text now reads:

“It has now become widely accepted that cell fate decisions are not as predetermined as originally thought. In response to injury or environmental perturbations, adult epithelial cells present a remarkable fate plasticity allowing them to respond to changes in tissue needs¹⁴⁻¹⁶. Yet, if dysregulated, this adaptability can promote epithelial disease and cancer^{15,17-19}. Particularly, in the context of the oesophagus, columnar metaplasia has been associated with the development of oesophageal cancer^{9,20-23}. Hence, understanding the largely unclear mechanisms balancing cell fate plasticity is critical to unveil its regenerative potential while preventing inherent tumorigenic risks²³.” (p18; l2-8).

Figure 1

- Include additional details on culture duration and viability, how long the peeled dermis and esophageal strips remain viable in the 3D culture?*

Heterotypic culture details. Reviewer #2

We thank the reviewer for pointing this out. We have now added additional technical details about our heterotypic culture approach in the methods section (please refer to **3D Oesophageal-Dermal Heterotypic and control cultures** section of Methods). For the reviewer's reference the maximum time we have been able to grow cells using tissue recombination assays under the presented experimental conditions has been 15 days, as characterised in (Doupe & Alcolea et al. Science 2012; **Supplementary Fig 9** in Science paper)¹³. Beyond that time the tissue constructs start shrinking and altering the architecture.

- Panel 1C could benefit from zoomed-in insets (e.g., progressive days of re-epithelialization) to visualize early stages of esophageal cell expansion over the dermis.*
- 1C-need clearer zoomed in images of the process.*

Re-epithelialisation. Reviewer #2

As suggested by the reviewer, new **Fig. 1E** includes zoomed in images to visualise the re-epithelialisation process.

- What stages are required for the formation of the hair follicles arise from esophageal tissue? Author shows the appearance of the follicles without clear indication and discussion on how they are formed.*

Not clear if mCherry cells only fill out empty areas. Are the follicles in specific stage? What are the stages of hair formation and how are they similar to embryonic formation of hair follicle?

oeHF formation. Reviewer #2

We agree that this point needs to be reinforced to avoid confusion. Our data suggests that the steps of oeHF formation, in heterotypic cultures, involves:

- i) The re-epithelialisation of the denuded skin dermis by oesophageal cells. Here oesophageal cells acquire a HF-like structure that resembles the stage of the HF that was originally occupying a given niche.
- ii) This process exposes oesophageal cells to the ectopic HF niche.

The instructive niche signals enable the lineage conversion process in competent oesophageal cells.

We have shown that this depends on the HIF1a-SOX9 axis, which is now further supported with new functional data (**Fig. 6**).

We have amended the original text to clarify the steps of oeHF formation. However, it is worth noting the fact that oesophageal cells undergo lineage conversion *in vivo* (in grafting experiments), where oesophageal cells receive instructive signals of developing skin without the “architectural footprint” of existing HFs in the dermis. This suggests that access to instructive niche signals represents the key step to initiating the lineage conversion process. The text now reads:

*“Interestingly, adult OE cells (red) were able to fully re-epithelialise the bare dermis and empty skin HF sockets, creating a de novo stratified epithelium (**Fig. 1E**) and forming structures reminiscent of *in vivo* HFs, hereafter referred as oesophageal-derived HFs (oeHFs; **Figs. 1C-F**).”* (p5; I17-20).

*“Overall, our data shows that the oesophageal-to-skin lineage conversion process involves series of events. These includes i) the re-epithelialisation of the denuded skin dermis by oesophageal cells. Here oesophageal cells repopulate the empty epidermal stroma forming HF-like structures as they fill the empty HF sockets. ii) This process exposes oesophageal cells to the ectopic HF niche. iii) The instructive niche signals enable the lineage conversion process in competent oesophageal cells in a HIF1a-SOX9 dependent manner. We go on to demonstrate that blocking HIF1a or its downstream target SOX9 lifts the regenerative blockage and favours cell fate plasticity by promoting the transition from oesophageal to skin lineage both *in vivo* and *in vitro* (**Proposed model Fig. 7I**), and that this may represent a more universal mechanism operating beyond oesophageal epithelium.”*(p17; I14-22).

- *Confirm the endogenous staining in figure 1c with epidermal markers, basal layer markers, to confirm that mCherry cells indeed convert to epidermal cells.*

Epithelial origin of oeHFs. Reviewer #2

We are grateful to the reviewer for this suggestion, which greatly enhances the clarity of the skin the heterotypic re-epithelialisation assay. **Supplementary Fig. 2** already contained a panel showing Krt14-CD34-PDGFRa staining to demonstrate the epithelial origin of the oeHF CD34 cells (**Supplementary Fig. 2 E**). For further transparency, we have now also included a panel in **Supplementary Fig. 1F** to show that tdTomato oesophageal-derived skin expresses P63 in the basal progenitor proliferative compartment.

- *Supplementary figure 1E- cp3 and ki67—need to show the staining in the tissue couples with basal layer marker*

Caspase 3. Reviewer #2

Response to reviewers' NCOMMS-25-11438

In line with the point above, we now include in **Supplementary Fig. 1F** a panel to show that basal proliferation (Ki67 staining) coincides with basal marker p63 expression. We additionally add representative active Caspase 3 images **Supplementary Figs. 1G-H**.

• Need farther verification to confirm epidermal tissues, staining with different markers and zoom in images to show stratification, is there epidermal turnover? Is it functional? The author mentions the formation of squamous stratified epithelium but there are no evidence for it.

Stratification and proliferation. Reviewer #2

In line with the reviewer's suggestion, new **Fig. 1E and Supplementary Fig. 1F** include zoomed in images to show the stratification of oesophageal-derived skin (**oeSKIN**). New data includes basal squamous epithelial marker p63, as well as Ki67 to reflect the active proliferation of basal cells in the newly formed squamous tissue.

• Image 1F is not enough to show pigmentation. Perform fontana mason analysis in order to indicate pigmentation. Are there melanocyte stem cells in the follicles?

Pigmentation. Reviewer #2

In relation to **oeHF** pigmentation, the newly added Fontana Mason staining shows that the denuded dermis retains melanin after peeling, which may be taken up by oesophageal keratinocytes during **oeHF** formation (**Supplementary Fig. 1L**). We further support this notion by showing how pigmented structures in **Fig. 1G and Supplementary Fig. 1K** are indeed of oesophageal origin (i.e. express tdTomato reporter).

• Are the structures that appear to be the sebaceous glands, actually the sebaceous glands? Perform oil staining and zoomed in images of the cells.

Oil red. Reviewer #2

As requested by the reviewer, we can confirm **oeHFs** contain cells expressing the sebaceous gland marker stearoyl-CoA desaturase 1 (SCD1; **Fig. 2B and Supplementary Fig. 2A**), and can accumulate lipid droplets (as shown by Oil Red O staining in **Supplementary Fig. 1M**).

• Is that the hair shaft in 1F middle and right HF?

Hair shaft. Reviewer #2

Indeed, the stratified cells inside the **oeHFs** create what resembles a hair shaft-like structure (**Supplementary Fig. 7D**). However, since we are not able to keep heterotypic cultures for more than 15 days, we have been unable to follow hair cycling long-term. Notwithstanding this, we do show that epithelial cells in **oeHF** are contributing to rudimentary **oeHF** shafts in *in vivo* skin hair reconstitution assays (**Fig. 7**).

Figure 2

• Use additional markers for hair-follicle identity including K15

Additional markers 1. Reviewer #2

In response to the reviewer's request, we have stained **oeSKIN** wholemounts for additional markers supporting the conversion from oesophageal to skin lineage (LHX2, KRT15, KRT10, SCD1 and SCA-1; **Fig. 2 and Supplementary Fig. 2A-E**). Additionally, Oil Red O has been used to mark lipid accumulation in oesophageal cells (akin of sebaceous glands; **Supplementary Fig. 1M**), and Fontana Masson as a

Response to reviewers' NCOMMS-25-11438

proxy to study melanin incorporation in **oeHFs** (**Supplementary Fig. 1L**). These data included in the revised version of the manuscript further supports the lineage conversion notion.

- *Use conventional epithelial markers (K14 for basal cells, K10 or others for differentiation) would confirm that the newly formed structures exhibit proper stratification rather than partial epithelial overgrowth.*

Additional markers 2. Reviewer #2

To address this point additional images showing basal and differentiation markers KRT14, p63, KRT10 have been added to **Supplementary Fig. 1F and Supplementary Fig. 2A-E**.

- *In figure supplementary figure 1C, positive staining of sox 2 indicate the presence of the dermal papilla in the peeled skin, is that true for every area where hair formed? Is the presence of the DP required for hair formation in this case? Is there any data regarding the specific fibroblasts populations contributing?*

Dermal Papillae. Reviewer #2

Apologies for the misunderstanding, we can confirm that in denuded (peeled) dermis there is widespread retention of dermal papillae (DP) of skin origin (EGFP positive; **Supplementary Fig. 1C**).

At the culture end-point, **oeSKIN** wholemounts were peeled from the dermis, in order to enable good antibody penetration and 3D confocal reconstruction of the tissue. Hence, the dermal papilla was not part of our systematic analytical pipeline. Despite this, the DP was found in a number of **oeHFs** at the end-point cultures even after peeling (**Supplementary Fig. 2I, J**), supporting its potential role in the lineage conversion process. Additionally, our lineage tracing data highlights the importance of the skin niche in the lineage conversion process (**Fig. 3**). The text has been amended to clarify this point:

*...“The peeled dermis consistently retained the architectural footprint of its appendages and associated stromal niches (**Supplementary Figs. 1B, C; see methods**)”...(p4; l31-32).*

*...“In line with this, dermal papillae derived from the host skin (EGFP) were found in heterotypic cultures (**Supplementary Fig. 2I,J**). This indicates that the instructive signals promoting lineage conversion originate in the dermal niche, reminiscent of normal skin physiology^{42,43}.”... (p6-7; l32-3).*

- *You mention low efficiency of hair formation, yet in images such as supp 1 D we see hair formation via esophageal tissue that is almost all over the tissue.*

oeHF efficiency. Reviewer #2

This point has already been addressed in our response to **Question 4** of this reviewer. In short, we acknowledged the lack of clarity on this point. We have now explained in the text the difference between re-epithelialisation efficiency (**Supplementary Fig. 1J**) which denotes the high efficiency at which the oesophagus re-epithelialised the denuded dermis, versus the low efficiency of the lineage conversion process (which involves HF marker expression and architectural features; **Fig. 2D, E**).

Figure 3

- *Please also show GFP channel in all the images in order to see the cells originated from skin stromal origin.*

EGFP. Reviewer #2

Our apologies for the lack of clarity. Please note that epithelial wholemounts were peeled before imaging to enable better antibody penetration, facilitate visualisation, and 3D reconstruction of the

tissues. This is particularly needed for this study due to limited oesophageal-to-skin lineage conversion. Due to the peeling process, EGFP+ signal from the host epidermis can only be visualised in samples that have been sectioned (where tissue composites retain EGFP-derived dermis **Fig. 1C,E and Supplementary Figs. 1F, 2D and 7E-I**). As already mentioned, all our images show all the channels indicated in the associated image caption. This is in all cases without exception. Hence, if a colour is not present, this means that given labelling/reporter is not there or undetectable; in the case of dermal EGFP due to the peeling process. In summary, the reason why EGFP is not detected in **oeSKIN** samples is because there was little to no contribution of skin cells (EGFP+) to the newly formed **oeSKIN** once the dermis is peeled away from stroma for wholemount purposes.

- *The conclusion that based on cd34 marker only the regeneration depends on dermal niche is insufficient. Need to check in other stem cell populations and cell types.*

Niche. Reviewer #2

The lineage tracing experiments (**Fig. 3**) investigate the cell dynamics involved in dermal re-epithelialisation and **oeHF** formation, revealing the polyclonal nature of the process. The profile of CD34 expression originally suggested that niche signal instructed the **oeHF** identity switch. New data including KRT15 expression, as an additional HF marker, now shows a much less apparent compartmentalisation. Hence, we have decided to underemphasise the potential relevance of specific anatomical cues within the HF dermal niche. We have amended the text referring to stromal signals in general:

*“Additional heterotypic experiments, where the adult OE was grown over the dermis of developing skin, one day post birth (P1), showed the specific upregulation of the marker LHX2, known to label developing HFs in vivo (**Fig. 2C**), in **oeHFs**. This observation revealed the critical role that the stroma plays in the identity switch, being able to rewire the **oeHF** fate towards developmental stages. In line with this, dermal papillae derived from the host skin (EGFP) were found in heterotypic cultures (**Supplementary Fig. 2I,J**). This indicates that the instructive signals promoting lineage conversion originate in the dermal niche, reminiscent of normal skin physiology^{42,43}.” (p6-7; l29-32; l1-3).*

- *The author shows polyclonal contribution for the esophageal cells make up the hair follicle, yet it will be valuable to add if different clones give rise to different populations and their actual contribution to hair follicle formation*

Lineage tracing. Reviewer #2

We apologise for the lack of clarity. We have now amended the text providing a more detailed account of the data.

Given that clonal induction is under the control of the ubiquitous Rosa26 promoter (R26CreERT), the contribution of labelled clones would be expected to be the same compared to other unlabelled clones present in the tissue, and by no means is selective of specific cell populations/cell states that may emerge during the re-epithelialisation process. In that vein, we observe that the emerging **oeHFs** are of polyclonal nature along the entire structure, with no clonal preferential enrichment or expansion in any specific areas. Given that these experiments were done under minimal (clonal) recombination, the data therefore argues against a specific cell subpopulation being responsible for **oeHF** formation. Please see newly generated images for reference (**Fig. 3E and Supplementary Fig. 3A-C**). Please refer

to main text for further clarity on this point: *“Starting at an initial clonal labelling covering just ~5% of the basal layer, fluorescent OE cells expanded and re-epithelialised the dermis colonizing in their way two thirds of all the oeHF structures present in the tissues ($66.37\% \pm 6.07$; mean \pm SD). As a result, upon clonal expansion, approximately 66% of all oeHFs formed contained at least one or more labelled cells, revealing that, in line with previous observations in oesophageal wound healing¹³, OE cells undergo a widespread recruitment during oeHF formation.”*(p8; l1-6).

Figure 6

- The text states there is a SOX9–HIF1a Co-localization : *“The data showed a positive correlation between SOX9 and HIF1a levels; with HIF1a expressing cells being more likely to be positive for SOX9, as measured by immunostaining (Supplementary Fig. 6C)”* , but panels explicitly showing co-staining are not shown.
- Need proof of concept for sox9 floxed model, are the knocked out cells the ones express HIF1a?

Figure 7

SOX9-HIF1a axis. Reviewer #2

We appreciate the reviewer's comment. As they will be able to infer from **Supplementary Fig. 5K** and as well-established, the HIF1a staining profile shows a diffuse staining pattern (punctate staining). Hence, we believe that a single image would not be representative of the data and would not be able to convey the message (see example below **Response Fig. 4**). For quantification purposes, we made use of an unbiased automated nuclear segmentation pipeline (Arivis 3.5.1. segmentation), which enabled us to quantify both HIF1a and SOX9 staining in oeSKIN. This is the data shown in **Supplementary Fig. 6D**.

Response Figure 4: oeSKIN immunostained for HIF1 α and SOX9. The image shows a representative HIF1 α punctate staining pattern, illustrating the need for quantification across multiple oeHF units to accurately reflect the expression profile.

- The paper mentions a low incidence of complete HF formation in vivo. A bar chart or table quantifying “follicles per graft” across several mice would help interpret these data.
- The image in 7G does not seem to be representative to the data mentioned in the text regarding number of hair follicles per grafts.

In vivo grafts. Reviewer #2

As per reviewer's request a graph showing the number of oeHFs per in vivo graft assay has been included in **Supplementary Fig. 7J**.

- *The low frequency of the HF formation event raises questions regarding the validity and the relevance of the mechanism*

oeHF inefficiency. Reviewer #2

We would like to note that our intention in the presented study is not to prove that oesophageal cells can be reprogrammed into fully functional HF structures using our environmental reprogramming approach. The novelty of this study centres around identifying the mechanisms that naturally protect our tissues from inadequate cell fate plasticity. We believe that identifying those will offer a benchmark to develop strategies to improve tissue regeneration and potentially prevent fate promiscuous behaviour in certain pathologies, such as cancer. We have noted this in the text:

...“Overall, these data demonstrate that HIF1a and its downstream effector SOX9 establish a regenerative barrier that prevents cell plasticity during tissue repair in oesophageal cells.” (p16; l6-8).

The revised manuscript also centres the narrative around barriers to lineage conversion rather than the lineage conversion process itself. Using robust cellular and molecular approaches, we have thoroughly established the existence of barriers protecting the identity of epithelial cells both *in vivo* as well as *ex vivo*. And we have gone one step further, unveiling the role of the HIF1a-SOX9 axis in the process, not only in the oesophagus, but new data also show that this barrier may be operating in other epithelial tissues such that of the trachea (**Supplementary Fig. 7K-M**). We believe that this work contributes to our understanding of what mechanisms limit adult cell fate plasticity preventing an aberrant and uncontrolled response that could lead to abnormal growth and tumorigenesis upon tissue perturbations. Our data also opens exciting new avenues to decipher the inner workings dictating cell fate plasticity, providing a benchmark to identify the principles underlying the full regenerative capacity of epithelial cells.

Reviewer #3 (Remarks to the Author):

Bejar et al. dissect the molecular mechanism of fate permissibility in adult cells using a 3D regenerative culture system where TdT-marked esophageal tissue is placed on epithelium-less GFP skin dermis and the esophageal epithelial (EE) cells crawl out and on to the skin dermal support. They nicely show that esophageal to skin lineage conversion of the migrating EE is polyclonal and inefficient (using CD34+) showing mixed EE and skin epithelial (SE) markers and that conversion is dermal niche cell-dependent, as removal of the LGR5-GFP dermal cells prevents CD34+ conversion in TdT+ migratory cells. scRNA seq of in vitro transitioning TdT cells indicates reactivation of the embryologically expressed HIF1a and Sox9 (Fig. S6F) among other gene expression changes. Addition of HIF1a small molecular inducers/inhibitors or K14Cre-ERT2 conditional Sox9 mutants cells allows more efficient conversion within the denuded skin microenvironment, providing evidence for an epigenetic block for fate conversion. Overall the methodology follows the lab's previous published in vitro wound healing model and standard chamber regeneration methods.

The work follows pioneering work from the Barrandon lab and others (refs. 25-31) showing p63+ non-skin epithelium retains plasticity to convert to skin in the presence of skin dermal cues, and that of the Fuchs lab showing that Sox9 can reprogram skin IFE to HF. While the present work presents an interesting observation, it is still preliminary and not at the level for this journal. The concept of plasticity is not new and the authors focus only on unidirectional plasticity of one tissue in the present work. The conversion requires additional characterization (see below) to distinguish loss of esophageal markers as well as gain of hair follicle vs. interfollicular epithelial fates. The present work provides a new mechanistic insight by showing HIF1a/SOX9 prevent efficient conversion in this system, but need

to demonstrate generalizability and reversibility to enhance the overall impact of the work. In addition previous work clearly shows the role for Sox9 as a pioneer factor for hair follicle lineage (Yang PMID: 37488435 ; Kadaja PMID: 24532713; Pyczek PMID: 31867038). The present work showing SOX9 removal enhances contribution to skin HF lineage is quite surprising and contradicts much previous work and so additional controls and experimental data is required to clarify this issue prior to publication.

Reviewer #3 - Author response: We are grateful for the reviewer's suggestions and insightful feedback, which certainly enhances the presentation of our manuscript. We are also pleased that the reviewer considered that our study *"presents an interesting observation"* and that *"the present work provides a new mechanistic insight"*. We have revisited the manuscript to include changes that address the points raised below.

Major Issues:

1. Generalizability of the work. Barrandon showed cornea, bladder, thymus niches are instructive so would help to confirm this as well in other K14+; p63+ tissues, not just esophagus. Using the authors wound migration assay they could implant one or more of these other tissues using the same Td-lineage tracing mouse allele and demonstrate the universality of the HIF1a/SOX9 universal block using DMOG/KC7F2/Sox9 fl/fl.

Question 1. Reviewer #3

We agree with the reviewer on the importance of showing the relevance of the mechanism described in our study beyond oesophageal-to-skin conversion. The revised manuscript now includes new data where tracheal epithelium is grown over the denuded skin dermis to investigate the mechanisms governing the lineage conversion process in a different epithelial tissue. In line with previous work, and as rightly indicated by the reviewer, our new data show that denuded skin dermis re-epithelialised by tracheal epithelium produced a squamous-like epithelium and HF-like structures expressing the HF marker CD34 and KRT15 (**Supplementary Fig. 7K-M**). In line with our observations in the oesophageal epithelium, the expression of HF markers in tracheal-derived HFs was significantly increased when knocking down SOX9 (in *Krt14^{CreERT};SOX9^{fl/fl}* mice), supporting the role of SOX9 in lineage plasticity across different tissues.

Interestingly, beyond previous work on SOX9 as a critical pioneering factor modulating cell fate plasticity^{9,44,45}, a new study has now revealed that SOX9 silencing favours tissue regeneration in a kidney injury model⁴⁶. This highlights the importance of SOX9 as a molecular switch determining the regenerative potential of epithelial cells. Our study provides a new perspective, unveiling the key role of SOX9 in modulating cell fate plasticity beyond physiological constraints.

2. Reversibility. Similarly, Barrandon elegantly showed that plasticity is reversable (Bonfanti PMID: 20725041) and the authors should show whether skin to EE transdifferentiation occurs and is stromal dependent. Again this could be done using the same assay performed in reverse with skin epidermal cells migrating onto esophageal mucosa.

Question 2. Reviewer #3

We appreciate the reviewer's comment in relation to reversibility. We can confirm that we have indeed performed these experiments. However, in our hands, skin did not show signs of skin-to-oesophagus lineage conversion using our model, it is unclear whether this is related to the instructive nature of the

oesophageal stroma or the plastic capacity of the skin under these experimental conditions. Please note that new experiments with tracheal epithelium also show evidence of lineage conversion; hence, in line with previous work³⁰, our results do not appear to represent an isolated finding.

We have included representative images of a new experiment where skin was used to re-epithelialise the denuded oesophageal stroma (submucosa; **Response Fig. 5**). The data show how in this setting the skin (EGFP+) retains the expression of the epidermal differentiation marker KRT10, and does not appear to acquire the expression of oesophageal markers such as SOX2 (oesophageal progenitor cells) and KRT4 (oesophageal differentiation marker). Since we do not have a definitive explanation for this apparent difference, we have opted to leave this data out. However, if the reviewer considers this appropriate, we would be happy to include this in the manuscript as an incidental observation of potential interest.

Response Figure 5: Representative image shows a xyz plane view of skin epithelium (green) grown *in vitro* on top of denuded oesophageal stroma for 10 days. Note the epidermal cells (green) re-epithelialise the oesophageal stroma and achieve several layers of stratification. They retain KRT10 expression in suprabasal differentiated cells, but do not acquire oesophageal progenitor (SOX2) nor differentiation (KRT4) markers. Scale bar 50µm.

3. Stromal Dependence. The authors show dermal dependence in Fig. 3B-F. However a critical control is required and the authors should show that replacing the skin substrate with esophageal dermal cells maintains esophageal fate and prevent skin transdifferentiation, and whether prostate, bladder corneal stromal cells regulate transdifferentiation to the cognate epithelial state rather than skin.

Question 3. Reviewer #3

We thank the reviewer for pointing this out. We have now expanded our HF marker panel in oesophageal-derived samples (experimental conditions, **oeSKIN**) as well as in our control samples, i.e. oesophageal epithelium grown *ex vivo* over its own oesophageal stroma (**oeOE**). The new data show that, unlike control homotypic stroma, oesophageal tissue exposed to the signals of the skin dermis acquires a robust expression of skin markers (**Fig. 2 and Supplementary Fig. 2A-J**). These results demonstrate that the lineage conversion process is triggered by the stromal identity, and the observed identity switch is not the results of an experimental artefact.

Additionally, we have included new experiments where the denuded stroma of bladder and tongue is re-epithelialised by oesophageal cells, following our standard *ex vivo* heterotypic assay protocol. These confirm that exposure to stroma of other epithelial tissues, i.e. of ectopic nature, triggers a plastic state permissive to cell fate rewiring, as shown by KRT8 upregulation (**Supplementary Fig. 2K-M**). Overall,

Response to reviewers' NCOMMS-25-11438

our data show that epithelial cells can rewire their fate, expressing ectopic markers, in response to instructive stromal cues from different tissue origins.

4. Role of Pioneer Sox 9 in HF lineage. Previous work mentioned above indicates Sox9 is required for HF lineage commitment from skin IFE. The authors show that wound EE reactivates embryonic Sox9 expression and K14 CreERT2 Sox9 fl/fl enhances rather than blocks transdifferentiation into the HF compartment. This is surprising and an apparent contradiction from previous work. In particular the quantitation of Sox 9 KO CD34+ in Fig 6B-E looks similar to that in the control and quantitation from Fig 2B, arguing against a true difference. The authors should perform scRNA seq on Sox9 fl/fl TdT eHF (or in Fig. 7H) and compare it to published datasets. For example SOX9 is a major regulator of the hedgehog pathway critical for HF growth and the so if HF growth is occurring via hedgehog /WNT signaling then the results would suggest an alternative HF pioneer pathway distinct from skin IFE to skin HF. This would be of interest to the broader audience.

Question 4. Reviewer #3

We thank the reviewer for raising this important point. Indeed, SOX9 is a well-established regulator of HF morphogenesis and maintenance, with essential roles in HF stem cell specification during development⁴⁷. However, SOX9 is not merely a transcription factor modulating HF formation, SOX9 has been shown to be a pioneering factor, acting as a master regulator that governs cell fate during development in multiple tissues³⁵⁻⁴¹. SOX9 various roles rely on dynamic spatiotemporal regulation and interaction with tissue-specific cofactors^{18,44,45,48}.

Given the complex regulatory nature of SOX9, it is not surprising that, when its dynamic expression is disrupted by sustained SOX9 overexpression within an ectopic environment, as in the case of our heterotypic regenerative model system (**Supplementary Fig 6G, H**), the role of SOX9 may differ from its physiological function in HFs. This mirrors previous studies in adult skin where persistent SOX9 expression (e.g., via SHH activation or direct overexpression)^{45,49} initiates HF-like programs but fails to complete HF lineage specification, including CD34+ HFSC formation, ultimately leading to tumorigenesis rather than faithful HF morphogenesis.

Our data support a similar phenomenon: oesophageal SOX9 expressing cells initiate aspects of the HF program but acts as a barrier to full oesophageal-to-HF conversion. Notably, we show that SOX9 is regulated by HIF1a in our system (not WNT or SHH, as in native skin), further highlighting the distinct regulatory landscape.

To strengthen our conclusion, we now provide new data:

- **SOX9 is modulated by hypoxia/HIF1a:** SOX9 levels increase in hypoxia and decrease with HIF1a inhibition; both effects are reversible (**Fig. 6A, B**).
- **SOX9 deletion upregulates HF marker expression in oeHFs:** *Krt14^{CreERT};Sox9^{fl/fl}* cultures show increased CD34 and KRT15 expression, confirming SOX9 suppresses HF identity in the oesophageal context (**Fig. 6C-F**). Here, we further support our conclusions by complementing observations on CD34 expression with KRT15, as an additional marker enriched in HFs.
- **Context specificity:** We have also included a direct comparison between oesophagus and skin. In contrast to oesophagus, *Sox9^{fl/fl}* skin-derived cultures show reduced HF markers, consistent with SOX9 having pro-HF role in native skin (**Figs. 6G-I and Supplementary Fig. 6J-L**). Moreover,

hypoxia suppresses HF markers in oeHFs but not in skin-derived cultures, reinforcing the tissue-specificity of the HIF1a–SOX9 axis.

Additionally, to test whether our mechanistic insights apply beyond the oesophageal epithelium, we have now included heterotypic cultures exposing tracheal epithelium to denuded dermis (**Supplementary Fig. 7K-M**). Our data shows that knocking down *Sox9* in tracheal epithelium also promotes lineage conversion, as shown by the increased expression of HF markers CD34 and KRT15 (**Supplementary Fig. 7K-M**). These results suggest that the HIF1a-SOX9 axis may represent a cell plasticity barrier operating across epithelial tissues.

- **New *Krt14^{CreERT};Sox9^{fl/fl}* single-cell RNA sequencing:** As part of a follow up project in my laboratory, we have performed preliminary scRNA-seq experiment showing differential gene expression signature across *Sox9* high, low and negative cells (as per transcript counts; negative= no transcript counts) in heterotypic cultures derived from induced *Krt14^{CreERT};Sox9^{fl/fl}* oesophagi (*Sox9* cKO; **Response Fig. 6**). According to our observations, *Sox9* cKO cells presented an increased plasticity and permissibility to response to signals instructing alternative fates (**Fig. 6**). Consistent with this, the new scRNA-seq analysis shows that *Sox9* negative cells are enriched in genes related to epidermal and skin development, as well as cell differentiation, including upregulation of canonical epidermal genes. We acknowledge the limitation that, within *Sox9* negative cells, there may be a mixture of true *Sox9* knockout cells and cells where *Sox9* expression was not detected due to technical dropouts. We are currently working towards identifying markers to better identify the *Sox9* high versus *Sox9* low/negative cells in order to be able to dissect this dataset in greater depth and explore the relevance of the SOX9 axis for epithelial plasticity and tissue regeneration.

Response to reviewers' NCOMMS-25-11438

Response Figure 6: (A) Heatmap showing expression of differentially expressed genes between Sox9-defined basal cell groups: Sox9 no counts (negative), low, and high expression. (B) Selected GO terms and representative genes for up- and down-regulated modules in Sox9 no counts basal cells. This analysis integrates the original dataset presented in the manuscript (Figs. 4 and 5, and Supplementary Figs. 4 and 5) with newly generated single-cell RNA-seq data from oeOE and oeSKIN cultures derived from *Krt14^{CreERT};Sox9^{fl/fl}* oesophagi, (including Sox9 knockout cells).

Cells were grouped by Sox9 expression into no counts (no detected transcripts), low (expression > 0 to ≤ 1), and high (expression > 1). Differential expression analysis (logFC > 0.5, adjusted P < 0.05) identified 1,274 genes. Average expression per group was calculated using Seurat's AggregateExpression function, followed by normalisation and scaling.

Response to reviewers' NCOMMS-25-11438

The heatmap was generated with the ComplexHeatmap package, using hierarchical clustering and k-means row splitting (row_km = 10) to define gene modules. GO enrichment analysis was performed per module using enrichGO (clusterProfiler) with FDR correction ($P < 0.05$). Top GO terms are shown as bar plots in (B), along with selected representative genes.

Given our newly generated data and what is known about HF development, one would anticipate that if the cells were to execute the full lineage conversion programme, SOX9 expression at the adequate physiological levels would be indeed required to support HFSC identity and HF maintenance. However, our observations and those of others evidence a much more complex scenario, where SOX9 expression levels may determine distinctive (context and tissue dependent) cell fate programs.

The intimate relationship between the molecular regulators defining cell fate plasticity remains largely unknown and requires further attention. The work presented in our manuscript starts to shed light onto the complex processes modulating cell fate plasticity, revealing the intricate nature of SOX9. Under physiological conditions, SOX9 acts as a pioneering factor governing cell fate during development^{47,50}, but when overexpressed during tissue regeneration it instead acts as a barrier that blocks the ability of cells to respond to alternative cell lineages and protects cell identity during tissue remodelling.

Minor Critiques:

5. It would be interesting to perform immunofluorescence for some key esophageal and skin transcription factors, like Sox2, Pax9, Lhx2. Whether the CD34+ cells exhibit reduced levels of esophageal transcription factors and elevated skin transcription factors, compared to the CD34- cells or interfollicular cells?

Minor point 5. Reviewer #3

We recognise the reviewer's concerns regarding HF markers. To address this point, we have now added a deeper characterisation of oesophageal derived hair follicles (**oeHFs**). New data includes:

- HF Markers- Staining of **oeSKIN** wholemounts for additional markers supporting the conversion from oesophageal to skin lineage (LHX2, KRT15, KRT10, SCD1 and SCA-1; **Fig. 2 and Supplementary Figs. 2A-E**).
- Oil Red O has been used to mark lipid accumulation in oesophageal cells (akin of sebaceous glands; **Supplementary Fig. 1M**).
- Fontana Masson as a proxy to study melanin incorporation in **oeHFs** (**Supplementary Fig. 1L**).
- Regarding the request of more OE markers, we would like to highlight that the repertoire of markers truly specific to the oesophagus—and not shared with skin—is limited. Among the best characterised are KRT4 and SOX2, for which we now provide a more detailed analysis of their downregulation in **oeSKIN** (**Supplementary Figs. 2F-H**). To further strengthen this point, we leveraged our scRNA-seq dataset to identify differentially expressed genes (DEGs) between native oesophagus and skin, and used these to illustrate the cell fate transition. As expected, genes enriched in **oeOE** compared to **sSKIN** show reduced expression in oesophageal-derived skin (**oeSKIN**), consistent with a progressive loss of oesophageal identity (**Figs. 4L-N**).

Overall, these data further the support lineage conversion notion.

6. In Figure 4H, what are the cells in grey color? From Figure 4G, these cells are annotated as differentiated cells. Which tissue origin are they from?

Minor point 6. Reviewer #3

Apologies for the confusion. **Fig. 4H** displays only the cells annotated as basal, which are the focus of our analysis. The origin of all cells across the UMAP is provided in **Supplementary Fig. 4A**. As shown there, the differentiated cell population includes contributions from all sample conditions. Please note that all cell types described in **Fig. 4G**, not only differentiated cells, contain cells from all experimental conditions, except for sSKIN cells, which were only spiked in samples from heterotypic cultures, i.e. oesophagus grown over denuded skin dermis, as a skin reference control (see **Fig. 4A-H and Supplementary Fig. 3I-M**).

7. In Figure 4I and 4J, the authors first generated transcriptional signatures for D3, eSkin-D10, and eEE-D10 cells, and then checked how many genes were overlapped in a pairwise way. Can the authors compare the similarity among these three types of cells based on whole transcriptome using PCA or Pearson's correlation analysis?

Minor point 7. Reviewer #3

We thank the reviewer for pointing this out. In current **Fig. 4I**, we perform an unbiased differential gene expression analysis across conditions in enriched clusters, with not prior signature generation. We apologise for the confusion, and, in retrospective, regret the choice of the word "signature" in this context as this would not be a signature *per se*. This merely represents an overall differential gene expression analysis across conditions. The text and figures have been amended to clarify this point.

8. In the scRNA-seq, can the authors distinguish hair follicle cells versus interfollicular cells in the eSkin samples? From the data presented here, these cells should be quite distinguishable.

Minor point 8. Reviewer #3

We appreciate the reviewer's point. Unfortunately, we have been unable to distinguish HF and interfollicular epidermis in our scRNA-seq data, neither in skin-derived HFs nor in oesophageal-derived HFs *in vitro*. However, we have not been surprised by this observation given that HFs recreated *in vitro* do not show the full HF morphology and marker localisation shown by their *in vivo* counterparts. A well-known challenge associated with *ex vivo* 3D epidermal cultures^{29,30}. To further support the notion that this is a feature associated with *ex vivo* conditions, we have now included additional controls (skin-derived *in vitro* HFs (sHF)) in relevant figures. These show that the unusual morphology and ectopic marker expression are found in *in vitro* reconstituted HFs regardless of whether they are derived from skin or oesophagus. We have alluded to this important point in the revised manuscript:

"Of note, oeHFs showed atypical morphology and ectopic expression, with markers localised outside their anatomical position within in vivo skin HFs. This, however, was observed to be a recurrent feature of ex vivo reconstituted HFs regardless of their skin or oesophageal origin (Compare sSKIN and oeSKIN versus Skin in Fig. 2B and Supplementary Fig. 2A) in line with previous reports^{26,27,31-33}. Reassuringly, oesophageal control samples in vivo (OE) and in vitro (oeOE) confirmed the lineage specificity of skin markers (Supplementary Figs. 2B,C)³⁴." (p6; l15-20).

9. The authors used % CD34+ cells per eHF and % CD34+ HF as index for fate conversion, and found that pharmaceutical inhibiting Hif1a or knockout of Sox9 increased CD34+ percentage. How about cells in the IFE, do they exhibit more conversion towards skin IFE fate?

Minor point 9. Reviewer #3

To address the reviewers' suggestions, the revised manuscript includes additional markers to support the lineage conversion process. Among those we have included skin appendages related markers (such as LHX2, KRT15, KRT10, PCAD and SCD1; **Fig. 2 and Supplementary Figs. 2A-E**), as well as the IFE marker (SCA-1; **Fig. 2B and Supplementary Figs. 2A**). Indeed, these markers showed an increased expression in **oeSKIN** and/or specific re-localisation to **oeSKIN** areas when compared to oesophageal epithelium. New experiments also demonstrate that *SOX9^{fl/fl}* **oeHF**s show an increased expression of not only CD34, but also KRT15, as an additional marker enriched in HF's (**Fig. 6**). Instead, hypoxia, which induces SOX9, reduced the expression of both markers, CD34 and KRT15 (**Fig. 6G-I**). These results validate our conclusions beyond CD34 expression.

Regarding IFE, the IFE marker SCA-1, unlike other epidermal markers, showed a widespread expression in the oesophageal-derived skin (**Fig. 2B and Supplementary Figs. 2A**), and hence we did not see the need to investigate its potential enrichment upon HIF1a-SOX9 targeting. Please note that most markers overlap in both, oesophagus and skin, and hence the number of markers available to determine lineage conversion represents a limitation.

Minor edits:

10. HIF1 has been recently shown to be the major driver for the skin wound healing response, including vascularization and epithelial and dermal migration. It would be of interest for the authors to discuss the relationship between wound healing and lineage fidelity in light of their work. In particular clusters C8 / C18 at D10 of the assay from this and their previous paper.

Minor point 10. Reviewer #3

We thank the reviewer for raising this point which highlights the relevance of our observations. We have amended the text to emphasise this point. The text now read:

“Interestingly, studies in the skin have shown that, although HIF-1a activation is essential for initiating tissue repair, its sustained activity can impair wound healing; contributing to persistent inflammation, fibrosis, and impaired wound resolution, as seen in chronic wounds^{51,52}. Hence, the HIF1a-SOX9-dependent regenerative blockage points to a remarkably simple self-regulatory process where, in response to injury, plasticity would initially be kept at bay by the same hypoxic signals that promote tissue repair in the first place. This mechanism would ensure that the fate outcome of individual cells contributing to tissue repair is not allocated until later stages of the injury response, when hypoxic barriers are lifted and the final instructive niche is established.” (p19; l15-22).

11. Pg. 10. Line 8-10 Fig S4F, G) These results reveal that, by partially restricting the regenerative capacity of epithelial cells, the fate conversion process becomes more efficient. The experimental design the authors term “less regenerative” is removing EE dermis and showing that it can transdifferentiate more. The term regenerative is quite vague and should be better defined here. Moreover an alternative explanation is that EE dermis contains inhibitory signals acting on the underlying skin dermis.

We apologise for the lack of clarity and we agree with the reviewer that the term regenerative required clarification. Rather than “less regenerative,” we now refer to this as “impaired re-epithelialisation conditions.” The text has been amended to clarify this point. The text reads:

*“Since our data, so far, suggested that **oeSKIN** cells remains locked at a regenerative state, we next asked whether altering the *in vitro* regenerative state, associated dermal re-epithelialisation and *de novo* skin reconstitution, influences the lineage conversion process. To this end, we adapted our *ex vivo* system to limit the ability of the OE to re-epithelialise the dermis. This was achieved by using peeled OE lacking the underlying stroma, which is known to promote cell migration and epithelial expansion during wound healing^{53,54} (**Supplementary Figs. 4H, I**). Indeed, when growing OE under impaired re-epithelialisation conditions, the OE formed **oeHFs** that contained a significantly higher number of CD34+ cells compared to our standard heterotypic approach (**Supplementary Figs. 4J, K**). These results reveal that, by partially restricting the re-epithelialisation capacity of epithelial cells, the fate conversion process becomes more efficient.”* (p11-12; l28-4).

Regarding the effect of the oesophageal stroma in the process, we can confirm that the lack of OE stroma delays re-epithelialisation in a generalised manner not only when growing oesophagus over the dermis. Re-epithelialisation is also inhibited when peeled oesophageal epithelium grows over oesophageal stroma (see oesophageal control culture below for reference; **Response Fig. 7**). This is associated with the lack of basement membrane adhesion, which needs to be re-established for cells to migrate and re-epithelialise the new substrate, independent of its origin. This assay allows for cells to be exposed to the environmental cues without immediately repopulating the substrate.

Response Figure 7: Re-epithelialisation assay comparing unpeeled and peeled oesophageal epithelium 7 days post-culture. Tiled images show brightfield and tdTomato fluorescence of epithelial outgrowth (oeOE) over oesophageal stroma (wildtype). The peeled epithelium exhibits lower expansion compared to the unpeeled condition, indicating reduced re-epithelialisation capacity. Scale bar 1mm.

12. There are numerous typographic errors.

Line 7 “maintain plasticity at check” should be “in check”.

Pg 8 line 1 “...that maker expression” misspelled

We are grateful to the reviewer for pointing this out. A thorough proof-reading of the text has been

done to reduce any typographic and misspelling mistakes.

REFERENCES

- 1 Magaletta, M. E. *et al.* Integration of single-cell transcriptomes and chromatin landscapes reveals regulatory programs driving pharyngeal organ development. *Nat Commun* **13**, 457 (2022). <https://doi.org/10.1038/s41467-022-28067-4>
- 2 Kuwahara, A. *et al.* Delineating the early transcriptional specification of the mammalian trachea and esophagus. *Elife* **9** (2020). <https://doi.org/10.7554/eLife.55526>
- 3 Kumar, N. *et al.* Decoding spatiotemporal transcriptional dynamics and epithelial fibroblast crosstalk during gastroesophageal junction development through single cell analysis. *Nat Commun* **15**, 3064 (2024). <https://doi.org/10.1038/s41467-024-47173-z>
- 4 Yang, Y. *et al.* A spatiotemporal and machine-learning platform facilitates the manufacturing of hPSC-derived esophageal mucosa. *Dev Cell* **60**, 1359-1376 e1310 (2025). <https://doi.org/10.1016/j.devcel.2024.12.030>
- 5 McGinn, J. *et al.* A biomechanical switch regulates the transition towards homeostasis in oesophageal epithelium. *Nat Cell Biol* **23**, 511-525 (2021). <https://doi.org/10.1038/s41556-021-00679-w>
- 6 Chen, L. *et al.* TGFβ1 induces fetal reprogramming and enhances intestinal regeneration. *Cell Stem Cell* **30**, 1520-1537 e1528 (2023). <https://doi.org/10.1016/j.stem.2023.09.015>
- 7 Nusse, Y. M. *et al.* Parasitic helminths induce fetal-like reversion in the intestinal stem cell niche. *Nature* **559**, 109-113 (2018). <https://doi.org/10.1038/s41586-018-0257-1>
- 8 Fernandez Vallone, V. *et al.* Trop2 marks transient gastric fetal epithelium and adult regenerating cells after epithelial damage. *Development* **143**, 1452-1463 (2016). <https://doi.org/10.1242/dev.131490>
- 9 Vercauteren Drubbel, A. *et al.* Reactivation of the Hedgehog pathway in esophageal progenitors turns on an embryonic-like program to initiate columnar metaplasia. *Cell Stem Cell* **28**, 1411-1427 e1417 (2021). <https://doi.org/10.1016/j.stem.2021.03.019>
- 10 Yui, S. *et al.* YAP/TAZ-Dependent Reprogramming of Colonic Epithelium Links ECM Remodeling to Tissue Regeneration. *Cell Stem Cell* **22**, 35-49 e37 (2018). <https://doi.org/10.1016/j.stem.2017.11.001>
- 11 Grommisch, D. *et al.* Defining the contribution of Troy-positive progenitor cells to the mouse esophageal epithelium. *Dev Cell* **59**, 1269-1283 e1266 (2024). <https://doi.org/10.1016/j.devcel.2024.03.011>
- 12 Vercauteren Drubbel, A. & Beck, B. Single-cell transcriptomics uncovers the differentiation of a subset of murine esophageal progenitors into taste buds in vivo. *Sci Adv* **9**, eadd9135 (2023). <https://doi.org/10.1126/sciadv.add9135>
- 13 Doupe, D. P. *et al.* A single progenitor population switches behavior to maintain and repair esophageal epithelium. *Science* **337**, 1091-1093 (2012). <https://doi.org/10.1126/science.1218835>
- 14 Gola, A. & Fuchs, E. Environmental control of lineage plasticity and stem cell memory. *Curr Opin Cell Biol* **69**, 88-95 (2021). <https://doi.org/10.1016/j.ceb.2020.12.015>
- 15 Merrell, A. J. & Stanger, B. Z. Adult cell plasticity in vivo: de-differentiation and transdifferentiation are back in style. *Nat Rev Mol Cell Biol* **17**, 413-425 (2016). <https://doi.org/10.1038/nrm.2016.24>
- 16 Chacon-Martinez, C. A., Koester, J. & Wickstrom, S. A. Signaling in the stem cell niche: regulating cell fate, function and plasticity. *Development* **145** (2018). <https://doi.org/10.1242/dev.165399>
- 17 Blanpain, C. & Fuchs, E. Stem cell plasticity. Plasticity of epithelial stem cells in tissue regeneration. *Science* **344**, 1242281 (2014). <https://doi.org/10.1126/science.1242281>

Response to reviewers' NCOMMS-25-11438

- 18 Ge, Y. *et al.* Stem Cell Lineage Infidelity Drives Wound Repair and Cancer. *Cell* **169**, 636-650 e614 (2017). <https://doi.org/10.1016/j.cell.2017.03.042>
- 19 Que, J., Garman, K. S., Souza, R. F. & Spechler, S. J. Pathogenesis and Cells of Origin of Barrett's Esophagus. *Gastroenterology* **157**, 349-364 e341 (2019). <https://doi.org/10.1053/j.gastro.2019.03.072>
- 20 Zhuang, L. & Fitzgerald, R. C. Cancer development: Origins in the oesophagus. *Nature* **550**, 463-464 (2017). <https://doi.org/10.1038/nature24150>
- 21 Nowicki-Osuch, K. *et al.* Molecular phenotyping reveals the identity of Barrett's esophagus and its malignant transition. *Science* **373**, 760-767 (2021). <https://doi.org/10.1126/science.abd1449>
- 22 Giroux, V. & Rustgi, A. K. Metaplasia: tissue injury adaptation and a precursor to the dysplasia-cancer sequence. *Nat Rev Cancer* **17**, 594-604 (2017). <https://doi.org/10.1038/nrc.2017.68>
- 23 Ge, Y. & Fuchs, E. Stretching the limits: from homeostasis to stem cell plasticity in wound healing and cancer. *Nat Rev Genet* **19**, 311-325 (2018). <https://doi.org/10.1038/nrg.2018.9>
- 24 Hong, Z. X. *et al.* Bioengineered skin organoids: from development to applications. *Mil Med Res* **10**, 40 (2023). <https://doi.org/10.1186/s40779-023-00475-7>
- 25 Sun, H., Zhang, Y. X. & Li, Y. M. Generation of Skin Organoids: Potential Opportunities and Challenges. *Front Cell Dev Biol* **9**, 709824 (2021). <https://doi.org/10.3389/fcell.2021.709824>
- 26 Lee, J. *et al.* Generation and characterization of hair-bearing skin organoids from human pluripotent stem cells. *Nat Protoc* **17**, 1266-1305 (2022). <https://doi.org/10.1038/s41596-022-00681-y>
- 27 Lee, J. *et al.* Hair-bearing human skin generated entirely from pluripotent stem cells. *Nature* **582**, 399-404 (2020). <https://doi.org/10.1038/s41586-020-2352-3>
- 28 Yang, R. *et al.* Generation of folliculogenic human epithelial stem cells from induced pluripotent stem cells. *Nat Commun* **5**, 3071 (2014). <https://doi.org/10.1038/ncomms4071>
- 29 Bonfanti, P. *et al.* Microenvironmental reprogramming of thymic epithelial cells to skin multipotent stem cells. *Nature* **466**, 978-982 (2010). <https://doi.org/10.1038/nature09269>
- 30 Claudinot, S. *et al.* Tp63-expressing adult epithelial stem cells cross lineages boundaries revealing latent hairy skin competence. *Nat Commun* **11**, 5645 (2020). <https://doi.org/10.1038/s41467-020-19485-3>
- 31 Kageyama, T. *et al.* Reprogramming of three-dimensional microenvironments for in vitro hair follicle induction. *Sci Adv* **8**, eadd4603 (2022). <https://doi.org/10.1126/sciadv.add4603>
- 32 Ramovs, V. *et al.* Characterization of the epidermal-dermal junction in hiPSC-derived skin organoids. *Stem Cell Reports* **17**, 1279-1288 (2022). <https://doi.org/10.1016/j.stemcr.2022.04.008>
- 33 Lei, M. *et al.* Epidermal-dermal coupled spheroids are important for tissue pattern regeneration in reconstituted skin explant cultures. *NPJ Regen Med* **8**, 65 (2023). <https://doi.org/10.1038/s41536-023-00340-0>
- 34 Giroux, V. *et al.* Long-lived keratin 15+ esophageal progenitor cells contribute to homeostasis and regeneration. *J Clin Invest* **127**, 2378-2391 (2017). <https://doi.org/10.1172/JCI88941>
- 35 Scott, C. E. *et al.* SOX9 induces and maintains neural stem cells. *Nat Neurosci* **13**, 1181-1189 (2010). <https://doi.org/10.1038/nn.2646>
- 36 Bi, W., Deng, J. M., Zhang, Z., Behringer, R. R. & de Crombrughe, B. Sox9 is required for cartilage formation. *Nat Genet* **22**, 85-89 (1999). <https://doi.org/10.1038/8792>
- 37 Mori-Akiyama, Y. *et al.* SOX9 is required for the differentiation of paneth cells in the intestinal epithelium. *Gastroenterology* **133**, 539-546 (2007). <https://doi.org/10.1053/j.gastro.2007.05.020>
- 38 Seymour, P. A. *et al.* SOX9 is required for maintenance of the pancreatic progenitor cell pool. *Proc Natl Acad Sci U S A* **104**, 1865-1870 (2007). <https://doi.org/10.1073/pnas.0609217104>

- 39 Antoniou, A. *et al.* Intrahepatic bile ducts develop according to a new mode of tubulogenesis regulated by the transcription factor SOX9. *Gastroenterology* **136**, 2325-2333 (2009). <https://doi.org/10.1053/j.gastro.2009.02.051>
- 40 Turcatel, G. *et al.* Lung mesenchymal expression of Sox9 plays a critical role in tracheal development. *BMC Biol* **11**, 117 (2013). <https://doi.org/10.1186/1741-7007-11-117>
- 41 Rockich, B. E. *et al.* Sox9 plays multiple roles in the lung epithelium during branching morphogenesis. *Proc Natl Acad Sci U S A* **110**, E4456-4464 (2013). <https://doi.org/10.1073/pnas.1311847110>
- 42 Saxena, N., Mok, K. W. & Rendl, M. An updated classification of hair follicle morphogenesis. *Exp Dermatol* **28**, 332-344 (2019). <https://doi.org/10.1111/exd.13913>
- 43 Millar, S. E. Molecular mechanisms regulating hair follicle development. *J Invest Dermatol* **118**, 216-225 (2002). <https://doi.org/10.1046/j.0022-202x.2001.01670.x>
- 44 Adam, R. C. *et al.* Pioneer factors govern super-enhancer dynamics in stem cell plasticity and lineage choice. *Nature* **521**, 366-370 (2015). <https://doi.org/10.1038/nature14289>
- 45 Yang, Y. *et al.* The pioneer factor SOX9 competes for epigenetic factors to switch stem cell fates. *Nat Cell Biol* **25**, 1185-1195 (2023). <https://doi.org/10.1038/s41556-023-01184-y>
- 46 Aggarwal, S. *et al.* SOX9 switch links regeneration to fibrosis at the single-cell level in mammalian kidneys. *Science* **383**, eadd6371 (2024). <https://doi.org/10.1126/science.add6371>
- 47 Nowak, J. A., Polak, L., Pasolli, H. A. & Fuchs, E. Hair follicle stem cells are specified and function in early skin morphogenesis. *Cell Stem Cell* **3**, 33-43 (2008). <https://doi.org/10.1016/j.stem.2008.05.009>
- 48 Yang, H., Adam, R. C., Ge, Y., Hua, Z. L. & Fuchs, E. Epithelial-Mesenchymal Micro-niches Govern Stem Cell Lineage Choices. *Cell* **169**, 483-496 e413 (2017). <https://doi.org/10.1016/j.cell.2017.03.038>
- 49 Vidal, V. P. *et al.* Sox9 is essential for outer root sheath differentiation and the formation of the hair stem cell compartment. *Curr Biol* **15**, 1340-1351 (2005). <https://doi.org/10.1016/j.cub.2005.06.064>
- 50 Kadaja, M. *et al.* SOX9: a stem cell transcriptional regulator of secreted niche signaling factors. *Genes Dev* **28**, 328-341 (2014). <https://doi.org/10.1101/gad.233247.113>
- 51 Hong, W. X. *et al.* The Role of Hypoxia-Inducible Factor in Wound Healing. *Adv Wound Care (New Rochelle)* **3**, 390-399 (2014). <https://doi.org/10.1089/wound.2013.0520>
- 52 Ruthenborg, R. J., Ban, J. J., Wazir, A., Takeda, N. & Kim, J. W. Regulation of wound healing and fibrosis by hypoxia and hypoxia-inducible factor-1. *Mol Cells* **37**, 637-643 (2014). <https://doi.org/10.14348/molcells.2014.0150>
- 53 Arwert, E. N., Hoste, E. & Watt, F. M. Epithelial stem cells, wound healing and cancer. *Nat Rev Cancer* **12**, 170-180 (2012). <https://doi.org/10.1038/nrc3217>
- 54 Gurtner, G. C., Werner, S., Barrandon, Y. & Longaker, M. T. Wound repair and regeneration. *Nature* **453**, 314-321 (2008). <https://doi.org/10.1038/nature07039>